# CONVERGENCE OF NEAR-LINEAR WIDTH ReLU NETWORKS WITH UNBALANCED INITIALIZATION

## ABSTRACT

The optimization of neural networks is fundamental in machine learning. While the conjecture that linear width suffices for convergence has been confirmed in some restrictive settings, a significant gap remains for non-smooth ReLU networks, for which prior works require substantially wider, polynomial-width networks. We significantly narrow this gap by developing an analysis that simultaneously achieves near-linear width and accelerated convergence for two-layer ReLU networks with shared first layer and vector-valued outputs. Our results are enabled by a novel unbalanced Gaussian initialization that tightly controls the kernel shift for the non-smooth ReLU activation. We prove that gradient descent (GD) achieves linear convergence for networks with only $\widetilde{\Omega}(Nn/\lambda)$ neurons, where $N$ is the sample size, $n$ is the output dimension, and $\lambda$ denotes the smallest eigenvalue of the (limiting) neural tangent kernel (NTK), which is standard in prior analyses operating in the NTK regime. Within the same framework, Nesterov's accelerated gradient (NAG) attains a provable speedup without sacrificing near-linear width, improving the iteration complexity from $O(n\kappa \log \frac{1}{\epsilon})$ to $O(\sqrt{n\kappa} \log \frac{1}{\epsilon})$, where $\kappa$ is the NTK condition number. Finally, our analysis establishes low-rank adaptivity: by introducing a sketching step at initialization and a subspace analysis, the width requirement reduces to $\widetilde{\Omega}(Nr/\lambda)$ for responses of rank $r \ll n$. By tackling the key analytical hurdles of non-smoothness and vector output with a shared first layer, our work substantially tightens the required width for provable convergence in ReLU networks and brings theory closer to long-standing conjectures.

## 1 INTRODUCTION

The study of neural network training dynamics has seen substantial progress in recent years, particularly in the overparameterized regime where networks can perfectly fit the training data (Du et al., 2018; Allen-Zhu et al., 2019). A central question is the minimum width needed to guarantee that first-order methods converge to a global minimum, and it is widely conjectured that a width linear in the sample size $N$ should suffice. While this has been confirmed for networks with smooth activation functions (Liu et al., 2022a; Bombari et al., 2022), a significant gap remains for the widely used non-smooth ReLU activation. For two-layer ReLU networks, existing theories require widths that scale polynomially with $N$ and depend on data properties, such as the smallest eigenvalue $\lambda$ of the (limiting) neural tangent kernel (NTK, Jacot et al. 2018). Although the best-known requirement has improved from $\Omega(N^6\lambda^{-4})$ (Du et al., 2018) to $\widetilde{\Omega}(N^2\lambda^{-2})$ (Munteanu et al., 2022), this is still far from the conjectured linear dependence on $N$.

We substantially narrow this gap by providing the first analysis showing that a near-linear width is sufficient for the provable convergence of two-layer ReLU networks trained with gradient descent (GD) and Nesterov's accelerated gradient (NAG). Our work tackles the more challenging and practical setting of vector-valued outputs with a shared first layer, which introduces significant analytical hurdles due to the Kronecker product structure of the empirical kernel not present in the simpler scalar-output models.

Our key enabling technique is a novel unbalanced Gaussian initialization scheme. By setting the output layer weights to be larger than the hidden layer by a factor, we bias training toward the NTK regime, which allows controlled feature movement; in contrast, the opposite scaling leads to the random feature regime where the hidden layer is essentially frozen (Nguyen, 2021). The use of

Gaussian-initialized weights provides the necessary technical machinery to derive tight bounds on the evolving kernel matrix, a crucial step for handling non-smooth activations.

We summarize our main contributions as follows:

- **Near-linear width for ReLU networks:** We first prove that a near-linear width of $\widetilde{\Omega}(Nn\lambda^{-1})$ is sufficient for GD to globally optimize two-layer ReLU networks with vector outputs in the NTK regime (Theorem 3.1). This significantly improves upon the previous best-known quadratic bound in this regime (Munteanu et al., 2022).

- **Provable acceleration at near-linear width:** We show that Nesterov's acceleration can be achieved in this challenging setting without requiring a larger network width, something prior analyses of acceleration for ReLU networks did not establish. Our framework confirms that NAG improves the rate from $O(n\kappa \log \frac{1}{\epsilon})$ to $O(\sqrt{n\kappa} \log \frac{1}{\epsilon})$ (Theorem 3.2).

- **Novel analysis for low-rank adaptivity:** For low-rank responses, we introduce a sketching step at initialization. While this reduces the width requirement to depend on the response rank $r \ll n$, it makes the empirical kernel rank-deficient, breaking standard contraction arguments used for convergence. We resolve this with a novel subspace analysis, proving that the training dynamics remain confined to an invariant, contracting subspace, thereby recovering a fast linear rate for a problem that is not strongly convex. This reduces the width requirement to $\widetilde{\Omega}(Nr\lambda^{-1})$ (Theorem 4.1).

Collectively, these results, summarized in Table 1, push the theory of ReLU network optimization significantly closer to long-standing conjectures. The remainder of the paper develops these results, with full proofs and numerical experiments provided in the appendix.

Table 1: Results for two-layer ReLU networks. $N$ is the sample size, $\lambda$ is the smallest NTK eigenvalue, $\kappa$ is the NTK condition number, $\kappa'$ is the condition number of the conjugate kernel (CK, Neal 2012), $\kappa_r$ and $r$ are the condition number and rank of the response matrix, respectively. "RF" denotes the random feature regime where the training dynamics are driven by CK.

| Output | Algorithm | Initialization | Width | Rate | Regime |
|---|---|---|---|---|---|
| Scalar $(n=1)$ | GD (Du et al., 2018) | Random | $\Omega(N^6\lambda^{-4})$ | $O(N^2\lambda^{-2}\log\frac{1}{\epsilon})$ | NTK |
| | GD (Song & Yang, 2019) | Random | $\widetilde{\Omega}(N^4\lambda^{-4})$ | $O(N^2\lambda^{-2}\log\frac{1}{\epsilon})$ | NTK |
| | HB (Wang et al., 2021) | Random | $\widetilde{\Omega}(N^4\kappa^2\lambda^{-4})$ | $O(\sqrt{\kappa}\log\frac{1}{\epsilon})$ | NTK |
| | NAG (Liu et al., 2022b) | Random | $\widetilde{\Omega}(N^4\kappa^2\lambda^{-2})$ | $O(\sqrt{\kappa}\log\frac{1}{\epsilon})$ | NTK |
| | GD (Munteanu et al., 2022) | Coupled | $\widetilde{\Omega}(N^2\lambda^{-2})$ | $O(N^2\lambda^{-2}\log\frac{1}{\epsilon})$ | NTK |
| | GD (Ours, Theorem 3.1) | Unbalanced | $\widetilde{\Omega}(N\lambda^{-1})$ | $O(\kappa\log\frac{1}{\epsilon})$ | NTK |
| | NAG (Ours, Theorem 3.2) | Unbalanced | $\widetilde{\Omega}(N\lambda^{-1})$ | $O(\sqrt{\kappa}\log\frac{1}{\epsilon})$ | NTK |
| $n$-dim. Vector | GD (Nguyen, 2021) | Unbalanced | $\Omega(N)$ | $O(\kappa'\log\frac{1}{\epsilon})$ | RF |
| | GD (Ours, Theorem 3.1) | Unbalanced | $\widetilde{\Omega}(Nn\lambda^{-1})$ | $O(n\kappa\log\frac{1}{\epsilon})$ | NTK |
| | NAG (Ours, Theorem 3.2) | Unbalanced | $\widetilde{\Omega}(Nn\lambda^{-1})$ | $O(\sqrt{n\kappa}\log\frac{1}{\epsilon})$ | NTK |
| | NAG (Ours, Theorem 4.1) | Unbalanced | $\widetilde{\Omega}(Nr\kappa_r^2\lambda^{-1})$ | $O(\sqrt{r\kappa}\kappa_r\log\frac{1}{\epsilon})$ | NTK |

## 1.1 ADDITIONAL RELATED WORKS

**Overparameterization and training regimes.** In the infinite-width limit, training suitably scaled neural networks by gradient descent reduces to kernel regression with a fixed NTK, explaining linearized ("lazy") dynamics (Jacot et al., 2018; Lee et al., 2019; Chizat et al., 2019). Different scalings place training in the random feature (or conjugate kernel, CK, Neal 2012) regime, an NTK-leaning regime that permits controlled feature movement, or richer feature-learning regimes. For two-layer networks with *smooth* activations, linear widths are attainable in various settings (Bombari et al., 2022; Liu et al., 2022a). In contrast, for the *non-smooth* ReLU activation, classical NTK-type analyses progressed from $\Omega(N^6)$ (Du et al., 2018) to $\widetilde{\Omega}(N^4)$ (Song & Yang, 2019; Oymak & Soltanolkotabi, 2020) and further to $\widetilde{\Omega}(N^2)$ via coupled initialization (Munteanu et al., 2022), yet still far from the conjectured linear width. Nguyen (2021) obtain linear width in the random feature regime by unbalancing toward a zero output layer. By contrast, our unbalanced scaling biases

training toward the NTK-leaning regime that allows more substantial hidden layer movement, which is often associated with improved representation learning; this controlled movement is crucial for bounding kernel shift with non-smooth ReLU activation, especially in the vector-valued case.

**Width and NTK eigenvalues.** Our near-linear width bounds scale with the smallest (limiting) NTK eigenvalue $\lambda$, which is standard in NTK regime analyses. Recent results give lower bounds or concentration for empirical NTK eigenvalues at initialization (scalar outputs) and clarify when $\lambda$ remains well-conditioned under mild geometric conditions (Karhadkar et al., 2024; Yang et al., 2024). These insights help justify $\widetilde{\Omega}(Nn/\lambda)$-type requirements in our vector-valued setting, where we further control *kernel shift* during training.

**Vector-valued outputs with a shared first layer.** Unlike training $n$ independent scalar heads, vector outputs with a shared first layer induce a *matrix-valued* kernel, which admits a Kronecker product structure when expressed in the standard kernel matrix form. This is classical in vector-valued RKHS and multi-output Gaussian processes (Micchelli & Pontil, 2005; Alvarez et al., 2012). Handling this coupling makes concentration and kernel-shift control more delicate than in the scalar case, and is central to our analysis.

**Acceleration and low-rank adaptivity.** Momentum methods provably accelerate NTK-style training for ReLU networks (Wang et al., 2021; Liu et al., 2022b; Liao & Kyrillidis, 2024), but prior proofs require widths at least $\widetilde{\Omega}(N^4)$ and focus on scalar outputs. We obtain NAG's $\sqrt{\kappa}$ speedup at *near-linear* width in the vector/shared-feature setting. Moreover, with sketching at initialization, we establish *low-rank adaptivity*, reducing width from $\widetilde{\Omega}(Nn/\lambda)$ to $\widetilde{\Omega}(Nr/\lambda)$ for rank-$r$ responses via a subspace analysis. Our sketching approach is in line with randomized sketching methods (Halko et al., 2011; Pilanci & Wainwright, 2016), and complements tailored-initialization analyses in matrix factorization and linear networks (Ward & Kolda, 2023; Xu et al., 2024).

## 2 PRELIMINARIES

Throughout this paper, $\|\cdot\|$ denotes the Euclidean norm of vectors or spectral norm of matrices, $\|\cdot\|_{\mathrm{F}}$ denotes the Frobenius norm of matrices. For any matrix, $[\cdot]_{i,:}$ and $[\cdot]_{:,i}$ denote the $i$-th row and column, respectively, $\sigma_i(\cdot)$ denotes the $i$-th largest singular value. For a square matrix, $\lambda_i(\cdot)$ denotes its $i$-th largest eigenvalue. For a matrix $\mathbf{X}$, we use $\mathrm{col}(\mathbf{X})$, $\mathrm{row}(\mathbf{X})$ and $\ker(\mathbf{X})$ to denote its column space, row space and null space respectively, and denote its condition number as $\kappa(\mathbf{X})$. We use $\odot$ and $\otimes$ to denote the Hadamard and Kronecker product, $\oplus$ to denote the direct sum of vector spaces, $\mathrm{vec}(\cdot)$ to denote the column-first vectorization of a matrix, and $\mathrm{diag}(\cdot)$ to denote the diagonal matrix from a vector. For scalars $a, b$, we occasionally use $a \vee b$ and $a \wedge b$ to denote their maximum and minimum. We use $\lesssim$ and $\gtrsim$ to omit absolute constants.

We consider training vector-valued two-layer ReLU networks for regression under the square loss:

$$\min_{\mathbf{V}, \mathbf{W}} \mathcal{L}(\mathbf{V}, \mathbf{W}) = \frac{1}{2} \left\| \mathbf{Y} - \mathbf{V}\sigma(\mathbf{W}^\top \mathbf{X}) \right\|_{\mathrm{F}}^2. \tag{1}$$

Here, $\mathbf{X} = (\mathbf{x}_1, \mathbf{x}_2, \ldots, \mathbf{x}_N) \in \mathbb{R}^{d \times N}$ denotes the input data where $d$ is the input dimension and $N$ is the sample size, $\mathbf{Y} = (\mathbf{y}_1, \mathbf{y}_2, \ldots, \mathbf{y}_N) \in \mathbb{R}^{n \times N}$ denotes the responses where each response is an $n$-dimensional vector, $\sigma(\cdot) = \max(\cdot, 0)$ denotes the element-wise ReLU activation, $\mathbf{V} \in \mathbb{R}^{n \times 2m}$ and $\mathbf{W} \in \mathbb{R}^{d \times 2m}$ are the output and hidden layers of neural network with even number $2m$ of neurons. We assume the data is normalized as in Du et al. (2018; 2019); Munteanu et al. (2022).

**Assumption 2.1.** For any $j \in [N]$, $\|\mathbf{x}_j\| = 1$, $\|\mathbf{y}_j\| \leq 1$.

### 2.1 GRADIENT DESCENT AND NESTEROV'S ACCELERATION

Let $\mathbf{R}(\mathbf{V}, \mathbf{W}) = \mathbf{V}\sigma(\mathbf{W}^\top \mathbf{X}) - \mathbf{Y}$ denote the residual. The (sub)gradient of the loss function in (1) w.r.t. $\mathbf{V}$ and $\mathbf{W}$ are:

$$\nabla_V \mathcal{L}(\mathbf{V}, \mathbf{W}) = \mathbf{R}(\mathbf{V}, \mathbf{W})\sigma(\mathbf{X}^\top \mathbf{W}), \quad \nabla_W \mathcal{L}(\mathbf{V}, \mathbf{W}) = \mathbf{X}((\mathbf{R}(\mathbf{V}, \mathbf{W})^\top \mathbf{V}) \odot \mathbb{1}[\mathbf{X}^\top \mathbf{W} \geq 0]),$$

where $\mathbb{1}[\cdot]$ denotes entry-wise indicator. At the $t$-th iteration, GD performs the following update:

$$\mathbf{V}_{t+1} = \mathbf{V}_t - \eta \nabla_V \mathcal{L}(\mathbf{V}_t, \mathbf{W}_t), \quad \mathbf{W}_{t+1} = \mathbf{W}_t - \eta \nabla_W \mathcal{L}(\mathbf{V}_t, \mathbf{W}_t), \tag{2}$$

where $\eta$ is the step size. For NAG, we take the form initially designed for smooth strongly convex loss (Nesterov, 2013) with a fixed momentum parameter $\beta$ for all iterations:

$$\mathbf{V}_{t+1} = (1+\beta)(\mathbf{V}_t - \eta\nabla_V\mathcal{L}(\mathbf{V}_t, \mathbf{W}_t)) - \beta(\mathbf{V}_{t-1} - \eta\nabla_V\mathcal{L}(\mathbf{V}_{t-1}, \mathbf{W}_{t-1})),$$
$$\mathbf{W}_{t+1} = (1+\beta)(\mathbf{W}_t - \eta\nabla_W\mathcal{L}(\mathbf{V}_t, \mathbf{W}_t)) - \beta(\mathbf{W}_{t-1} - \eta\nabla_W\mathcal{L}(\mathbf{V}_{t-1}, \mathbf{W}_{t-1})), \quad (3)$$

where we set $\mathbf{V}_{-1} = \mathbf{V}_0$ and $\mathbf{W}_{-1} = \mathbf{W}_0$. When $\beta = 0$, NAG (3) reduces to GD (2). Therefore, we can study the convergence rate of GD and NAG in a unified form.

## 2.2 LIMITING AND EMPIRICAL KERNELS

Our analysis relies on the neural tangent kernel (NTK, Jacot et al. 2018). We make the following standard assumption on the limiting NTK $\overline{\mathbf{K}} \in \mathbb{R}^{N \times N}$ of two-layer ReLU networks, defined as $\overline{\mathbf{K}}_{j,k} = \mathbb{E}_{\mathbf{w}\sim\mathcal{N}(0,\mathbf{I}_d)}\left[\mathbf{x}_j^\top \mathbf{x}_k \cdot \mathbb{1}[\mathbf{w}^\top\mathbf{x}_j \geq 0, \mathbf{w}^\top\mathbf{x}_k \geq 0]\right]$.

**Assumption 2.2.** Assume $\overline{K}$ has smallest eigenvalue $\lambda := \lambda_N(\overline{\mathbf{K}}) > 0$, and thus its condition number $\kappa := \kappa(\overline{\mathbf{K}}) < \infty$.

Assumption 2.2 holds when no two data points are parallel (Du et al., 2018). The value of $\lambda$ depends on data collinearity and remains well-behaved for common distributions (see Remark 3.4).

Since we consider training both layers, the limiting conjugate kernel (CK, Neal 2012) $\overline{\mathbf{G}} := \mathbb{E}_{\mathbf{w}\sim\mathcal{N}(0,\mathbf{I}_d)}[\sigma(\mathbf{X}^\top\mathbf{w})\sigma(\mathbf{w}^\top\mathbf{X})]$ is also involved. According to the original definition in Jacot et al. (2018), the full NTK for training both layers is $\overline{\mathbf{K}} + \overline{\mathbf{G}}$. We refer to $\overline{\mathbf{K}}$ as the NTK to align with prior works that only consider optimizing the hidden layer (Du et al., 2018; Munteanu et al., 2022).

In our finite-width setting, the empirical NTK and CK matrices at iteration $t$ are defined as $\mathbf{K}_t := \sum_{i=1}^{2m} \mathbf{B}_{i,t}^\top \otimes \mathbf{A}_{i,t}$ and $\mathbf{G}_t := (\sigma(\mathbf{X}^\top\mathbf{W}_t)\sigma(\mathbf{W}_t^\top\mathbf{X})) \otimes \mathbf{I}_n$ respectively, where $\mathbf{A}_{i,t} = [\mathbf{V}_t]_{:,i}[\mathbf{V}_t]_{:,i}^\top$ and $\mathbf{B}_{i,t} = \operatorname{diag}(\mathbb{1}[\mathbf{W}_t^\top\mathbf{X} \geq 0]_{i,:})\mathbf{X}^\top\mathbf{X}\operatorname{diag}(\mathbb{1}[\mathbf{W}_t^\top\mathbf{X} \geq 0]_{i,:})$.

## 3 RESULTS FOR GENERAL VECTOR RESPONSES

We first introduce our unbalanced initialization scheme for general vector responses:

$$\mathbf{V}_0 = (c_1\mathbf{\Phi}, -c_1\mathbf{\Phi}), \quad \mathbf{W}_0 = (c_2\mathbf{\Psi}, c_2\mathbf{\Psi}), \quad (4)$$

where $\mathbf{\Phi} \in \mathbb{R}^{n \times m}$ and $\mathbf{\Psi} \in \mathbb{R}^{d \times m}$ are standard Gaussian random matrices, and $c_1, c_2 > 0$ are scaling factors. The first and second halves of neurons are coupled at initialization, ensuring the initial network output is zero, allowing flexible choice of $c_1$ and $c_2$ without changing the initial loss.

The key to our results is to make this initialization sufficiently *unbalanced* by imposing specific scaling conditions on $c_1$ and $c_2$. For GD, we require the following conditions:

$$\frac{c_1^3}{c_2} \geq \widetilde{\Omega}\left(\frac{\sqrt{Nd}(\sqrt{N} \vee \kappa^4(\mathbf{K}_0))}{\sqrt{n}\lambda\kappa^3(\mathbf{K}_0)}\right), \quad \frac{c_1}{c_2} \geq \widetilde{\Omega}\left(\frac{\sqrt{d}}{\sqrt{n}\kappa(\mathbf{K}_0)}\right), \quad c_1c_2 \geq \widetilde{\Omega}\left(\frac{\sqrt{Nn}\kappa(\mathbf{K}_0)}{\lambda}\right). \quad (5)$$

Since changing the scaling factors does not affect the condition number of $\mathbf{K}_0$, there is no recursive definition in (5). With this unbalanced scaling, we establish the main result for GD.

**Theorem 3.1** (General vector responses, GD). *Consider a two-layer ReLU network with initialization* (4) *satisfying the scaling conditions in* (5)*. If the network width is near-linear,*

$$m \geq \max\left\{\Omega\left(\frac{Nn}{\lambda} \cdot \operatorname{poly}\left(\log\frac{Nnd\kappa}{\delta}\right)\right), \Omega\left(\frac{1}{\delta}\right)\right\},$$

*then for any $\delta \in (0, 1)$, GD with step size $\eta = 1/L$ achieves $\mathcal{L}(\mathbf{V}_T, \mathbf{W}_T) \leq \epsilon \cdot \mathcal{L}(\mathbf{V}_0, \mathbf{W}_0)$ in*

$$T = O\left(n\kappa \cdot \log(1/\epsilon)\right)$$

*iterations with probability at least $1 - \delta$, where $L = \lambda_1(\mathbf{K}_0)$ is the largest eigenvalue of the initial empirical NTK, $\lambda$ and $\kappa$ are the smallest eigenvalue and condition number of the limiting NTK (Assumption 2.2), respectively.*

Theorem 3.1 shows that if the output layer is sufficiently large compared to the hidden layer, then GD exhibits global linear convergence at a near-linear width.

For NAG, we have similar initialization scalings that lead to acceleration at the same width:

$$\frac{c_1^3}{c_2} \geq \widetilde{\Omega}\left(\frac{N\sqrt{d}}{\sqrt{n}\lambda}\right), \quad \frac{c_1}{c_2} \geq \widetilde{\Omega}\left(\frac{\sqrt{d}}{\sqrt{n}}\right), \quad c_1 c_2 \geq \widetilde{\Omega}\left(\frac{\sqrt{Nn}}{\lambda}\right). \tag{6}$$

**Theorem 3.2** (General vector responses, NAG). *Consider a two-layer ReLU network with initialization* (4) *satisfying the scaling conditions in* (6). *If the network width is near-linear,*

$$m \geq \max\left\{\Omega\left(\frac{Nn}{\lambda} \cdot \mathrm{poly}\left(\log \frac{Nnd\kappa}{\delta}\right)\right), \Omega\left(\frac{1}{\delta}\right)\right\},$$

*then for any $\delta \in (0,1)$, NAG with step size $\eta = 1/L$ and momentum $\beta = (\sqrt{L} - \sqrt{\mu})/(\sqrt{L} + \sqrt{\mu})$ finds a solution satisfying $\mathcal{L}(\mathbf{V}_T, \mathbf{W}_T) \leq \epsilon \cdot \mathcal{L}(\mathbf{V}_0, \mathbf{W}_0)$ in*

$$T = O\left(\sqrt{n\kappa} \cdot \log(1/\epsilon)\right)$$

*iterations with probability at least $1 - \delta$, where $L = \lambda_1(\mathbf{K}_0)$, $\mu = \lambda_{Nn}(\mathbf{K}_0)$.*

Theorem 3.2 shows that NAG accelerates the convergence rate from $O(n\kappa)$ to $O(\sqrt{n\kappa})$. It simultaneously achieves the best-known convergence rate and smallest-known width requirement for ReLU networks. Numerical evidence of such an acceleration is provided in Figure 1 in Appendix A.

**Remark 3.3.** Our unbalanced scaling allows the hidden layer to change by a factor of $O\left(\frac{\lambda}{Nn\sqrt{d}}\right)$. For comparison, the opposite scaling in Nguyen (2021) leads to the random feature regime where the hidden layer weight is only allowed to change by a factor of $O\left(\frac{\lambda_N^3(\overline{\mathbf{G}})}{N^3\sqrt{d}}\right)$, essentially reducing the problem to linear regression with slower convergence (see Figures 3 and 4 in Appendix A).

**Remark 3.4.** The smallest NTK eigenvalue $\lambda := \lambda_N(\overline{\mathbf{K}})$ depends on the data collinearity $\delta := \min_{i \neq j} \min(\|\mathbf{x}_i - \mathbf{x}_j\|, \|\mathbf{x}_i + \mathbf{x}_j\|)$. By Lemma 7 in Karhadkar et al. (2024), $\lambda \gtrsim \left(1 + \frac{d\log(1/\delta)}{\log d}\right)^{-3}\delta^2$. For Gaussian or uniform spherical data (equivalent after normalization), $\delta = \Omega(N^{-2/d})$ with high probability, and thus $\lambda = \widetilde{\Omega}(1)$ whenever $\log N \lesssim d$, which holds in the typical high-dimensional regime. Therefore, the dependence on $\lambda$ would not dominate the dependence on $N$. Meanwhile, it is difficult to derive an explicit dependence for general distributions.

### 3.1 PROOF OVERVIEW FOR GENERAL RESPONSES

Our proof strategy is to show that the residual dynamics closely approximates a fixed, contracting linear system, a common approach in the NTK literature (Wang et al., 2021; Liu et al., 2022b). However, its success hinges on tightly controlling the deviation from the linear system, and prior analyses fall short in our setting due to two interacting technical hurdles. First, the shared vector-output layer induces a complex Kronecker product structure in the initial kernel $\mathbf{K}_0$. To overcome this, we leverage matrix Chernoff bounds to prove that a near-linear width is sufficient for the kernel to be well-conditioned at initialization, establishing a stable starting point for the dynamics (Lemma 3.6). Second, and more critically, the non-smooth ReLU activation causes discrete shifts in the kernel matrix during training, and a naive analysis of this process leads back to the suboptimal $\widetilde{\Omega}(N^2)$ width requirement (Munteanu et al., 2022). Our key innovation, the *unbalanced Gaussian initialization*, provides the necessary machinery to tightly control this evolution, allowing us to break this quadratic barrier (Lemma 3.9). Building on these two pillars, the proof proceeds by an inductive argument to show that the kernel shift and other error terms remain bounded, ensuring the dynamics stay approximately contractive and converge linearly.

#### 3.1.1 INITIAL CONDITION

Firstly, we have $\|\mathbf{r}_0\| \equiv \|\mathbf{Y}\|_{\mathrm{F}}$ by coupled initialization. Define $R_V := \max_{i \in [2m]} \|[\mathbf{V}_0]_{:,i}\|$, $R_W := \max_{i \in [2m]} \|[\mathbf{W}_0]_{:,i}\|$. We have the following bounds with proof in Appendix C.1.

**Lemma 3.5.** *Suppose $\mathbf{V}_0$ and $\mathbf{W}_0$ are as* (4), *then with probability at least $1 - \frac{2}{m} - 2e^{-\frac{9m(n \wedge d)}{64}}$,*

$$\frac{1}{2}c_1\sqrt{n} \leq R_V \leq c_1(\sqrt{n} + 2\sqrt{\log m}),$$
$$\frac{1}{2}c_2\sqrt{d} \leq R_W \leq c_2(\sqrt{d} + 2\sqrt{\log m}).$$

By Lemma 3.5, $R_V$ and $R_W$ are roughly $O(c_1\sqrt{n})$ and $O(c_2\sqrt{d})$.

The convergence rate depends on the condition number of the initial empirical NTK, $\mathbf{K}_0$. We show that a width of $\widetilde{\Omega}(Nn)$ suffices to bound its extreme eigenvalues under our initialization.

**Lemma 3.6** (Eigenvalues of $\mathbf{K}_0$). *Suppose $\mathbf{V}_0$ and $\mathbf{W}_0$ are as* (4), $\delta \in (0,1)$, *if the width satisfies*

$$m = \max\left\{\widetilde{\Omega}\left(\frac{\|\mathbf{X}\|^2 n}{\lambda}\log\frac{Nn}{\delta}\right), \Omega\left(\frac{1}{\delta}\right)\right\},$$

*then $\lambda_{Nn}(\mathbf{K}_0) \geq \frac{1}{2}c_1^2 m\lambda$ and $\lambda_1(\mathbf{K}_0) \leq 9c_1^2 mn\lambda_1(\overline{\mathbf{K}})$ with probability at least $1-\delta$.*

The proof is provided in Appendix C.2. Unlike prior works that use entry-wise deviation bounds requiring quadratic width (Du et al., 2018; Song & Yang, 2019), we achieve near-linear width by exploiting the Kronecker product structure and positive-semidefiniteness of the summands via matrix Chernoff bound (Proposition B.5). While similar ideas have been adopted for scalar-valued networks at initialization (Yang et al., 2024; Karhadkar et al., 2024; Kim & Pilanci, 2024), vector-valued networks pose additional challenges due to the Kronecker structure and unbounded support of Gaussians. By Lemma 3.6, $\mathbf{K}_0$ has condition number $O(n\kappa)$ with high probability. The dependence on $n$ arises from the largest eigenvalue and is likely to be necessary: ignoring $\mathbf{B}_{i,0}$ and considering only $\mathbf{A}_{i,0}$ reduces the problem to the concentration of Gaussian sample covariance, where this dimensional dependence appears naturally (Koltchinskii & Lounici, 2017). Lemma 3.6 allows even smaller width than $\widetilde{\Omega}(Nn)$ for initialization when $\|\mathbf{X}\| \ll \|\mathbf{X}\|_{\mathrm{F}}$. For uniformly distributed data where $\|\mathbf{X}\|^2 = O(N/d)$ (Vershynin, 2010), the width requirement become $m = \widetilde{\Omega}(Nn/d)$, matching the expressivity bound in Zhang et al. (2021) up to logarithmic factors.

### 3.1.2 TRACKING THE RESIDUAL DYNAMICS

We now track the residual dynamics. Let $\mathbf{P}_t = \mathbf{V}_{t+1} - \mathbf{V}_t$ and $\mathbf{Q}_t = \mathbf{W}_{t+1} - \mathbf{W}_t$ denote the updates at each step, $\mathbf{R}_t := \mathbf{R}(\mathbf{V}_t, \mathbf{W}_t)$ denote the residual and $\mathbf{r}_t = \mathrm{vec}(\mathbf{R}_t)$ denote vectorized residual. For $j \in [N]$, let $S_j$ be the indices of neurons that do not change their activation patterns on $\mathbf{x}_j$, i.e., $S_j = \left\{i \in [2m] \mid \mathbb{1}[\mathbf{W}_t^\top\mathbf{X} \geq 0]_{i,j} \equiv \mathbb{1}[\mathbf{W}_0^\top\mathbf{X} \geq 0]_{i,j}, \forall t \geq 0\right\}$, and let $S_j^\perp = [2m] \setminus S_j$.

**Proposition 3.7** (Dynamics). *NAG admits the following residual dynamics:*

$$\begin{pmatrix}\mathbf{r}_{t+1} \\ \mathbf{r}_t\end{pmatrix} = \mathbf{T}_{\mathrm{NAG}}\begin{pmatrix}\mathbf{r}_t \\ \mathbf{r}_{t-1}\end{pmatrix} + \begin{pmatrix}\boldsymbol{\xi}_t \\ 0\end{pmatrix}, \tag{7}$$

*where $\boldsymbol{\xi}_t$ is specified as* (10) *in Appendix D and*

$$\mathbf{T}_{\mathrm{NAG}} := \begin{pmatrix}(1+\beta)(\mathbf{I}_{Nn} - \eta\mathbf{K}_0) & -\beta(\mathbf{I}_{Nn} - \eta\mathbf{K}_0) \\ \mathbf{I}_{Nn} & 0\end{pmatrix}.$$

*When $\beta = 0$,* (7) *corresponds to the GD dynamics $\mathbf{r}_{t+1} = \mathbf{T}_{\mathrm{GD}}\mathbf{r}_t + \boldsymbol{\xi}_t$, where $\mathbf{T}_{\mathrm{GD}} = \mathbf{I}_{Nn} - \eta\mathbf{K}_0$.*

The proof is provided in Appendix D.1. By Proposition 3.7, the evolution of the residual is mainly driven by the transition matrices $\mathbf{T}_{\mathrm{GD}}$ and $\mathbf{T}_{\mathrm{NAG}}$ corresponding to the reference systems for GD and NAG. By Lemma 3.6, $\mathbf{K}_0$ has full rank and bounded condition number. Therefore, with proper choice of $\eta$ and $\beta$, $\mathbf{T}_{\mathrm{GD}}$ and $\mathbf{T}_{\mathrm{NAG}}$ are contraction maps. We provide the proof in Appendix D.2.

**Lemma 3.8** (Contraction). *Denote $L := \lambda_1(\mathbf{K}_0)$, $\mu := \lambda_{Nn}(\mathbf{K}_0)$. By setting $\beta = 0$ and $\eta = \frac{1}{L}$ it holds $\|\mathbf{T}_{\mathrm{GD}}\| \leq 1 - \frac{1}{\kappa(\mathbf{K}_0)}$. By setting $\beta = \frac{\sqrt{L}-\sqrt{\mu}}{\sqrt{L}+\sqrt{\mu}}$ and $\eta = \frac{1}{L}$ it holds $\|\mathbf{T}_{\mathrm{NAG}}\| \leq 1 - \frac{1}{\sqrt{\kappa(\mathbf{K}_0)}}$.*

### 3.1.3 CONTROLLING ERROR TERMS

We prove convergence via an inductive argument. We assume that up to iteration $t$, the weights remain close to their initialization and the residual contracts exponentially. We then show that these conditions continue to hold at iteration $t + 1$. For this purpose, we bound the deviation from the reference system, including the kernel shift in NTK, $\mathbf{K}_t - \mathbf{K}_0$, and the whole CK, $\mathbf{G}_t$, by leveraging the assumption that the weights remain close to their unbalanced initialization.

**Lemma 3.9** (Kernel shift)**.** *For any $\delta \in (0,1)$, $R_1 > 0$ and $R_2 \in (0, c_2)$, suppose the width satisfies*

$$m \geq \Omega\left(\frac{c_2}{R_2} \cdot \log^2\left(\frac{2nm}{\delta}\right)\right).$$

*Then with probability at least $1 - \delta$, following inequalities hold for any $\mathbf{V}_t$ and $\mathbf{W}_t$ such that $\|[\mathbf{V}_t]_{:,i} - [\mathbf{V}_0]_{:,i}\| \leq R_1$, $\|[\mathbf{W}_t]_{:,i} - [\mathbf{W}_0]_{:,i}\| \leq R_2$, $\forall i \in [2m]$:*

$$\|\mathbf{K}_0 - \mathbf{K}_t\| \leq 6mNR_V^2(R_2/c_2) + 4mNR_1(2R_V + R_1),$$

$$\|\mathbf{G}_t\| \leq 2m\|\mathbf{X}\|^2(R_W + R_2)^2.$$

*Moreover, if $R_V^2 \cdot \frac{R_2}{c_2} \geq R_1(2R_V + R_1)$, then $\|\mathbf{K}_0 - \mathbf{K}_t\| \leq 10mNR_V^2(R_2/c_2)$.*

The proof is provided in Appendix E.1. As mentioned earlier, bounding the kernel shift for vector-valued ReLU networks is non-trivial. Naive reduction to the scalar output case could yield worse dependencies on $N$ and $\frac{R_2}{c_2}$, making linear width insufficient. By Lemma 3.9, the kernel shift can be properly controlled by $\|\mathbf{r}_t\|$, thus shrinking exponentially given the induction condition.

We then bound the error induced by the change in the activation pattern and higher-order error terms. Here, higher-order terms refer to those whose norm can be bounded by degree-2 monomials of $\|\mathbf{r}_t\|$, $\|\mathbf{P}_t\|_{\mathrm{F}}$ and $\|\mathbf{Q}_t\|_{\mathrm{F}}$. For GD, we have $\|\mathbf{P}_t\|_{\mathrm{F}} = \|\nabla_V \mathcal{L}(\mathbf{V}_t, \mathbf{W}_t)\|_{\mathrm{F}} = O(\|\mathbf{r}_t\|)$, and similarly $\|\mathbf{Q}_t\|_{\mathrm{F}} = O(\|\mathbf{r}_t\|)$, hence they are properly controlled under the induction assumption. For NAG, by recursion we have

$$\mathbf{P}_t = -\eta\nabla\mathcal{L}(\mathbf{V}_t, \mathbf{W}_t) + \eta\beta^{t+1}\nabla\mathcal{L}(\mathbf{V}_0, \mathbf{W}_0) - \eta\sum_{s=0}^{t}\beta^{t+1-s}\nabla\mathcal{L}(\mathbf{V}_s, \mathbf{W}_s).$$

Therefore, as long as $\beta \leq \theta^2$, by triangle inequality it holds $\|\mathbf{P}_t\|_{\mathrm{F}} = O(\theta^t)$. Similarly we have $\|\mathbf{Q}_t\|_{\mathrm{F}} = O(\theta^t)$, and thus higher-order terms can be controlled in $O(\theta^{2t})$.

For the change in activation pattern, we first call the following result from Wang et al. (2021).

**Lemma 3.10** (Lemma 11 in Wang et al. (2021))**.** *Suppose $\mathbf{W}_0$ is as (4), $R_2 \in (0, c_2)$, $\|[\mathbf{W}_t]_{:,i} - [\mathbf{W}_0]_{:,i}\| \leq R_2$ for all $t \geq 0$. Denote $\bar{S} := \max_{j \in [N]}|S_j^\perp|$. Then with probability at least $1 - N \cdot \exp(-mR_2/c_2)$,*

$$|S_j^\perp| \leq 8mR_2/c_2, \ \forall j \in [N].$$

For completeness, we provide a proof in Appendix G.2. Lemma 3.10 relates the pattern change to the hidden layer weight change. The error terms involving $\mathbf{M}^\perp$ and $\boldsymbol{\varphi}_t$ in Proposition 3.7 can be bounded in terms of $\bar{S}$ and factors of $O(\|\mathbf{r}_t\|)$ or $O(\|\mathbf{Q}_t\|_{\mathrm{F}})$, which are of $O(\theta^t)$. Wrapping up these terms, we have the following bound on the error $\|\boldsymbol{\xi}_t\|$.

**Lemma 3.11.** *Suppose $0 \leq \beta \leq \theta^2 \leq \theta < 1$. If for all $0 \leq s \leq t$ and $i \in [2m]$, $\|\mathbf{r}_s\| \leq C_1\theta^s\|\mathbf{Y}\|_{\mathrm{F}}$, $\|[\mathbf{V}_s]_{:,i} - [\mathbf{V}_0]_{:,i}\| \leq R_1$, $\|[\mathbf{W}_s]_{:,i} - [\mathbf{W}_0]_{:,i}\| \leq R_2$ and the events in Lemma 3.9 happen, then*

$$\|\boldsymbol{\xi}_t\| \leq \frac{2\theta^t}{1-\theta}\eta C_1\sqrt{\bar{S}}C_V^2\|\mathbf{Y}\|_{\mathrm{F}} + 3\eta\bar{S}N(R_V + R_1)^2C_1\theta^t\|\mathbf{Y}\|_{\mathrm{F}}$$

$$+ \frac{8\theta^{2t}}{(1-\theta)^2}\eta^2C_1^2mC_VC_W\|\mathbf{X}\|_{\mathrm{F}}\|\mathbf{Y}\|_{\mathrm{F}}^2 + 6\left(5N\sqrt{\frac{R_2}{c_2}}R_V^2 + C_W^2\right)m\eta C_1\theta^t\|\mathbf{Y}\|_{\mathrm{F}},$$

*where $C_V := \|\mathbf{X}\|_{\mathrm{F}}(R_V + R_1)$, $C_W := \|\mathbf{X}\|_{\mathrm{F}}(R_W + R_2)$.*

Moreover, the weights at the next step are still close to their initialization under some additional conditions. We provide proofs for Lemmas 3.11 and 3.12 in Appendices G.3 and G.4.

**Lemma 3.12.** *With the same assumptions as Lemma 3.11, if*

$$\eta C_1 C_W\|\mathbf{Y}\|_{\mathrm{F}} \leq (1-\theta)^2 R_1, \quad \eta C_1 C_V\|\mathbf{Y}\|_{\mathrm{F}} \leq (1-\theta)^2 R_2,$$

*then for all $i \in [2m]$, $\|[\mathbf{V}_{t+1}]_{:,i} - [\mathbf{V}_0]_{:,i}\| \leq R_1$, and $\|[\mathbf{W}_{t+1}]_{:,i} - [\mathbf{W}_0]_{:,i}\| \leq R_2$.*

Finally, by properly choosing the width $m$ and scaling factors $c_1$ and $c_2$ in (4), we can verify that all the induction conditions are satisfied, and we get the GD convergence rate in Theorem 3.1. The full proofs for GD and NAG in Theorems 3.1 and 3.2 are provided in Appendices G.5 and G.6.

**Remark 3.13.** While initialization only requires $\widetilde{\Omega}(\|\mathbf{X}\|^2)$ width, our full convergence proof requires $\widetilde{\Omega}(N)$ due to the dependence on $N$ in Lemmas 3.9, 3.11 and 3.12.

## 4 RESULTS FOR LOW-RANK RESPONSES

In the general results of Section 3, the network width and convergence rate depend on the ambient output dimension $n$. This can be suboptimal when the response matrix $\mathbf{Y}$ has a low-rank structure (i.e., $\mathrm{rank}(\mathbf{Y}) =: r \ll n$). To exploit this structure, we introduce a sketching step at initialization:

$$\mathbf{V}_0 = (c_1 \mathbf{Y}\boldsymbol{\Phi}, -c_1 \mathbf{Y}\boldsymbol{\Phi}), \quad \mathbf{W}_0 = (c_2 \boldsymbol{\Psi}, c_2 \boldsymbol{\Psi}), \tag{8}$$

where $\boldsymbol{\Phi} \in \mathbb{R}^{N \times m}$. Compared to (4), the initialization of $\mathbf{V}_0$ now involves a random projection of the response matrix $\mathbf{Y}$, which embeds its low-dimensional structure directly into the output layer. However, this intuitive step introduces a major analytical challenge. In particular, the sketching step renders the initial empirical kernel $\mathbf{K}_0$ rank-deficient, a critical departure from the full-rank case in Section 3. Consequently, the idealized linear dynamics are no longer a contraction on the entire space, and the convergence proof from Section 3.1 no longer applies.

Our analysis overcomes this by proving that the dynamics remain confined to an invariant, contracting subspace corresponding to the positive eigenvalues of $\mathbf{K}_0$. The success of this approach again hinges on the following unbalanced scaling, which ensure the dynamics are well-behaved:

$$\frac{c_1^3}{c_2} \geq \widetilde{\Omega}\left(\frac{\sqrt{N}d(\kappa(\mathbf{Y}) \vee \sqrt{N})}{\sqrt{r}\lambda\lambda_r^2(\mathbf{Y})\|\mathbf{Y}\|}\right), \ \frac{c_1}{c_2} \geq \widetilde{\Omega}\left(\frac{\sqrt{d}}{\sqrt{r}\|\mathbf{Y}\|}\right), \ c_1 c_2 \geq \widetilde{\Omega}\left(\frac{\sqrt{Nr}\kappa(\mathbf{Y})}{\lambda\lambda_r(\mathbf{Y})}\right). \tag{9}$$

**Theorem 4.1** (Low-rank vector responses). *Consider a two-layer ReLU network with sketching initialization* (8) *satisfying the scaling conditions in* (9). *If the network width is near-linear in both the sample size $N$ and the rank $r$,*

$$m \geq \max\left\{\Omega\left(\frac{Nr\kappa^2(\mathbf{Y})}{\lambda} \cdot \mathrm{poly}\left(\log\frac{Nrd\kappa\kappa(\mathbf{Y})}{\delta}\right)\right), \Omega\left(\frac{1}{\delta}\right)\right\},$$

*then for any $\delta \in (0,1)$, NAG finds a solution satisfying $\mathcal{L}(\mathbf{V}_T, \mathbf{W}_T) \leq \epsilon \cdot \mathcal{L}(\mathbf{V}_0, \mathbf{W}_0)$ in*

$$T = O\left(\sqrt{r}\kappa(\mathbf{Y}) \cdot \log(1/\epsilon)\right)$$

*iterations with probability at least $1 - \delta$, where $r = \mathrm{rank}(\mathbf{Y})$ and $\kappa$ is the NTK condition number.*

Theorem 4.1 shows that sketching enables low-rank adaptivity in both width requirement and convergence rate, with an added dependency on the condition number of the response matrix. When $\kappa(\mathbf{Y}) = O(1)$, the result matches Theorem 3.2 with $n$ replaced by $r$. This can happen, for instance, if $r \ll \min(N, n)$ and the observed responses are $r$-dimensional Gaussian responses randomly embedded into $n$-dimensional Euclidean space (Vershynin, 2010).

### 4.1 PROOF OVERVIEW FOR LOW-RANK RESPONSES

As established, the central challenge is the rank deficiency of the initial kernel $\mathbf{K}_0$, which invalidates the global contraction argument from the general case. Our proof overcomes this by demonstrating that the residual dynamics are surprisingly confined to an invariant, contracting subspace where a linear convergence rate can be recovered. The first step in our analysis is to precisely characterize this subspace; we prove that the positive eigenspace of $\mathbf{K}_0$ aligns perfectly with the low-rank structure of the responses, showing that $\mathrm{col}(\mathbf{K}_0) = \mathbb{R}^N \otimes \mathrm{col}(\mathbf{Y})$ (Lemma 4.4). The crucial technical step is then to prove that the dynamics remain confined within this subspace throughout training. We show by induction that for all $t \geq 0$, both the residual $\mathbf{r}_t$ and the error term $\boldsymbol{\xi}_t$ lie in this invariant subspace (Lemma 4.5), a highly non-trivial result given the non-linear ReLU activation. With this subspace confinement established, we formally prove a contraction mapping for the operator $\mathbf{T}_{\mathrm{NAG}}$ when restricted to this subspace (Lemma 4.6). This re-establishes the key condition for linear convergence, and the remainder of the proof follows the inductive argument from Section 3.1: controlling the kernel shift and other error terms to derive the final accelerated rate in Theorem 4.1.

We first provide a counterpart of Lemma 3.5 for initialization (8) with a spectral lower bound.

**Lemma 4.2.** *Suppose $\mathbf{V}_0$ and $\mathbf{W}_0$ are initialized as* (8), *$\tau \geq 0$, then with probability at least* $1 - \frac{2}{m} - 2e^{-\frac{9m(r\wedge d)}{64}} - 2e^{-\min\{(m-r+1)\log\frac{1}{c\tau}, c'm\}}$,

$$\frac{1}{2}c_1\sigma_r(\mathbf{Y})\sqrt{r} \leq R_V \leq c_1\|\mathbf{Y}\|(\sqrt{r} + 2\sqrt{\log m}),$$

$$\frac{1}{2}c_2\sqrt{d} \leq R_W \leq c_2(\sqrt{d} + 2\sqrt{\log m}),$$

$$\sigma_r(\mathbf{V}_0) \geq c_1\sqrt{2}\tau\left(\sqrt{m} - \sqrt{r-1}\right)\sigma_r(\mathbf{Y}).$$

*If the failure probability is small for some $\tau > 0$ and $m \geq r$, then $\mathrm{col}(\mathbf{V}_0) = \mathrm{col}(\mathbf{Y})$.*

The proof is provided in Appendix C.3. By Lemma 4.2, $R_V$ scales with $\Theta(\sqrt{r})$ instead of $\Theta(\sqrt{n})$, and $\mathbf{V}_0$ keeps the column space of $\mathbf{Y}$ with high probability. Equipped with these results, we show the following counterpart of Lemma 3.6 with complete proof in Appendix C.4 for initialization (8).

**Lemma 4.3** (Eigenvalues of $\mathbf{K}_0$). *Suppose $\mathbf{V}_0$ and $\mathbf{W}_0$ are as (8), $\delta \in (0, 1)$, if the width satisfies*

$$m \geq \max \left\{ \widetilde{\Omega} \left( \frac{\|\mathbf{X}\|^2 r\kappa^2(\mathbf{Y})}{\lambda} \log \frac{Nr}{\delta} \right), \Omega \left( \frac{1}{\delta} \right) \right\},$$

*then $\lambda_{Nr}(\mathbf{K}_0) \geq \frac{1}{2} c_1^2 m \lambda \sigma_r^2(\mathbf{Y})$ and $\lambda_1(\mathbf{K}_0) \leq 9c_1^2 mr\lambda_1(\overline{\mathbf{K}}) \|\mathbf{Y}\|^2$ with probability at least $1 - \delta$.*

By Lemma 4.3, $\mathrm{rank}(\mathbf{K}_0) \geq Nr$ and $\kappa(\mathbf{K}_0) = O(r\kappa\kappa^2(\mathbf{Y}))$ with high probability. We characterize the positive eigensubspace of $\mathbf{K}_0$ and show its connection to the column space of $\mathbf{Y}$.

**Lemma 4.4** (Eigensubspace of $\mathbf{K}_0$). *Assume the results in Lemma 4.3 holds, then*

$$\ker(\mathbf{K}_0) \perp \mathrm{row}(\mathbf{K}_0) = \mathrm{col}(\mathbf{K}_0) = \mathbb{R}^N \otimes \mathrm{col}(\mathbf{Y}).$$

We further show that $\mathbf{r}_t$ and $\boldsymbol{\xi}_t$ are in the positive eigensubspace of $\mathbf{K}_0$ for all $t \geq 0$.

**Lemma 4.5.** *Assume the results in Lemma 4.3 holds, then $\{\mathbf{r}_t, \boldsymbol{\xi}_t\}_{t \geq 0} \subseteq \mathbb{R}^N \otimes \mathrm{col}(\mathbf{Y})$.*

The proofs of Lemmas 4.4 and 4.5 are provided in Appendices F.1 and F.2. With these two lemmas, we can show that residual and error are contracted by the NAG dynamics in the low-rank setting.

**Lemma 4.6** (Low-rank contraction). *Denote $L := \lambda_1(\mathbf{K}_0)$, $\mu := \lambda_{Nr}(\mathbf{K}_0)$, $\kappa(\mathbf{K}_0) = \frac{L}{\mu}$. By setting $\beta = \frac{\sqrt{L} - \sqrt{\mu}}{\sqrt{L} + \sqrt{\mu}}$ and $\eta = \frac{1}{L}$ it holds for any $\mathbf{u}, \mathbf{v} \in \mathbb{R}^N \otimes \mathrm{col}(\mathbf{Y})$ that*

$$\left\| \mathbf{T}_{\mathrm{NAG}} \begin{pmatrix} \mathbf{u} \\ \mathbf{v} \end{pmatrix} \right\| \leq \left( 1 - \frac{1}{\sqrt{\kappa(\mathbf{K}_0)}} \right) \left\| \begin{pmatrix} \mathbf{u} \\ \mathbf{v} \end{pmatrix} \right\|$$

*In particular, $\mathbf{r}_t$ and $\boldsymbol{\xi}_t$ are contracted.*

The proof is provided in Appendix D.3. After resolving the low-rank subspace issue, the rest of the proof follows Section 3.1. By modifying the proof of Lemma 3.9 (see Appendix E.2), we show that kernel shift is properly controlled by the change in weights. Meanwhile, Lemmas 3.11 and 3.12 still hold for initialization (8). Therefore, we can show Theorem 4.1 by properly choosing $c_1$, $c_2$ and $m$ as for Theorem 3.2. The complete proof is provided in Appendix G.7.

## 5 CONCLUSION AND FUTURE DIRECTIONS

We establish the first convergence guarantees for two-layer ReLU networks that simultaneously achieve near-linear width and provable acceleration, significantly narrowing the gap between theory and the long-standing conjecture that linear width suffices. Our analysis, enabled by a novel unbalanced Gaussian initialization, demonstrates that: (i) a near-linear width of $\widetilde{\Omega}(Nn/\lambda)$ is sufficient for GD to find a global minimum; (ii) NAG provably accelerates convergence to an $O(\sqrt{n\kappa})$ rate without requiring additional width; and (iii) for low-rank responses, a sketching technique adapts the width requirement to the intrinsic rank $r \ll n$. These findings substantially advance the state of the art for training non-smooth neural networks in the challenging NTK regime.

Our work lays a rigorous foundation for some critical avenues of future research. One is to extend our optimization analysis to the study of generalization. While our focus is on optimization, the ability to guarantee convergence under mild overparameterization provides a crucial stepping stone. Understanding how the choice of unbalanced scaling impacts the learned features and subsequent generalization is an important open question that our framework is well-suited to explore.

Furthermore, our analysis considers full-batch gradient methods. Extending these near-linear width guarantees to the stochastic setting (SGD) is a highly challenging but vital next step for aligning theory with modern practice. Similarly, adapting our techniques from shallow networks under square loss to deeper architectures and other loss functions, such as cross-entropy, would broaden the applicability of these results. We believe the tight control over kernel dynamics demonstrated in our work provides a robust analytical toolkit for tackling these future challenges.

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

# A NUMERICAL EXPERIMENTS

This section provides supporting numerical experiments. All experiments are conducted on an Intel Core i7-13700F CPU and an NVIDIA GeForce RTX 4060 Ti GPU.

Firstly, we compare the convergence rates of GD and NAG. We set input dimension $d = 20$, output dimension $n = 10$, sample size $N = 100$, and width $2m = 2000$. We generate input data $\{\mathbf{x}_i\}_{i=1}^{N}$ and $\{\mathbf{y}_i\}_{i=1}^{N}$ independently from uniform distribution on shpere $\mathbb{S}^{d-1}$. We apply initialization (4) with $c_1 = 10$ and $c_2 = 0.1$. We choose step size $\eta = \frac{1}{L}$ for GD and NAG, and momentum parameter $\beta = \frac{\sqrt{L}-\sqrt{\mu}}{\sqrt{L}+\sqrt{\mu}}$, according to Theorems 3.1 and 3.2. We run 5 trials independently and plot the GD and NAG loss curves in Figure 1. As illustrated, NAG achieves an accelerated convergence rate.

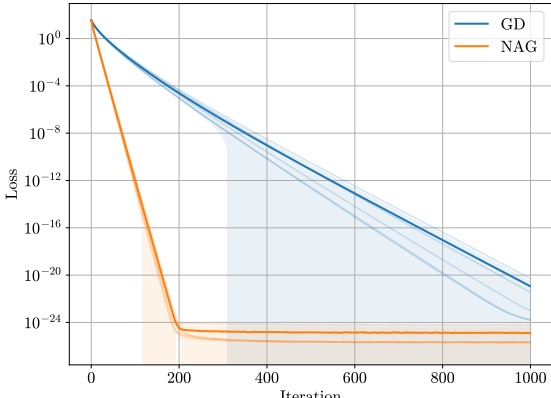

Figure 1: Convergence rates of GD and NAG. Solid curves represent average loss values across five independent runs, with semi-transparent lines showing individual runs. Shaded regions denote $\pm 1$ standard deviation from the mean. NAG converges much faster than GD.

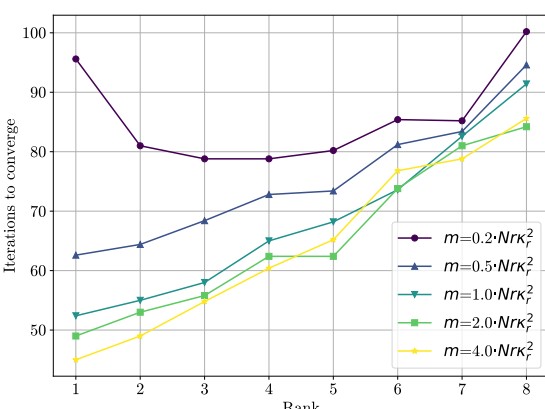

Figure 2: Iteration complexity vs rank of response matrix. When width is of $\widetilde{\Omega}(Nr\kappa_r^2)$ as required in Theorem 4.1, the convergence rates exhibit similar behaviors and grow with the rank as predicted (green and yellow lines). Smaller widths are out of the scope of our theory and exhibit different convergence behaviors (blue and purple lines).

Secondly, we demonstrate the low-rank adaptivity when applying initialization (8). We keep $d$, $N$, $c_1$ and $c_2$ as before, while changing the output dimension to $n = 50$. We sample input data from

the uniform distribution on unit sphere $\mathbb{S}^{d-1}$. For low-rank responses, we first generate standard Gaussian random matrices $\widetilde{\mathbf{Y}}_1 \in \mathbb{R}^{n \times r}$ and $\widetilde{\mathbf{Y}}_2 \in \mathbb{R}^{N \times r}$, then form $\widetilde{\mathbf{Y}} = \widetilde{\mathbf{Y}}_1 \widetilde{\mathbf{Y}}_2^\top$, and finally normalize each column of $\widetilde{\mathbf{Y}}$ to get $\mathbf{Y}$ so that $\|\mathbf{y}_i\| = 1, \forall i \in [N]$. The rank $r$ is chosen from 1 to 8. We select $m = \alpha \cdot Nr\kappa_r^2$, where $\kappa_r := \kappa(\mathbf{Y})$ denotes the condition number of the response matrix and $\alpha$ is a scaling factor chosen from $\{2^{-2}, 2^{-1}, 2^0, 2^1, 2^2\}$. We apply NAG with $\eta$ and $\beta$ specified in Theorem 4.1. We run 5 trials independently and plot the average number of iterations to make the loss smaller than tolerance $10^{-6}$ for each $\alpha$ and $r$ in Figure 2. As depicted, when $\alpha$ is large, the iteration complexity grows with the rank, and increasing its value further would not change the iteration complexity by much, indicating the width requirement in Theorem 4.1 is fulfilled. The width for $\alpha = 1$ ranges from 200 to 7000 depending on the rank, which is much smaller than the prediction of Theorem 3.2 yielding $2Nn = 10000$, demonstrating low-rank adaptivity. When $\alpha$ is small, the width may fail to meet the requirement in Theorem 4.1, resulting in slower convergence which is out of the scope of our theory.

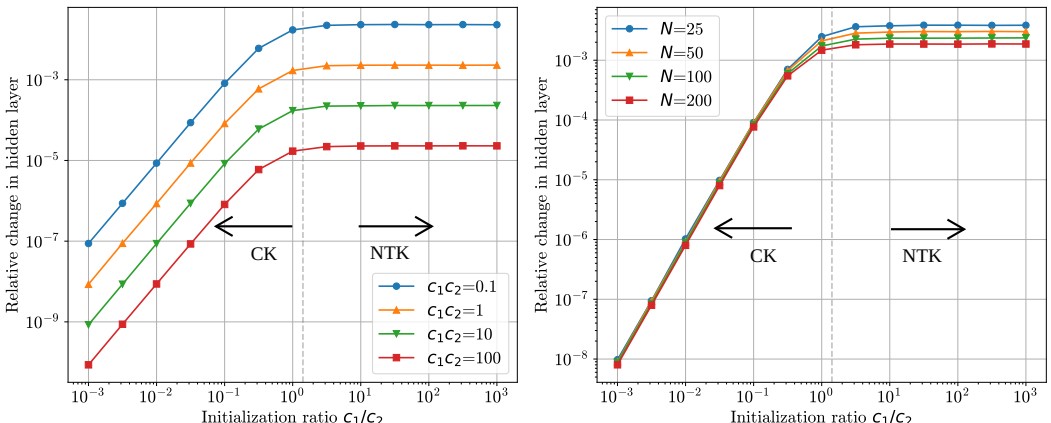

Figure 3: Hidden layer weight change (normalized by initial weight) vs initialization ratio after 100 NAG iterations. If the hidden layer is initialized to be larger than the output layer (towards the left in plots), then the weight change is small and the training falls into the CK (or random feature) regime. If the output layer is initialized to be larger than the hidden layer (towards the right in plots), then the weight change is relatively large and the training falls into the NTK regime. In the left plot, we fix $N = 100$. In the right plot, we fix $c_1 c_2 = 1$.

Thirdly, we illustrate that different unbalancedness leads to different training regimes, namely the CK (random feature) regime and the NTK regime. For this part, we set $d = 20$, $n = 10$, $m = Nn$, and apply NAG with initialization (4) for $T = 100$ iterations. To illustrate smoother transitions between the two kernel regimes, we set $\eta = \frac{1}{L'}$ and $\beta = \frac{\sqrt{L'} - \sqrt{\mu'}}{\sqrt{L'} + \sqrt{\mu'}}$ where $L' = \lambda_1(\mathbf{H}_0)$, $\mu' = \lambda_{Nn}(\mathbf{H}_0)$, and $\mathbf{H}_0 = \mathbf{K}_0 + \mathbf{G}_0$ is the total kernel matrix at initialization. The input data and responses are again sampled uniformly from $\mathbb{S}^{d-1}$. We plot the relative weight change in the hidden layer $\frac{\|\mathbf{W}_T - \mathbf{W}_0\|_F}{\|\mathbf{W}_0\|_F}$ with respect to the varying initialization scaling ratio $c_1/c_2$ in Figure 3, and plot the training loss $\mathcal{L}(\mathbf{V}_T, \mathbf{W}_T)$ along with the condition number $L'/\mu'$ of $\mathbf{H}_0$ in Figure 4. As illustrated, when the ratio $c_1/c_2$ is small, the hidden layer is larger at initialization and CK dominates the dynamics, leading to smaller weight change and slower convergence. When the ratio is large, the output layer is larger and NTK dominates the dynamics, leading to larger weight change and faster convergence. We (also Song et al. (2021); Bombari et al. (2022)) consider the latter scaling, while Nguyen (2021) considers the former one.

## B PROBABILITY TOOLS

### B.1 CONCENTRATION INEQUALITIES

This section provides some basic concentration inequalities for random variables.

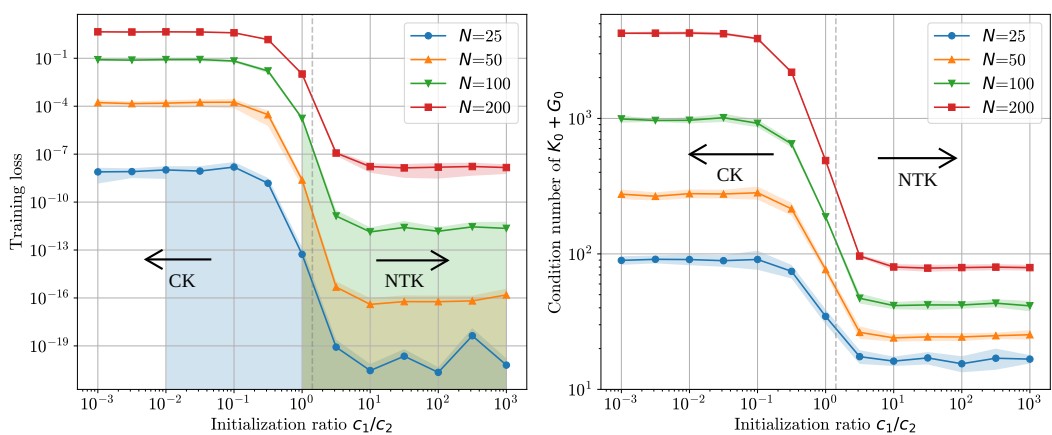

Figure 4: Training loss after 100 NAG iterations and condition number of empirical kernel matrix vs initialization ratio. We fix $c_1 c_2 = 1$. The left plot shows that the convergence of training loss in the CK regime (hidden layer larger, to left) is slower than in the NTK regime (output layer larger, to right). This phenomenon is captured by the difference between condition numbers of the initial kernel matrix $\mathbf{H}_0 = \mathbf{K}_0 + \mathbf{G}_0$ in the two regimes, as shown in the right plot.

**Proposition B.1** (Lemma 1 in Laurent & Massart (2000)). *Let $W \sim \chi^2(r)$ be the squared Euclidean norm of a standard Gaussian random vector in $\mathbb{R}^r$. Then for any $s > 0$,*

$$\mathbb{P}\left(W \geq r + 2\sqrt{rs} + 2s\right) \leq e^{-s},$$
$$\mathbb{P}\left(W \leq r - 2\sqrt{rs}\right) \leq e^{-s}.$$

**Lemma B.2** (Bernstein's inequality, Theorem 2.8.1 in Vershynin (2018)). *Let $X_1, \ldots, X_m$ be independent zero-mean sub-exponential random variables. Then for every $t \geq 0$,*

$$\mathbb{P}\left(\left|\sum_{i=1}^m X_i\right| \geq t\right) \leq 2\exp\left(-c\min\left\{\frac{t^2}{\sum_{i=1}^m \|X_i\|_{\psi_1}^2}, \frac{t}{\max_{1 \leq i \leq m} \|X_i\|_{\psi_1}}\right\}\right),$$

*where $c > 0$ is an absolute constant.*

**Lemma B.3** (Weighted Chernoff, Theorem 1 in Raghavan (1988)). *Suppose $\max_{1 \leq i \leq m} a_i > 0$, $X_1, X_2, \ldots, X_m$ are independent Bernoulli random variables with $\mathbb{E}[X_i] = p_i$, then for $\delta > 0$,*

$$\mathbb{P}\left(\sum_{i=1}^m a_i X_i > (1+\delta)\sum_{i=1}^m a_i p_i\right) \leq \left[\frac{e^\delta}{(1+\delta)^{(1+\delta)}}\right]^{\left(\sum_{i=1}^m a_i p_i\right)/\left(\max_{1 \leq i \leq m} a_i\right)}.$$

### B.2 SINGULAR VALUE OF RANDOM MATRIX

This section provides a tail bound for the smallest singular value of sub-Gaussian random matrices.

**Proposition B.4** (Rudelson & Vershynin (2009)). *Let $\mathbf{A}$ be an $N \times n$ random matrix, $N \geq n$, whose elements are i.i.d. zero mean sub-Gaussian random variables with unit variance. Then for $\tau \geq 0$, we have*

$$\mathbb{P}\left(\sigma_n(\mathbf{A}) \leq \tau(\sqrt{N} - \sqrt{n-1})\right) \leq (c_1\tau)^{N-n+1} + e^{-c_2 N}$$

*where $c_1, c_2 > 0$ depend (polynomially) only on the sub-Gaussian moment.*

### B.3 MATRIX CONCENTRATION

This section provides concentration inequalities for eigenvalues of the sum of random matrices.

**Proposition B.5** (Matrix Chernoff, Theorem 4.1 in Gittens & Tropp (2011)). *Suppose* $\{\mathbf{Z}_i\}_{i=1}^m$ *is a set of independent random positive-semidefinite matrices with dimension* $n$, $\|\mathbf{Z}_i\| \leq B$. *For* $k \in [n]$, *let* $\mu_k = \lambda_k(\sum_{i=1}^m \mathbb{E}\mathbf{Z}_i)$, *then*

$$\mathbb{P}\left(\lambda_k\left(\sum_{i=1}^m \mathbf{Z}_i\right) \geq (1+\delta)\mu_k\right) \leq (n-k+1)\left[\frac{e^\delta}{(1+\delta)^{1+\delta}}\right]^{\mu_k/B}, \ \forall \delta > 0,$$

$$\mathbb{P}\left(\lambda_k\left(\sum_{i=1}^m \mathbf{Z}_i\right) \leq (1-\delta)\mu_k\right) \leq k\left[\frac{e^{-\delta}}{(1-\delta)^{1-\delta}}\right]^{\mu_k/B}, \ \forall \delta \in [0,1).$$

## C  PROOF FOR INITIALIZATION

This section provides proofs for Gaussian initialization (4) and Sketching initialization (8).

### C.1  PROOF OF LEMMA 3.5

*Proof of Lemma 3.5.* Applying Proposition B.1 with $s = 2\log m$ for the upper part and taking the union bound, we get

$$\mathbb{P}\left(\exists i \in [m], \ \|[\boldsymbol{\Phi}]_{:,i}\| > \sqrt{n} + 2\sqrt{\log m}\right) \leq \frac{1}{m},$$

$$\mathbb{P}\left(\exists i \in [m], \ \|[\boldsymbol{\Psi}]_{:,i}\| > \sqrt{d} + 2\sqrt{\log m}\right) \leq \frac{1}{m},$$

where $\boldsymbol{\Phi}$ and $\boldsymbol{\Psi}$ are defined as in (4). On the other hand, since the columns of $\boldsymbol{\Phi}$ and $\boldsymbol{\Psi}$ are independent, plugging $s = \frac{9n}{64}$ in Proposition B.1, we get

$$\mathbb{P}\left(\forall i \in [m], \ \|[\boldsymbol{\Phi}]_{:,i}\| \leq \frac{\sqrt{n}}{2}\right) \leq e^{-\frac{9nm}{64}}.$$

Similarly, we have

$$\mathbb{P}\left(\forall i \in [m], \ \|[\boldsymbol{\Psi}]_{:,i}\| \leq \frac{\sqrt{d}}{2}\right) \leq e^{-\frac{9dm}{64}}.$$

Combining the upper and lower bounds with coupled initialization and definition of $R_V$ and $R_W$, we get the final results. $\qquad\square$

### C.2  PROOF OF LEMMA 3.6

*Proof of Lemma 3.6.* In this proof, we omit the iteration index and denote $\mathbf{A}_i := \mathbf{A}_{i,0}$, $\mathbf{B}_i := \mathbf{B}_{i,0}$. Given coupled initialization (4), we have $\mathbf{K}_0 = 2\left(\sum_{i=1}^m \mathbf{B}_i^\top \otimes \mathbf{A}_i\right)$, hence it suffices to consider the sum of the first $m$ neurons and multiply the final results by a factor of 2. For any $i \in [m]$, the expectation of $\mathbf{B}_i^\top \otimes \mathbf{A}_i$ is given by

$$\begin{aligned}
\mathbb{E}[\mathbf{B}_i^\top \otimes \mathbf{A}_i] &= \mathbb{E}[\mathbf{B}_i] \otimes \mathbb{E}[\mathbf{A}_i] \\
&= \mathbb{E}_{\mathbf{w} \sim \mathcal{N}(0,\mathbf{I}_d)}\left[\mathrm{diag}(\mathbb{1}[\mathbf{w}^\top \mathbf{X} \geq 0])\mathbf{X}^\top \mathbf{X}\mathrm{diag}(\mathbb{1}[\mathbf{w}^\top \mathbf{X} \geq 0])\right] \otimes \left(c_1^2 \mathbf{I}_n\right) \\
&= c_1^2 \overline{\mathbf{K}} \otimes \mathbf{I}_n.
\end{aligned}$$

Let $\overline{L} = c_1^2 \lambda_1(\overline{\mathbf{K}})$, $\overline{\mu} = c_1^2 \lambda$, then Assumption 2.2 implies

$$\lambda_1\left(\mathbb{E}[\mathbf{B}_i^\top \otimes \mathbf{A}_i]\right) = \overline{L} \geq \lambda_{Nn}\left(\mathbb{E}[\mathbf{B}_i^\top \otimes \mathbf{A}_i]\right) = \overline{\mu} > 0.$$

We first derive a lower bound for $\lambda_{Nn}(\mathbf{K}_0)$. Let $\mathbf{Z}_i = (\mathbf{B}_i^\top \otimes \mathbf{A}_i) \cdot \mathbb{1}[\|[\mathbf{V}_0]_{:,i}\| \leq M_V]$ where $M_V > 0$ is a threshold to be specified, we have

$$\begin{aligned}
\mathbb{E}[\mathbf{Z}_i] &= \mathbb{E}\left[(\mathbf{B}_i^\top \otimes \mathbf{A}_i) \cdot \mathbb{1}[\|[\mathbf{V}_0]_{:,i}\| \leq M_V]\right] \\
&= c_1^2 \overline{\mathbf{K}} \otimes \mathbf{I}_n - \mathbb{E}\left[(\mathbf{B}_i^\top \otimes \mathbf{A}_i) \cdot \mathbb{1}[\|[\mathbf{V}_0]_{:,i}\| > M_V]\right],
\end{aligned}$$

and hence

$$\mu_{Nn} := \lambda_{Nn} \left( \sum_{i=1}^{m} \mathbb{E}[\mathbf{Z}_i] \right)$$

$$= \lambda_{Nn} \left( m \cdot \mathbb{E}[\mathbf{B}_i^\top \otimes \mathbf{A}_i] - \sum_{i=1}^{m} \mathbb{E}\left[ (\mathbf{B}_i^\top \otimes \mathbf{A}_i) \cdot \mathbb{1}[\|[\mathbf{V}_0]_{:,i}\| > M_V] \right] \right)$$

$$\geq m\bar{\mu} - \sum_{i=1}^{m} \lambda_1 \left( \mathbb{E}[\mathbf{B}_i] \otimes \mathbb{E}\left[ \mathbf{A}_i \cdot \mathbb{1}[\|[\mathbf{V}_0]_{:,i}\| > M_V] \right] \right)$$

$$= m\bar{\mu} - \sum_{i=1}^{m} \lambda_1 \left( \overline{\mathbf{K}} \right) \lambda_1 \left( \mathbb{E}\left[ \mathbf{A}_i \cdot \mathbb{1}[\|[\mathbf{V}_0]_{:,i}\| > M_V] \right] \right),$$

where the third line is from Weyl's inequality and the fourth line is the property of the eigenvalue of the Kronecker product. For any $i \in [m]$, let $\boldsymbol{\phi}_i := [\boldsymbol{\Phi}]_{:,i}$, we have $\boldsymbol{\phi}_i \sim \mathcal{N}(0, \mathbf{I}_n)$ and

$$\lambda_1 \left( \mathbb{E}\left[ \mathbf{A}_i \cdot \mathbb{1}[\|[\mathbf{V}_0]_{:,i}\| > M_V] \right] \right) \leq \mathbb{E}\left[ \lambda_1 \left( \mathbf{A}_i \cdot \mathbb{1}[\|[\mathbf{V}_0]_{:,i}\| > M_V] \right) \right]$$

$$\leq \mathbb{E}\left[ \lambda_1 \left( c_1^2 \boldsymbol{\phi}_i \boldsymbol{\phi}_i^\top \cdot \mathbb{1}\left[ c_1 \|\boldsymbol{\phi}_i\| > M_V \right] \right) \right]$$

$$\leq \mathbb{E}_{\boldsymbol{\phi} \sim \mathcal{N}(0, \mathbf{I}_n)} \left[ c_1^2 \|\boldsymbol{\phi}\|^2 \cdot \mathbb{1}\left[ c_1 \|\boldsymbol{\phi}\| > M_V \right] \right]$$

$$= c_1^2 \mathbb{E}_{W \sim \chi^2(n)} \left[ W \cdot \mathbb{1}\left[ c_1 \sqrt{W} > M_V \right] \right]$$

$$= c_1^2 \int_{\left(\frac{M_V}{c_1}\right)^2}^{\infty} \mathbb{P}\left( W > x \right) \, \mathrm{d}x$$

$$\leq c_1^2 \int_{\underline{s}}^{\infty} e^{-s} \left( 2 + \sqrt{\frac{n}{s}} \right) \, \mathrm{d}s$$

$$\leq c_1^2 e^{-\underline{s}} \left( 2 + \sqrt{\frac{n}{\underline{s}}} \right).$$

Here, the first line is by the convexity of the top eigenvalue and Jensen's inequality. The second line is by relaxing the condition within the indicator. The fifth line uses the integral identity for non-negative random variables. The sixth line uses the tail bound for $\chi^2(n)$ distribution (Proposition B.1) with change of variable $x = n + 2\sqrt{ns} + 2s$, and $M_V = c_1 \sqrt{n + 2\sqrt{n\underline{s}} + 2\underline{s}}$ for some $\underline{s} > 0$.

By choosing $\underline{s} = \max\{n, \log(6\kappa(\overline{\mathbf{K}}))\}$, we have

$$\mu_{Nn} \geq m\bar{\mu} - m\overline{L} e^{-\underline{s}} \left( 2 + \sqrt{\frac{n}{\underline{s}}} \right)$$

$$\geq mc_1^2 \lambda - 3mc_1^2 \lambda_1(\overline{\mathbf{K}})(6\kappa(\overline{\mathbf{K}}))^{-1}$$

$$\geq \frac{1}{2} m\bar{\mu},$$

and correspondingly

$$M_V = c_1 \cdot \max \left\{ \sqrt{5n}, \sqrt{n + 2\sqrt{n \log(6\kappa(\overline{\mathbf{K}}))} + 2\log(6\kappa(\overline{\mathbf{K}}))} \right\}$$

$$\leq c_1 \left( \sqrt{5n} + \sqrt{2\log(6\kappa(\overline{\mathbf{K}}))} \right).$$

Moreover, for every $i \in [m]$ we have

$$\|\mathbf{Z}_i\| = \|\mathbf{B}_i\| \|\mathbf{A}_i \cdot \mathbb{1}[\|[\mathbf{V}_0]_{:,i}\| \leq M_V]\|$$

$$\leq \|\mathrm{diag}(\mathbb{1}[\mathbf{W}_0^\top \mathbf{X} \geq 0]_{i,:}) \mathbf{X}^\top \mathbf{X} \mathrm{diag}(\mathbb{1}[\mathbf{W}_0^\top \mathbf{X} \geq 0]_{i,:})\| M_V^2$$

$$\leq \|\mathbf{X}\|^2 M_V^2.$$

By plugging $\mathbf{Z}_i$ into the matrix Chernoff bound (Proposition B.5) and setting $\delta = \frac{1}{2}$ for the smallest eigenvalue, we get

$$
\mathbb{P}\left(\lambda_{Nn}\left(\sum_{i=1}^{m}\mathbf{B}_i^\top \otimes \mathbf{A}_i\right) \leq \frac{1}{4}m\bar{\mu}\right)
$$

$$
\leq \mathbb{P}\left(\lambda_{Nn}\left(\sum_{i=1}^{m}(\mathbf{B}_i^\top \otimes \mathbf{A}_i)\cdot \mathbb{1}[\|[\mathbf{V}_0]_{:,i}\| \leq M_V]\right) \leq \frac{1}{2}\mu_{Nn}\right)
$$

$$
\leq \exp\left(-\frac{\mu_{Nn}}{10\|\mathbf{X}\|^2 M_V^2} + \log(Nn)\right)
$$

$$
\leq \exp\left(-\frac{m\lambda}{40\|\mathbf{X}\|^2 (5n + 2\log(6\kappa(\overline{\mathbf{K}})))} + \log(Nn)\right),
$$

and the bounds for $\lambda_1(\mathbf{K}_0)$ and $\lambda_{Nn}(\mathbf{K}_0)$ follow immediately. The first inequality is because $\mathbf{B}_i$ and $\mathbf{A}_i$ are positive-semidefinite, thus removing summands will only make the eigenvalue smaller. Plugging in $m = \widetilde{\Omega}\left(\frac{\|\mathbf{X}\|^2 n}{\lambda}\log\frac{Nn}{\delta_1}\right)$ we get failure probability less than $\delta_1$.

We now derive an upper bound for $\lambda_1(\mathbf{K}_0)$. Conditioned on $\mathbf{V}_0$, $\mathbf{A}_i \preceq \|[\mathbf{V}_0]_{:,i}\|^2 \mathbf{I}_n$ for all $i \in [m]$. Denote $a_i = \|[\mathbf{V}_0]_{:,i}\|^2$, then by Kronecker product and positive semidefinite summands, $\mathbf{B}_i^\top \otimes \mathbf{A}_i \preceq a_i \cdot \mathbf{B}_i \otimes \mathbf{I}_n \preceq a_i \|\mathbf{X}\|^2 \cdot \mathbf{I}_{Nn}$ and

$$
\lambda_1\left(\sum_{i=1}^{m}\mathbf{B}_i^\top \otimes \mathbf{A}_i\right) \leq \lambda_1\left(\sum_{i=1}^{m}a_i \cdot \mathbf{B}_i^\top \otimes \mathbf{I}_n\right) \leq \lambda_1\left(\sum_{i=1}^{m}a_i \cdot \mathbf{B}_i\right).
$$

Let $\mathbf{Z}_i = a_i\mathbf{B}_i$, then $\lambda_1\left(\sum_{i=1}^{m}\mathbb{E}[\mathbf{Z}_i]\right) = \left(\sum_{i=1}^{m}a_i\right)\lambda_1(\overline{\mathbf{K}})$. By Proposition B.5, and independence between $\mathbf{V}_0$ and $\mathbf{W}_0$, we have

$$
\mathbb{P}\left(\lambda_1\left(\sum_{i=1}^{m}\mathbf{Z}_i\right) \geq \frac{3}{2}\left(\sum_{i=1}^{m}a_i\right)\lambda_1(\overline{\mathbf{K}})\right) \leq \exp\left(-\frac{\left(\sum_{i=1}^{m}a_i\right)\lambda_1(\overline{\mathbf{K}})}{10\|\mathbf{X}\|^2 \max_{i\in[m]}a_i} + \log n\right).
$$

By definition, $\sum_{i=1}^{m}a_i = c_1^2\|\mathbf{\Phi}\|_{\mathrm{F}}^2 = c_1^2 W$ where $W \sim \chi^2(mn)$, $\max_{i\in[m]}a_i = R_V^2$. By setting $s = \frac{mn}{4}$ in Proposition B.1, we have $\frac{mn}{2} \leq W \leq \frac{5mn}{2} < 3mn$ with probability at least $1 - 2e^{-\frac{mn}{4}}$. On the other hand, by Lemma 3.5, $R_V^2 \leq 2c_1^2(n + 4\log m)$ with probability at least $1 - \frac{2}{m} - 2e^{-\frac{9m(n\wedge d)}{64}}$. Hence with probability at least $1 - \frac{2}{m} - 4e^{-\frac{9m(n\wedge d)}{64}}$,

$$
\mathbb{P}\left(\lambda_1\left(\sum_{i=1}^{m}\mathbf{B}_i^\top \otimes \mathbf{A}_i\right) > \frac{9}{2}c_1^2 mn\lambda_1(\overline{\mathbf{K}})\right)
$$

$$
\leq \mathbb{P}\left(\lambda_1\left(\sum_{i=1}^{m}\mathbf{Z}_i\right) \geq \frac{3}{2}\left(\sum_{i=1}^{m}a_i\right)\lambda_1(\overline{\mathbf{K}})\right)
$$

$$
\leq \exp\left(-\frac{mn\lambda_1(\overline{\mathbf{K}})}{40\|\mathbf{X}\|^2 (n + 4\log m)} + \log n\right),
$$

where the first inequality is from the upper bound of $W$, the second inequality is from the lower bound of $W$ and the upper bound of $R_V$. By setting $m = \max\left\{\widetilde{\Omega}\left(\frac{\|\mathbf{X}\|^2 n}{\lambda}\log\frac{Nn}{\delta_1}\right), \Omega\left(\frac{1}{\delta_1}\right)\right\}$, the failure probability can be bounded by $\delta_1$. $\qquad\square$

## C.3 PROOF OF LEMMA 4.2

*Proof of Lemma 4.2.* We only need to show the $R_V$ part, as $R_W$ is the same as in Lemma 3.5. Suppose $\mathbf{Y}$ has singular value decomposition $\mathbf{Y} = \mathbf{U}_Y\mathbf{\Sigma}_Y\mathbf{V}_Y^\top$, then the columns of $\mathbf{V}_Y^\top\mathbf{\Phi} \in \mathbb{R}^{r\times m}$ are independent Gaussian vectors with distribution $\mathcal{N}(0, \mathbf{V}_Y^\top\mathbf{V}_Y) = \mathcal{N}(0, \mathbf{I}_r)$.

Applying Proposition B.1 with $s = 2\log m$ for the upper part and taking the union bound, we get

$$\mathbb{P}\left(\exists i \in [m], \, \left\|[\mathbf{V}_Y^\top \boldsymbol{\Phi}]_{:,i}\right\| > \sqrt{r} + 2\sqrt{\log m}\right) \leq \frac{1}{m}.$$

On the other hand, since the columns of $\boldsymbol{\Phi}$ and $\boldsymbol{\Psi}$ are independent, plugging $s = \frac{9n}{64}$ in Proposition B.1, we get

$$\mathbb{P}\left(\forall i \in [m], \, \left\|[\mathbf{V}_Y^\top \boldsymbol{\Phi}]_{:,i}\right\| \leq \frac{\sqrt{r}}{2}\right) \leq e^{-\frac{9rm}{64}}.$$

Combining the upper and lower bounds and the singular values of $\mathbf{Y}$ yields the results for $R_V$ and $R_W$.

By Proposition B.4,

$$\mathbb{P}\left(\sigma_r(\mathbf{V}_Y^\top \boldsymbol{\Phi}) \leq \tau\left(\sqrt{m} - \sqrt{r-1}\right)\right) \leq e^{-(m-r+1)\log\frac{1}{c\tau}} + e^{-c'm}$$

for some absolute constants $c$ and $c'$ and any $\tau \geq 0$. Therefore, we have

$$\mathbb{P}\left(\tau(\sqrt{m} - \sqrt{r-1}) \leq \sigma_r(\mathbf{V}_Y^\top \boldsymbol{\Phi}) \leq \sigma_1(\mathbf{V}_Y^\top \boldsymbol{\Phi}) \leq 2\sqrt{m} + \sqrt{r}\right) \geq 1 - \delta,$$

where $\delta = 2e^{-\min\{(m-r+1)\log\frac{1}{c\tau}, c'm\}}$, and the spectral bound follow immediately from the fact that

$$c_1\sqrt{2} \cdot \sigma_r(\mathbf{V}_Y^\top \boldsymbol{\Phi})\sigma_r(\mathbf{Y}) \leq \sigma_r(\mathbf{V}_0),$$

where the factor $\sqrt{2}$ is due to the coupled initialization. $\qquad\square$

### C.4 PROOF OF LEMMA 4.3

*Proof of Lemma 4.3.* Similar to the proof of Lemma 3.6, we omit the iteration index and denote $\mathbf{A}_i := \mathbf{A}_{i,0}$, $\mathbf{B}_i := \mathbf{B}_{i,0}$, and consider the sum of the first $m$ neurons and multiply the final results by a factor of 2. For any $i \in [m]$, the expectation of $\mathbf{B}_i^\top \otimes \mathbf{A}_i$ is given by

$$\begin{aligned}
\mathbb{E}[\mathbf{B}_i^\top \otimes \mathbf{A}_i] &= \mathbb{E}[\mathbf{B}_i] \otimes \mathbb{E}[\mathbf{A}_i] \\
&= \mathbb{E}_{\mathbf{w}\sim\mathcal{N}(0,\mathbf{I})}\left[\mathrm{diag}(\mathbb{1}[\mathbf{w}^\top\mathbf{X} \geq 0])\mathbf{X}^\top\mathbf{X}\mathrm{diag}(\mathbb{1}[\mathbf{w}^\top\mathbf{X} \geq 0])\right] \otimes \left(c_1^2\mathbf{Y}\mathbf{Y}^\top\right) \\
&= \overline{\mathbf{K}} \otimes \left(c_1^2\mathbf{Y}\mathbf{Y}^\top\right),
\end{aligned}$$

where we use $\mathbb{E}[\mathbf{A}_i] = \mathbb{E}[[\mathbf{V}_0]_{:,i}[\mathbf{V}_0]_{:,i}^\top] = c_1^2\mathbb{E}_{\boldsymbol{\phi}_i\sim\mathcal{N}(0,\mathbf{I}_N)}[\mathbf{Y}\boldsymbol{\phi}_i\boldsymbol{\phi}_i^\top\mathbf{Y}^\top] = c_1^2\mathbf{Y}\mathbf{I}_N\mathbf{Y}^\top = c_1^2\mathbf{Y}\mathbf{Y}^\top$.

Let $\overline{L} = c_1^2\lambda_1(\overline{\mathbf{K}})\lambda_1(\mathbf{Y}\mathbf{Y}^\top)$, $\overline{\mu} = c_1^2\lambda\lambda_r(\mathbf{Y}\mathbf{Y}^\top)$, then Assumption 2.2 and $\mathrm{rank}(\mathbf{Y}) = r$ implies

$$\lambda_1\left(\mathbb{E}[\mathbf{B}_i^\top \otimes \mathbf{A}_i]\right) = \overline{L} \geq \lambda_{Nr}\left(\mathbb{E}[\mathbf{B}_i^\top \otimes \mathbf{A}_i]\right) = \overline{\mu} > 0 = \lambda_{Nr+1}\left(\mathbb{E}[\mathbf{B}_i^\top \otimes \mathbf{A}_i]\right).$$

We first derive a lower bound for $\lambda_{Nr}(\mathbf{K}_0)$. Let $\mathbf{Z}_i = (\mathbf{B}_i^\top \otimes \mathbf{A}_i) \cdot \mathbb{1}[\|[\mathbf{V}_0]_{:,i}\| \leq M_V]$ where $M_V > 0$ is a threshold to be specified, we have

$$\begin{aligned}
\mathbb{E}[\mathbf{Z}_i] &= \mathbb{E}\left[(\mathbf{B}_i^\top \otimes \mathbf{A}_i) \cdot \mathbb{1}[\|[\mathbf{V}_0]_{:,i}\| \leq M_V]\right] \\
&= \overline{\mathbf{K}} \otimes \left(c_1^2\mathbf{Y}\mathbf{Y}^\top\right) - \mathbb{E}\left[(\mathbf{B}_i^\top \otimes \mathbf{A}_i) \cdot \mathbb{1}[\|[\mathbf{V}_0]_{:,i}\| > M_V]\right],
\end{aligned}$$

and hence

$$\begin{aligned}
\mu_{Nn} &:= \lambda_{Nn}\left(\sum_{i=1}^m \mathbb{E}[\mathbf{Z}_i]\right) \\
&= \lambda_{Nn}\left(m \cdot \mathbb{E}[\mathbf{B}_i^\top \otimes \mathbf{A}_i] - \sum_{i=1}^m \mathbb{E}\left[(\mathbf{B}_i^\top \otimes \mathbf{A}_i) \cdot \mathbb{1}[\|[\mathbf{V}_0]_{:,i}\| > M_V]\right]\right) \\
&\geq m\overline{\mu} - \sum_{i=1}^m \lambda_1\left(\mathbb{E}[\mathbf{B}_i] \otimes \mathbb{E}\left[\mathbf{A}_i \cdot \mathbb{1}[\|[\mathbf{V}_0]_{:,i}\| > M_V]\right]\right) \\
&= m\overline{\mu} - \sum_{i=1}^m \lambda_1\left(\overline{\mathbf{K}}\right)\lambda_1\left(\mathbb{E}\left[\mathbf{A}_i \cdot \mathbb{1}[\|[\mathbf{V}_0]_{:,i}\| > M_V]\right]\right),
\end{aligned}$$

where the third line is from Weyl's inequality and the fourth line is the property of the eigenvalue of the Kronecker product. Suppose $\mathbf{Y}$ has SVD $\mathbf{Y} = \mathbf{U}_Y \mathbf{\Sigma}_Y \mathbf{V}_Y^\top$. For any $i \in [m]$, we have $\mathbf{V}_Y^\top \boldsymbol{\phi}_i \sim \mathcal{N}(0, \mathbf{I}_r)$, thus

$$
\begin{aligned}
\lambda_1 \left( \mathbb{E} \left[ \mathbf{A}_i \cdot \mathbb{1}[\|[\mathbf{V}_0]_{:,i}\| > M_V] \right] \right) &\leq \mathbb{E} \left[ \lambda_1 \left( \mathbf{A}_i \cdot \mathbb{1}[\|[\mathbf{V}_0]_{:,i}\| > M_V] \right) \right] \\
&\leq \mathbb{E} \left[ \lambda_1 \left( c_1^2 \mathbf{U}_Y \mathbf{\Sigma}_Y \mathbf{V}_Y^\top \boldsymbol{\phi}_i \boldsymbol{\phi}_i^\top \mathbf{V}_Y \mathbf{\Sigma}_Y \mathbf{U}_Y^\top \cdot \mathbb{1} \left[ c_1 \|\mathbf{Y}\| \|\mathbf{V}_Y^\top \boldsymbol{\phi}_i\| > M_V \right] \right) \right] \\
&\leq \mathbb{E}_{\boldsymbol{\psi} \sim \mathcal{N}(0, \mathbf{I}_r)} \left[ c_1^2 \|\mathbf{Y}\|^2 \|\boldsymbol{\psi}\|^2 \cdot \mathbb{1} \left[ c_1 \|\mathbf{Y}\| \|\boldsymbol{\psi}\| > M_V \right] \right] \\
&= c_1^2 \|\mathbf{Y}\|^2 \mathbb{E}_{W \sim \chi^2(r)} \left[ W \cdot \mathbb{1} \left[ c_1 \|\mathbf{Y}\| \sqrt{W} > M_V \right] \right] \\
&= c_1^2 \|\mathbf{Y}\|^2 \int_{\left( \frac{M_V}{c_1 \|\mathbf{Y}\|} \right)^2}^\infty \mathbb{P} \left( W > x \right) \, \mathrm{d}x \\
&\leq c_1^2 \|\mathbf{Y}\|^2 \int_{\underline{s}}^\infty e^{-s} \left( 2 + \sqrt{\frac{r}{s}} \right) \, \mathrm{d}s \\
&\leq c_1^2 \|\mathbf{Y}\|^2 e^{-\underline{s}} \left( 2 + \sqrt{\frac{r}{\underline{s}}} \right).
\end{aligned}
$$

Here, the first line is by the convexity of the top eigenvalue and Jensen's inequality. The second line is by relaxing the condition within the indicator. The third line substitutes $\boldsymbol{\psi} = \mathbf{V}_Y^\top \boldsymbol{\phi}_i$, and the fourth line substitutes $W = \|\boldsymbol{\psi}\|^2$. The fifth line uses the integral identity for non-negative random variables. The sixth line uses the tail bound for $\chi^2(r)$ distribution (Proposition B.1) with change of variable $x = r + 2\sqrt{rs} + 2s$, and $M_V = c_1 \|\mathbf{Y}\| \sqrt{r + 2\sqrt{r\underline{s}} + 2\underline{s}}$ for some $\underline{s} > 0$.

By choosing $\underline{s} = \max\{r, \log(6\kappa(\overline{\mathbf{K}})\kappa(\mathbf{YY}^\top))\}$, we have

$$
\begin{aligned}
\mu_{Nr} &\geq m\bar{\mu} - m\bar{L}e^{-\underline{s}} \left( 2 + \sqrt{\frac{r}{\underline{s}}} \right) \\
&\geq mc_1^2 \lambda \lambda_r(\mathbf{YY}^\top) - 3mc_1^2 \lambda_1(\overline{\mathbf{K}})\lambda_1(\mathbf{YY}^\top)(6\kappa(\overline{\mathbf{K}})\kappa(\mathbf{YY}^\top))^{-1} \\
&\geq \frac{1}{2}m\bar{\mu},
\end{aligned}
$$

and correspondingly

$$
\begin{aligned}
M_V &= c_1 \|\mathbf{Y}\| \cdot \max \left\{ \sqrt{5r}, \sqrt{r + 2\sqrt{r\log(6\kappa(\overline{\mathbf{K}})\kappa(\mathbf{YY}^\top))} + 2\log(6\kappa(\overline{\mathbf{K}})\kappa(\mathbf{YY}^\top))} \right\} \\
&\leq c_1 \|\mathbf{Y}\| \left( \sqrt{5r} + \sqrt{2\log(6\kappa(\overline{\mathbf{K}})\kappa(\mathbf{YY}^\top))} \right).
\end{aligned}
$$

Moreover, for every $i \in [m]$ we have

$$
\|\mathbf{Z}_i\| = \|\mathbf{B}_i\| \|\mathbf{A}_i \cdot \mathbb{1}[\|[\mathbf{V}_0]_{:,i}\| \leq M_V]\| \leq \|\mathbf{X}\|^2 M_V^2.
$$

By plugging $\mathbf{Z}_i$ into the matrix Chernoff bound (Proposition B.5) and setting $\delta = \frac{1}{2}$ for the smallest eigenvalue, we get

$$
\begin{aligned}
&\mathbb{P} \left( \lambda_{Nr} \left( \sum_{i=1}^m \mathbf{B}_i^\top \otimes \mathbf{A}_i \right) \leq \frac{1}{4}m\bar{\mu} \right) \\
&\leq \mathbb{P} \left( \lambda_{Nr} \left( \sum_{i=1}^m (\mathbf{B}_i^\top \otimes \mathbf{A}_i) \cdot \mathbb{1}[\|[\mathbf{V}_0]_{:,i}\| \leq M_V] \right) \leq \frac{1}{2}\mu_{Nr} \right) \\
&\leq \exp \left( -\frac{\mu_{Nr}}{10 \|\mathbf{X}\|^2 M_V^2} + \log(Nr) \right) \\
&\leq \exp \left( -\frac{m\lambda\lambda_r(\mathbf{YY}^\top)}{40 \|\mathbf{X}\|^2 \|\mathbf{Y}\|^2 (5r + 2\log(6\kappa(\overline{\mathbf{K}})\kappa(\mathbf{YY}^\top)))} + \log(Nr) \right) \\
&\leq \exp \left( -\frac{m\lambda}{40 \|\mathbf{X}\|^2 \kappa(\mathbf{YY}^\top)(5r + 2\log(6\kappa(\overline{\mathbf{K}})\kappa(\mathbf{YY}^\top)))} + \log(Nr) \right),
\end{aligned}
$$

and the bounds for $\lambda_1(\mathbf{K}_0)$ and $\lambda_{Nr}(\mathbf{K}_0)$ follow immediately. The first inequality is because $\mathbf{B}_i$ and $\mathbf{A}_i$ are positive-semidefinite, thus removing summands will only make the eigenvalue smaller. Plugging in $m = \Omega\left(\frac{\|\mathbf{X}\|^2 r \kappa(\mathbf{Y}\mathbf{Y}^\top) \log(\kappa(\overline{\mathbf{K}})\kappa(\mathbf{Y}\mathbf{Y}^\top))}{\lambda} \log \frac{Nr}{\delta_1}\right)$ we get failure probability less than $\delta_1$.

We now derive an upper bound for $\lambda_1(\mathbf{K}_0)$. Similar to the proof of Lemma 3.6 in Appendix C.2, we denote $a_i = \|[\mathbf{V}_0]_{:,i}\|^2$. Let $\mathbf{Z}_i = a_i \mathbf{B}_i$, then conditioned on $\mathbf{V}_0$, by Proposition B.5 we have

$$\mathbb{P}\left(\lambda_1\left(\sum_{i=1}^m \mathbf{Z}_i\right) \geq \frac{3}{2}\left(\sum_{i=1}^m a_i\right) \lambda_1(\overline{\mathbf{K}})\right) \leq \exp\left(-\frac{(\sum_{i=1}^m a_i)\lambda_1(\overline{\mathbf{K}})}{10\|\mathbf{X}\|^2 \max_{i \in [m]} a_i} + \log n\right).$$

By definition, $\sum_{i=1}^m a_i = c_1^2 \left\|\mathbf{U}_Y \mathbf{\Sigma}_Y \mathbf{V}_Y^\top \mathbf{\Phi}\right\|_{\mathrm{F}}^2$. Let $W \sim \chi^2(mr)$, then by independence of entries of $\mathbf{V}_Y^\top \mathbf{\Phi} \in \mathbb{R}^{r \times m}$ due to orthogonality, we have

$$c_1^2 \sigma_r^2(\mathbf{Y}) W \leq \sum_{i=1}^m a_i \leq c_1^2 \|\mathbf{Y}\|^2 W.$$

By setting $s = \frac{mr}{4}$ in Proposition B.1, we have $\frac{mr}{2} \leq W \leq \frac{5mr}{2} < 3mr$ with probability at least $1 - 2e^{-\frac{mr}{4}}$. On the other hand, By Lemma 4.2, $R_V^2 \leq 2c_1^2 \|\mathbf{Y}\|^2 (r + 4\log m)$ with probability at least $1 - \frac{2}{m} - 2e^{-\frac{9m(r \wedge d)}{64}}$. Hence with probability at least $1 - \frac{2}{m} - 4e^{-\frac{9m(r \wedge d)}{64}}$,

$$\mathbb{P}\left(\lambda_1\left(\sum_{i=1}^m \mathbf{B}_i^\top \otimes \mathbf{A}_i\right) > \frac{9}{2} c_1^2 \|\mathbf{Y}\|^2 mr\lambda_1(\overline{\mathbf{K}})\right)$$

$$\leq \mathbb{P}\left(\lambda_1\left(\sum_{i=1}^m \mathbf{Z}_i\right) \geq \frac{3}{2}\left(\sum_{i=1}^m a_i\right) \lambda_1(\overline{\mathbf{K}})\right)$$

$$\leq \exp\left(-\frac{mr\lambda_1(\overline{\mathbf{K}})}{40\|\mathbf{X}\|^2 (r + 4\log m)\kappa^2(\mathbf{Y})} + \log n\right),$$

where the first inequality is from the upper bound of $W$, the second inequality is from the lower bound of $W$ and the upper bound of $R_V$. By setting $m = \max\left\{\widetilde{\Omega}\left(\frac{Nr\kappa^2(\mathbf{Y})}{\lambda}\log\frac{\|\mathbf{X}\|^2 r}{\delta_1}\right), \Omega\left(\frac{1}{\delta_1}\right)\right\}$, the above probability is less than $\delta_1$ and we get the final result. $\qquad\square$

# D    PROOF FOR DYNAMICS

This section provides detailed proof of the residual dynamics in Proposition 3.7 and Lemmas 3.8 and 4.6. We define several additional notations. Let $\mathbf{H}_t = \mathbf{G}_t + \mathbf{K}_t$ where $\mathbf{G}_t$ and $\mathbf{K}_t$ are defined in Sec.2.2. Let $\mathbf{J}_t = \mathbb{1}[\mathbf{W}_t^\top \mathbf{X} \geq 0]$ be the activation pattern at the $t$-th iteration, $\mathbf{M}$ be the mask of the neuron-data pairs that do not change their activation pattern during training, i.e., $\mathbf{M}_{i,j} = \mathbb{1}[[\mathbf{J}_t]_{i,j} \equiv [\mathbf{J}_0]_{i,j}, \forall t \geq 0]$, and $\mathbf{M}^\perp = \mathbf{1}_n \mathbf{1}_N^\top - \mathbf{M}$ be its negation. For $j \in [N]$, $S_j = \{i \in [2m] \mid \mathbf{M}_{i,j} = 1\}$, and let $S_j^\perp = [2m] \setminus S_j$. In Proposition 3.7, we have

$$\boldsymbol{\xi}_t = \boldsymbol{\iota}_t + \boldsymbol{\zeta}_t, \tag{10}$$

$$\boldsymbol{\iota}_t = (1+\beta)\eta(\mathbf{K}_0 - \mathbf{H}_t)\mathbf{r}_t - \beta\eta(\mathbf{K}_0 - \mathbf{H}_{t-1})\mathbf{r}_{t-1}, \tag{11}$$

$$\boldsymbol{\zeta}_t = \mathrm{vec}\Big(\beta\eta\mathbf{R}_{t-1}\sigma(\mathbf{X}^\top \mathbf{W}_{t-1})(\sigma(\mathbf{W}_t^\top \mathbf{X}) - \sigma(\mathbf{W}_{t-1}^\top \mathbf{X}))$$

$$+ \beta\eta\mathbf{P}_{t-1}\left((((\mathbf{V}_{t-1}^\top \mathbf{R}_{t-1}) \odot \mathbf{J}_{t-1})\mathbf{X}^\top \mathbf{X}) \odot \mathbf{J}_{t-1} \odot \mathbf{M}\right)$$

$$+ \beta\mathbf{P}_{t-1}\left((\mathbf{Q}_{t-1}^\top \mathbf{X}) \odot \mathbf{J}_0 \odot \mathbf{M}\right)$$

$$+ \mathbf{P}_t\left(\sigma(\mathbf{W}_{t+1}^\top \mathbf{X}) - \sigma(\mathbf{W}_t^\top \mathbf{X})\right)$$

$$+ \mathbf{V}_t\left(((\mathbf{W}_{t+1}^\top \mathbf{X}) \odot \mathbf{J}_{t+1} - (\mathbf{W}_t^\top \mathbf{X}) \odot \mathbf{J}_t) \odot \mathbf{M}^\perp\right)$$

$$- \beta\mathbf{V}_{t-1}\left(((\mathbf{W}_t^\top \mathbf{X}) \odot \mathbf{J}_t - (\mathbf{W}_{t-1}^\top \mathbf{X}) \odot \mathbf{J}_{t-1}) \odot \mathbf{M}^\perp\right)$$

$$+ (1+\beta)\eta\boldsymbol{\varphi}_t - \beta\eta\boldsymbol{\varphi}_{t-1}\Big), \tag{12}$$

$$[\boldsymbol{\varphi}_t]_{:,j} = \sum_{i \in S_j^\perp} [\mathbf{A}_{i,t}\mathbf{R}_t\mathbf{B}_{i,t}]_{:,j}, \forall j \in [N]. \tag{13}$$

### D.1 PROOF OF PROPOSITION 3.7

*Proof of Proposition 3.7.* In this proof we use blue color to mark the terms that go into the linear part or $\iota_t$ of the dynamics, while we use red color to mark the terms that go into $\zeta_t$. The NAG update is written as

$$\begin{pmatrix} \mathbf{V}_{t+1} \\ \mathbf{W}_{t+1} \end{pmatrix} = \begin{pmatrix} (1+\beta)(\mathbf{V}_t - \eta\mathbf{R}_t\sigma(\mathbf{X}^\top\mathbf{W}_t)) - \beta(\mathbf{V}_{t-1} - \eta\mathbf{R}_{t-1}\sigma(\mathbf{X}^\top\mathbf{W}_{t-1})) \\ (1+\beta)(\mathbf{W}_t - \eta\mathbf{X}((\mathbf{R}_t^\top\mathbf{V}_t) \odot \mathbb{1}[\mathbf{X}^\top\mathbf{W}_t \geq 0])) - \beta(\mathbf{W}_{t-1} - \eta\mathbf{X}((\mathbf{R}_{t-1}^\top\mathbf{V}_{t-1}) \odot \mathbb{1}[\mathbf{X}^\top\mathbf{W}_{t-1} \geq 0])) \end{pmatrix}.$$

and we have

$$\mathbf{P}_t = \beta\mathbf{P}_{t-1} - (1+\beta)\eta\mathbf{R}_t\sigma(\mathbf{X}^\top\mathbf{W}_t) + \beta\eta\mathbf{R}_{t-1}\sigma(\mathbf{X}^\top\mathbf{W}_{t-1}),$$

$$\mathbf{Q}_t = \beta\mathbf{Q}_{t-1} - (1+\beta)\eta\mathbf{X}((\mathbf{R}_t^\top\mathbf{V}_t) \odot \mathbb{1}[\mathbf{X}^\top\mathbf{W}_t \geq 0]) + \beta\eta\mathbf{X}((\mathbf{R}_{t-1}^\top\mathbf{V}_{t-1}) \odot \mathbb{1}[\mathbf{X}^\top\mathbf{W}_{t-1} \geq 0]).$$

According to the NAG update rule, we have

$$\begin{aligned} \mathbf{R}_{t+1} &= \mathbf{V}_{t+1}\sigma(\mathbf{W}_{t+1}^\top\mathbf{X}) - \mathbf{Y} \\ &= (\mathbf{V}_t + \mathbf{P}_t)\sigma(\mathbf{W}_{t+1}^\top\mathbf{X}) - \mathbf{Y} \\ &= \mathbf{R}_t + \underbrace{\mathbf{P}_t\sigma(\mathbf{W}_t^\top\mathbf{X})}_{T_1} + \underbrace{\mathbf{V}_t\left(\sigma(\mathbf{W}_{t+1}^\top\mathbf{X}) - \sigma(\mathbf{W}_t^\top\mathbf{X})\right)}_{T_2} + \underbrace{\mathbf{P}_t\left(\sigma(\mathbf{W}_{t+1}^\top\mathbf{X}) - \sigma(\mathbf{W}_t^\top\mathbf{X})\right)}_{T_3}. \end{aligned}$$

For the first part, we have

$$\begin{aligned} T_1 &= \left(\beta\mathbf{P}_{t-1} - (1+\beta)\eta\mathbf{R}_t\sigma(\mathbf{X}^\top\mathbf{W}_t) + \beta\eta\mathbf{R}_{t-1}\sigma(\mathbf{X}^\top\mathbf{W}_{t-1})\right)\sigma(\mathbf{W}_t^\top\mathbf{X}) \\ &= \beta\mathbf{P}_{t-1}\sigma(\mathbf{W}_t^\top\mathbf{X}) - (1+\beta)\eta\mathbf{R}_t\sigma(\mathbf{X}^\top\mathbf{W}_t)\sigma(\mathbf{W}_t^\top\mathbf{X}) + \beta\eta\mathbf{R}_{t-1}\sigma(\mathbf{X}^\top\mathbf{W}_{t-1})\sigma(\mathbf{W}_t^\top\mathbf{X}) \\ &= \beta\mathbf{P}_{t-1}\sigma(\mathbf{W}_t^\top\mathbf{X}) - (1+\beta)\eta\mathbf{R}_t\sigma(\mathbf{X}^\top\mathbf{W}_t)\sigma(\mathbf{W}_t^\top\mathbf{X}) + \beta\eta\mathbf{R}_{t-1}\sigma(\mathbf{X}^\top\mathbf{W}_{t-1})(\sigma(\mathbf{W}_{t-1}^\top\mathbf{X})) \\ &\quad + \beta\eta\mathbf{R}_{t-1}\sigma(\mathbf{X}^\top\mathbf{W}_{t-1})(\sigma(\mathbf{W}_t^\top\mathbf{X}) - \sigma(\mathbf{W}_{t-1}^\top\mathbf{X})) \\ &= -(1+\beta)\eta\mathbf{R}_t\sigma(\mathbf{X}^\top\mathbf{W}_t)\sigma(\mathbf{W}_t^\top\mathbf{X}) + \beta\eta\mathbf{R}_{t-1}\sigma(\mathbf{X}^\top\mathbf{W}_{t-1})(\sigma(\mathbf{W}_{t-1}^\top\mathbf{X})) \\ &\quad + \beta(\mathbf{V}_t - \mathbf{V}_{t-1})\sigma(\mathbf{W}_t^\top\mathbf{X}) + \beta\eta\mathbf{R}_{t-1}\sigma(\mathbf{X}^\top\mathbf{W}_{t-1})(\sigma(\mathbf{W}_t^\top\mathbf{X}) - \sigma(\mathbf{W}_{t-1}^\top\mathbf{X})) \\ &= \beta\mathbf{R}_t - \beta\mathbf{R}_{t-1} - (1+\beta)\eta\mathbf{R}_t\sigma(\mathbf{X}^\top\mathbf{W}_t)\sigma(\mathbf{W}_t^\top\mathbf{X}) + \beta\eta\mathbf{R}_{t-1}\sigma(\mathbf{X}^\top\mathbf{W}_{t-1})(\sigma(\mathbf{W}_{t-1}^\top\mathbf{X})) \\ &\quad - \beta\mathbf{V}_{t-1}(\sigma(\mathbf{W}_t^\top\mathbf{X}) - \sigma(\mathbf{W}_{t-1}^\top\mathbf{X})) + \beta\eta\mathbf{R}_{t-1}\sigma(\mathbf{X}^\top\mathbf{W}_{t-1})(\sigma(\mathbf{W}_t^\top\mathbf{X}) - \sigma(\mathbf{W}_{t-1}^\top\mathbf{X})). \end{aligned}$$

For the second part, we have

$$\begin{aligned} T_2 &= \mathbf{V}_t\left(\sigma(\mathbf{W}_{t+1}^\top\mathbf{X}) - \sigma(\mathbf{W}_t^\top\mathbf{X})\right) \\ &= \mathbf{V}_t\left((\mathbf{W}_{t+1}^\top\mathbf{X}) \odot \mathbb{1}[\mathbf{W}_{t+1}^\top\mathbf{X} \geq 0] - (\mathbf{W}_t^\top\mathbf{X}) \odot \mathbb{1}[\mathbf{W}_t^\top\mathbf{X} \geq 0]\right) \\ &= \mathbf{V}_t\left((\mathbf{W}_{t+1}^\top\mathbf{X}) \odot \mathbb{1}[\mathbf{W}_0^\top\mathbf{X} \geq 0] \odot \mathbf{M} - (\mathbf{W}_t^\top\mathbf{X}) \odot \mathbb{1}[\mathbf{W}_0^\top\mathbf{X} \geq 0] \odot \mathbf{M}\right) \\ &\quad + \underbrace{\mathbf{V}_t\left((\mathbf{W}_{t+1}^\top\mathbf{X}) \odot \mathbb{1}[\mathbf{W}_{t+1}^\top\mathbf{X} \geq 0] \odot \mathbf{M}^\perp - (\mathbf{W}_t^\top\mathbf{X}) \odot \mathbb{1}[\mathbf{W}_t^\top\mathbf{X} \geq 0] \odot \mathbf{M}^\perp\right)}_{T_5} \\ &= \beta\mathbf{V}_t\left((\mathbf{Q}_{t-1}^\top\mathbf{X}) \odot \mathbb{1}[\mathbf{W}_0^\top\mathbf{X} \geq 0] \odot \mathbf{M}\right) + T_5 \\ &\quad - (1+\beta)\eta\underbrace{\mathbf{V}_t\left((((\mathbf{R}_t^\top\mathbf{V}_t) \odot \mathbb{1}[\mathbf{X}^\top\mathbf{W}_t \geq 0])^\top\mathbf{X}^\top\mathbf{X}) \odot \mathbb{1}[\mathbf{W}_t^\top\mathbf{X} \geq 0] \odot \mathbf{M}\right)}_{T_7} \\ &\quad + \beta\eta\underbrace{\mathbf{V}_t\left((((\mathbf{R}_{t-1}^\top\mathbf{V}_{t-1}) \odot \mathbb{1}[\mathbf{X}^\top\mathbf{W}_{t-1} \geq 0])^\top\mathbf{X}^\top\mathbf{X}) \odot \mathbb{1}[\mathbf{W}_{t-1}^\top\mathbf{X} \geq 0] \odot \mathbf{M}\right)}_{T_8}. \end{aligned}$$

For any $l \in [n]$, $j \in [N]$,

$$[T_7]_{l,j} = \sum_{i=1}^{2m} [\mathbf{V}_t]_{l,i} [(((\mathbf{R}_t^\top \mathbf{V}_t) \odot \mathbb{1}[\mathbf{X}^\top \mathbf{W}_t \geq 0])^\top \mathbf{X}^\top \mathbf{X}) \odot \mathbb{1}[\mathbf{W}_t^\top \mathbf{X} \geq 0] \odot \mathbf{M}]_{i,j}$$

$$= \sum_{i=1}^{2m} [\mathbf{V}_t]_{l,i} [(((\mathbf{R}_t^\top \mathbf{V}_t) \odot \mathbb{1}[\mathbf{X}^\top \mathbf{W}_t \geq 0])^\top \mathbf{X}^\top \mathbf{X})]_{i,j} \mathbb{1}[\mathbf{W}_t^\top \mathbf{X} \geq 0]_{i,j} [\mathbf{M}]_{i,j}$$

$$= \left( \sum_{i=1}^{2m} - \sum_{i \in S_j^\perp} \right) [\mathbf{V}_t]_{l,i} [(((\mathbf{V}_t^\top \mathbf{R}_t) \odot \mathbb{1}[\mathbf{W}_t^\top \mathbf{X} \geq 0]) \mathbf{X}^\top \mathbf{X})]_{i,j} \mathbb{1}[\mathbf{W}_t^\top \mathbf{X} \geq 0]_{i,j}$$

$$= \left( \sum_{i=1}^{2m} - \sum_{i \in S_j^\perp} \right) \sum_{k=1}^{N} [\mathbf{V}_t]_{l,i} [\mathbf{V}_t^\top \mathbf{R}_t]_{i,k} \mathbb{1}[\mathbf{W}_t^\top \mathbf{X} \geq 0]_{i,k} [\mathbf{X}^\top \mathbf{X}]_{k,j} \mathbb{1}[\mathbf{W}_t^\top \mathbf{X} \geq 0]_{i,j}$$

$$= \left( \sum_{i=1}^{2m} - \sum_{i \in S_j^\perp} \right) \sum_{k=1}^{N} \sum_{r=1}^{m} [\mathbf{V}_t]_{l,i} [\mathbf{V}_t]_{r,i} [\mathbf{R}_t]_{r,k} \mathbb{1}[\mathbf{W}_t^\top \mathbf{X} \geq 0]_{i,k} [\mathbf{X}^\top \mathbf{X}]_{k,j} \mathbb{1}[\mathbf{W}_t^\top \mathbf{X} \geq 0]_{i,j}.$$

For $T_7$, we have

$$[T_7]_{l,j} = \left( \sum_{i=1}^{2m} - \sum_{i \in S_j^\perp} \right) \sum_{k=1}^{N} \sum_{r=1}^{m} [\mathbf{A}_{i,t}]_{l,r} [\mathbf{R}_t]_{r,k} [\mathbf{B}_{i,t}]_{k,j}$$

$$= \left( \sum_{i=1}^{2m} - \sum_{i \in S_j^\perp} \right) [\mathbf{A}_{i,t} \mathbf{R}_t \mathbf{B}_{i,t}]_{l,j},$$

thus

$$T_7 = \sum_{i=1}^{2m} \mathbf{A}_{i,t} \mathbf{R}_t \mathbf{B}_{i,t} - T_9,$$

where $[T_9]_{l,j} = \sum_{i \in S_j^\perp} [\mathbf{A}_{i,t} \mathbf{R}_t \mathbf{B}_{i,t}]_{l,j}$. Meanwhile,

$$T_8 = \mathbf{V}_t \left( (((\mathbf{R}_{t-1}^\top \mathbf{V}_{t-1}) \odot \mathbb{1}[\mathbf{X}^\top \mathbf{W}_{t-1} \geq 0])^\top \mathbf{X}^\top \mathbf{X}) \odot \mathbb{1}[\mathbf{W}_{t-1}^\top \mathbf{X} \geq 0] \odot \mathbf{M} \right)$$

$$= (\mathbf{V}_{t-1} + \mathbf{P}_{t-1}) \left( (((\mathbf{R}_{t-1}^\top \mathbf{V}_{t-1}) \odot \mathbb{1}[\mathbf{X}^\top \mathbf{W}_{t-1} \geq 0])^\top \mathbf{X}^\top \mathbf{X}) \odot \mathbb{1}[\mathbf{W}_{t-1}^\top \mathbf{X} \geq 0] \odot \mathbf{M} \right)$$

$$= \sum_{i=1}^{2m} \mathbf{A}_{i,t-1} \mathbf{R}_{t-1} \mathbf{B}_{i,t-1} - T_{10}$$

$$+ \mathbf{P}_{t-1} \left( (((\mathbf{R}_{t-1}^\top \mathbf{V}_{t-1}) \odot \mathbb{1}[\mathbf{X}^\top \mathbf{W}_{t-1} \geq 0])^\top \mathbf{X}^\top \mathbf{X}) \odot \mathbb{1}[\mathbf{W}_{t-1}^\top \mathbf{X} \geq 0] \odot \mathbf{M} \right),$$

where $T_{10}$ is $T_9$ with $t$ replaced by $t-1$. The remaining part of $T_1$ and $T_2$ can be combined:

$$\beta \mathbf{V}_t \left( (\mathbf{Q}_{t-1}^\top \mathbf{X}) \odot \mathbb{1}[\mathbf{W}_0^\top \mathbf{X} \geq 0] \odot \mathbf{M} \right) - \beta \mathbf{V}_{t-1} (\sigma(\mathbf{W}_t^\top \mathbf{X}) - \sigma(\mathbf{W}_{t-1}^\top \mathbf{X}))$$

$$= \beta \mathbf{P}_{t-1} \left( (\mathbf{Q}_{t-1}^\top \mathbf{X}) \odot \mathbb{1}[\mathbf{W}_0^\top \mathbf{X} \geq 0] \odot \mathbf{M} \right) - \beta T_6,$$

where $T_6$ is $T_5$ with $t$ replaced by $t-1$. By changing the notations we get the results in the proposition. $\qquad \square$

### D.2 PROOF OF LEMMA 3.8

*Proof of Lemma 3.8.* Since $\mathbf{K}_0$ and $\mathbf{I}_{Nn}$ can be simultaneously diagonalized, eigenvalues of $\mathbf{T}_{\mathrm{GD}}$ correspond to $1 - \eta \lambda_i(\mathbf{K}_0)$, $i \in [Nn]$. Plugging in $\eta = \frac{1}{L}$ yields $\|\mathbf{T}_{\mathrm{GD}}\| = 1 - \frac{\mu}{L} = 1 - \frac{1}{\kappa(\mathbf{K}_0)}$.

For NAG, suppose $\lambda$ is an eigenvalue of $\mathbf{T}_{\mathrm{NAG}}$, then we have

$$\det(\mathbf{T}_{\mathrm{NAG}} - \lambda \mathbf{I}_{2Nn}) = \det((\beta + \lambda^2 - (1+\beta)\lambda)\mathbf{I}_{Nn} + (\eta(1+\beta)\lambda - \eta\beta)\mathbf{K}_0) = 0.$$

By simultaneous diagonalization the above equation becomes

$$\lambda^2 - (1+\beta)\lambda + \beta + \eta(1+\beta)\lambda_i(\mathbf{K}_0)\lambda - \eta\beta\lambda_i(\mathbf{K}_0) = 0$$

for some $1 \leq i \leq Nn$. Solving the equation yields

$$\lambda = \frac{1}{2}\left((1+\beta)(1-\eta\lambda_i(\mathbf{K}_0)) \pm \sqrt{(1-\eta\lambda_i(\mathbf{K}_0))\left(-4\beta + (1+\beta)^2(1-\eta\lambda_i(\mathbf{K}_0))\right)}\right).$$

By checking the monotonicity of $|\lambda|$ with respect to $1 - \eta\lambda_i(\mathbf{K}_0) \in [1-\eta L, 1-\eta\mu]$, we have

$$|\lambda| \leq \max\Bigg\{\frac{1}{2}\left((1+\beta)(1-\eta\mu) + \sqrt{(1-\eta\mu)\left(-4\beta + (1+\beta)^2(1-\eta\mu)\right)}\right),$$
$$\frac{1}{2}\left(-(1+\beta)(1-\eta L) + \sqrt{(1-\eta L)\left(-4\beta + (1+\beta)^2(1-\eta L)\right)}\right)\Bigg\}.$$

Choosing step size $\eta = \frac{1}{L}$ and momentum parameter $\beta = \frac{\sqrt{L}-\sqrt{\mu}}{\sqrt{L}+\sqrt{\mu}}$ yields $|\lambda| \leq 1 - \sqrt{\frac{\mu}{L}}$. $\qquad\square$

### D.3 PROOF OF LEMMA 4.6

*Proof of Lemma 4.6.* Similar to the proof of Lemma 3.8 in Appendix D.2, let $\lambda$ be an eigenvalue of $\mathbf{T}_{\mathrm{NAG}}$, then

$$\lambda = \frac{1}{2}\left((1+\beta)(1-\eta\lambda_i(\mathbf{K}_0)) \pm \sqrt{(1-\eta\lambda_i(\mathbf{K}_0))\left(-4\beta + (1+\beta)^2(1-\eta\lambda_i(\mathbf{K}_0))\right)}\right).$$

By Lemma 4.3, $\lambda_i(\mathbf{K}_0) = 0$ for any $i > Nr$, hence $\lambda = 1$ or $\lambda = \beta < 1$. The corresponding eigensubspaces are

$$\mathcal{H}_1 = \left\{(\mathbf{u}^\top, \mathbf{v}^\top)^\top \mid \mathbf{u} = \mathbf{v} \in \ker(\mathbf{K}_0)\right\},$$
$$\mathcal{H}_\beta = \left\{(\mathbf{u}^\top, \mathbf{v}^\top)^\top \mid \mathbf{u} = \beta\mathbf{v} \in \ker(\mathbf{K}_0)\right\}.$$

The dimensions are $\dim(\mathcal{H}_1) = \dim(\mathcal{H}_\beta) = \dim(\ker(\mathbf{K}_0)) = N(n-r)$. It is easy to verify that whenever $0 < \beta < 1$,

$$\mathcal{H}_1 \oplus \mathcal{H}_\beta = \ker(\mathbf{K}_0) \times \ker(\mathbf{K}_0).$$

The orthogonal complement of $\mathcal{H}_1 \oplus \mathcal{H}_\beta$ corresponds to the eigensubspace with positive eigenvalues. By checking the dimension and orthogonality, we have

$$(\mathcal{H}_1 \oplus \mathcal{H}_\beta)^\perp = \mathrm{col}(\mathbf{K}_0) \times \mathrm{col}(\mathbf{K}_0).$$

For $i \leq Nr$, $\lambda_i(\mathbf{K}_0) > 0$ and the corresponding subspace lie in $\mathrm{col}(\mathbf{K}_0) \times \mathrm{col}(\mathbf{K}_0)$. By Lemma 4.4, $\mathrm{col}(\mathbf{K}_0) = \mathbb{R}^N \otimes \mathrm{col}(\mathbf{Y})$, thus $\mathbf{u}$ and $\mathbf{v}$ reside in this positive eigensubspace. Similar to Appendix D.2, by choosing $\eta = \frac{1}{L}$ and $\beta = \frac{\sqrt{L}-\sqrt{\mu}}{\sqrt{L}+\sqrt{\mu}}$ we have $|\lambda| \leq 1 - \sqrt{\frac{\mu}{L}}$ and hence show the contraction of $(\mathbf{u}^\top, \mathbf{v}^\top)^\top$. The contraction of $\mathbf{r}_t$ and $\boldsymbol{\xi}_t$ follows immediately from Lemma 4.5. $\quad\square$

## E  PROOF FOR KERNEL SHIFT

We first state an auxiliary result on the mean absolute value of the product of correlated Gaussian random variables.

**Proposition E.1** (Corollary 3.1 in Li & Wei (2009)). *Let* $\mathbf{u}, \mathbf{v} \in \mathbb{R}^N$, $\boldsymbol{\phi} \sim \mathcal{N}(0, \mathbf{I}_N)$, $\rho := \frac{\mathbf{u}^\top\mathbf{v}}{\|\mathbf{u}\|\|\mathbf{v}\|}$, *then*

$$\mathbb{E}\left|(\mathbf{u}^\top\boldsymbol{\phi})(\mathbf{v}^\top\boldsymbol{\phi})\right| = \frac{2}{\pi}\left(\sqrt{1-\rho^2} + \rho\arcsin\rho\right)\|\mathbf{u}\|\|\mathbf{v}\| \geq \frac{2}{\pi}\|\mathbf{u}\|\|\mathbf{v}\|.$$

With this result, we show the kernel shift can be properly controlled by the change in weights with high probability.

### E.1 PROOF OF LEMMA 3.9

*Proof of Lemma 3.9.* Given our coupled initialization, we can consider the first $m$ neurons and then multiply the final result by a factor of 2, as we have $\mathbf{A}_{i,0} = \mathbf{A}_{i+m,0}$ and $\mathbf{B}_{i,0} = \mathbf{B}_{i+m,0}$ for all $i \in [m]$ and the radius $R_1$ and $R_2$ are uniform among all $i \in [2m]$. We decompose the kernel shift as follows:

$$
\|\mathbf{K}_0 - \mathbf{K}_t\| \leq 2 \left\| \sum_{i=1}^{m} \left( \mathbf{B}_{i,t}^\top \otimes \mathbf{A}_{i,t} - \mathbf{B}_{i,0}^\top \otimes \mathbf{A}_{i,0} \right) \right\|_{\mathrm{F}}
$$

$$
\leq 2 \left\| \sum_{i=1}^{m} \left( \mathbf{B}_{i,t} - \mathbf{B}_{i,0} \right)^\top \otimes \mathbf{A}_{i,0} \right\|_{\mathrm{F}} + 2 \left\| \sum_{i=1}^{m} \mathbf{B}_{i,t}^\top \otimes \left( \mathbf{A}_{i,t} - \mathbf{A}_{i,0} \right) \right\|_{\mathrm{F}}.
$$

For the first part, we have

$$
\left\| \sum_{i=1}^{m} \left( \mathbf{B}_{i,t} - \mathbf{B}_{i,0} \right)^\top \otimes \mathbf{A}_{i,0} \right\|_{\mathrm{F}}^2 = \sum_{p=1}^{n} \sum_{q=1}^{n} \sum_{j=1}^{N} \sum_{k=1}^{N} \left( \sum_{i=1}^{m} [\mathbf{B}_{i,t} - \mathbf{B}_{i,0}]_{j,k} [\mathbf{A}_{i,0}]_{p,q} \right)^2
$$

$$
\leq \sum_{p=1}^{n} \sum_{q=1}^{n} \sum_{j=1}^{N} \sum_{k=1}^{N} \left( \sum_{i=1}^{m} s_{i,j,k} a_{i,p,q} \right)^2,
$$

where we denote

$$
a_{i,p,q} := |[\mathbf{A}_{i,0}]_{p,q}|, \quad s_{i,j,k} := |[\mathbf{J}_t]_{i,j}[\mathbf{J}_t]_{i,k} - [\mathbf{J}_0]_{i,j}[\mathbf{J}_0]_{i,k}|.
$$

For notation simplicity, we fix $p, q, j, k$ for now and denote $s_i = s_{i,j,k}$ and $a_i = a_{i,p,q}$. By construction, $s_i$'s are independent Bernoulli random variables, and

$$
\mathbb{E}[s_i] = \mathbb{P}(s_i = 1) \leq \frac{2R_2}{c_2},
$$

where the inequality follows from Lemma G.2 in Munteanu et al. (2022). Therefore, we can construct another set of independent Bernoulli random variables $X_1, \ldots, X_m$ such that $\mathbb{E}[X_i] = \frac{2R_2}{c_2}$ and $X_i \geq s_i$ for all $i \in [m]$. By weighted Chernoff inequality (Lemma B.3), we have the following bound for fixed $a_i$'s:

$$
\mathbb{P} \left( \sum_{i=1}^{m} a_i s_i \geq \frac{3R_2}{c_2} \sum_{i=1}^{m} a_i \right) \leq \mathbb{P} \left( \sum_{i=1}^{m} a_i X_i \geq \frac{3}{2} \cdot \sum_{i=1}^{m} a_i \mathbb{E}[X_i] \right)
$$

$$
\leq \exp \left( -\frac{1}{10} \frac{\sum_{i=1}^{m} a_i \cdot \frac{2R_2}{c_2}}{\max_{i \in [m]} a_i} \right). \tag{14}
$$

Now we study the concentration properties for $a_i$'s to ensure the above bound is valid. By Proposition E.1 we have

$$
\mathbb{E}[a_i] = c_1^2 \mathbb{E} \left| [\mathbf{\Phi}]_{p,i} [\mathbf{\Phi}]_{q,i} \right| = c_1^2 \mathbb{E} \left| (\mathbf{e}_p^\top \boldsymbol{\phi}_i)(\mathbf{e}_q^\top \boldsymbol{\phi}_i) \right| \geq \frac{2c_1^2}{\pi}.
$$

On the other hand, since $[\mathbf{\Phi}]_{p,i}$ and $[\mathbf{\Phi}]_{q,i}$ are Gaussian, $a_i$ and its centered version are sub-exponential with bounded $\psi_1$-norm

$$
\|a_i - \mathbb{E}[a_i]\|_{\psi_1} \lesssim \|a_i\|_{\psi_1} \lesssim c_1^2.
$$

By plugging in $t = \frac{c_1^2 m}{\pi}$ in Bernstein's inequality (Lemma B.2), we have

$$
\mathbb{P} \left( \left| \sum_{i=1}^{m} (a_i - \mathbb{E}[a_i]) \right| \geq \frac{c_1^2 m}{\pi} \right) \leq 2 \exp \left( -c' m \right),
$$

where $c'$ is an absolute constant. Therefore, for any $\delta_2 \in (0,1)$, by setting $m \geq \Omega(\log \frac{2n}{\delta_2})$, with probability at least $1 - \delta_2$ we have

$$
\sum_{i=1}^{m} a_{i,p,q} \geq \frac{c_1^2 m}{\pi}, \ \forall p, q \in [n].
$$

For the denominator, by union bound on sub-Gaussian tails, it holds

$$\mathbb{P}\left(\exists i \in [m], \exists p \in [n], \ |[\mathbf{\Phi}]_{p,i}| > \sqrt{c \log \frac{2mn}{\delta_3}}\right) \le \delta_3$$

for any $\delta_3 \in (0,1)$, where $c$ is an absolute constant. Thus with probability at least $1 - \delta_3$, we have

$$a_{i,p,q} \lesssim c_1^2 \log\left(\frac{2mn}{\delta_3}\right), \ \forall i \in [m], \forall p, q \in [n].$$

Plug the lower bound on the numerator and upper bound on the denominator back to (14), with probability at least $1 - \delta_2 - \delta_3$ we have

$$\mathbb{P}\left(\sum_{i=1}^m a_i s_i \ge \frac{3R_2}{c_2} \sum_{i=1}^m a_i\right) \le \exp\left(-\frac{c'' m R_2}{c_2 \log \frac{2mn}{\delta_3}}\right)$$

for some absolute constant $c'' > 0$. For any $\delta_4 \in (0,1)$, by setting

$$m = \max\left\{\Omega\left(\frac{c_2}{R_2} \cdot \log\left(\frac{2mn}{\delta_3}\right) \log\left(\frac{n}{\delta_4}\right)\right), \Omega\left(\log\left(\frac{2n}{\delta_2}\right)\right)\right\},$$

we have with probability at least $1 - \delta_2 - \delta_3 - \delta_4$ that

$$\left\|\sum_{i=1}^m (\mathbf{B}_{i,t} - \mathbf{B}_{i,0})^\top \otimes \mathbf{A}_{i,0}\right\|_F^2 \le \left(\frac{3NR_2}{c_2}\right)^2 \sum_{p=1}^n \sum_{q=1}^n \left(\sum_{i=1}^m |[\mathbf{A}_{i,0}]_{p,q}|\right)^2$$

$$\le \left(\frac{3NR_2}{c_2}\right)^2 \sum_{p=1}^n \sum_{q=1}^n \left(m \sum_{i=1}^m |[\mathbf{A}_{i,0}]_{p,q}|^2\right)$$

$$\le \left(\frac{3NR_2}{c_2}\right)^2 m \sum_{i=1}^m \|\mathbf{A}_{i,0}\|_F^2$$

$$\le \left(\frac{3mNR_2}{c_2}\right)^2 R_V^4,$$

where the second line uses the Cauchy–Schwarz inequality.

For the second part, we have

$$\left\|\sum_{i=1}^m \mathbf{B}_{i,t}^\top \otimes (\mathbf{A}_{i,t} - \mathbf{A}_{i,0})\right\|_F^2 \le \sum_{p=1}^n \sum_{q=1}^n \sum_{j=1}^N \sum_{k=1}^N \left(\sum_{i=1}^m [\mathbf{B}_{i,t}]_{j,k}^2\right)\left(\sum_{i=1}^m [\mathbf{A}_{i,t} - \mathbf{A}_{i,0}]_{p,q}^2\right)$$

$$\le mN^2 \sum_{i=1}^m \|\mathbf{A}_{i,t} - \mathbf{A}_{i,0}\|_F^2$$

$$\le 4m^2 N^2 R_1^2 (2R_V + R_1)^2,$$

where we use Cauchy–Schwarz for the first line and $[\mathbf{B}_{i,t}]_{j,k} \le 1$ for the second line. Combining the results, we get

$$\|\mathbf{K}_0 - \mathbf{K}_t\| \le 6mN\frac{R_2}{c_2}R_V^2 + 4mNR_1(2R_V + R_1).$$

Meanwhile, by 1-Lipschitzness of ReLU, we have

$$\|\mathbf{G}_t\| \le \left\|\sigma(\mathbf{X}^\top \mathbf{W}_t)\sigma(\mathbf{W}_t^\top \mathbf{X})\right\| \le \left\|\mathbf{W}_t^\top \mathbf{X}\right\|_F^2 \le 2m(R_W + R_2)^2 \|\mathbf{X}\|^2 \le 2mN(R_W + R_2)^2.$$

Picking $\delta_i = \frac{\delta}{3}$ for $i = 2, 3, 4$ we get the final result. $\qquad\square$

## E.2 LOW-RANK KERNEL SHIFT

For low-rank responses, we show the bound on kernel shift still holds.

**Lemma E.2** (Low-rank kernel shift). *Lemma 3.9 holds for sketching initialization* (8).

*Proof of Lemma E.2.* Most of the proof follows Appendix E.1. We only need to show that under initialization (8), the concentration of $\sum_{i=1}^m a_i s_i$ is valid, where $a_i$ and $s_i$ are defined as in Appendix E.1. By weighted Chernoff inequality (Lemma B.3), we have

$$\mathbb{P}\left(\sum_{i=1}^m a_i s_i \geq \frac{3R_2}{c_2} \sum_{i=1}^m a_i\right) \leq \exp\left(-\frac{1}{10} \frac{\sum_{i=1}^m a_i \cdot \frac{2R_2}{c_2}}{\max_{i \in [m]} a_i}\right). \tag{15}$$

Given sketching initialization (8) and Proposition E.1, we have

$$\mathbb{E}[a_i] = c_1^2 \mathbb{E}\left|[\mathbf{Y\Phi}]_{p,i}[\mathbf{Y\Phi}]_{q,i}\right| \geq \frac{2c_1^2}{\pi} \|[\mathbf{Y}]_{p,:}\| \|[\mathbf{Y}]_{q,:}\|.$$

On the other hand, we have

$$\|[\mathbf{Y\Phi}]_{p,i}\|_{\psi_2} \lesssim \|[\mathbf{Y}]_{p,:}\|,$$

thus $a_i$ and its centered version are sub-exponential with $\psi_1$-norm bounded by

$$\|a_i - \mathbb{E}[a_i]\|_{\psi_1} \lesssim \|a_i\|_{\psi_1} \lesssim c_1^2 \|[\mathbf{Y}]_{p,:}\| \|[\mathbf{Y}]_{q,:}\|.$$

By plugging in $t = \frac{c_1^2 m \|[\mathbf{Y}]_{p,:}\| \|[\mathbf{Y}]_{q,:}\|}{\pi}$ in Bernstein's inequality (Lemma B.2), we have

$$\mathbb{P}\left(\left|\sum_{i=1}^m (a_i - \mathbb{E}[a_i])\right| \geq \frac{c_1^2 m}{\pi} \|[\mathbf{Y}]_{p,:}\| \|[\mathbf{Y}]_{q,:}\|\right) \leq 2\exp\left(-c'm\right)$$

for some absolute constant $c'$. Therefore, for any $\delta_2 \in (0, 1)$, by setting $m \geq \Omega(\log \frac{2n}{\delta_2})$, we have

$$\sum_{i=1}^m a_{i,p,q} \geq \frac{c_1^2 m}{\pi} \|[\mathbf{Y}]_{p,:}\| \|[\mathbf{Y}]_{q,:}\|, \ \forall p, q \in [n].$$

with probability at least $1 - \delta_2$. For the denominator, by union bound on sub-Gaussian tails, it holds

$$\mathbb{P}\left(\exists i \in [m], \exists p \in [n], \ |[\mathbf{Y\Phi}]_{p,i}| > \sqrt{c \log \frac{2mn}{\delta_3}} \|[\mathbf{Y}]_{p,:}\|\right) \leq \delta_3$$

for any $\delta_3 \in (0, 1)$, where $c$ is an absolute constant. Thus with probability at least $1 - \delta_3$, we have

$$a_{i,p,q} \lesssim c_1^2 \|[\mathbf{Y}]_{p,:}\| \|[\mathbf{Y}]_{q,:}\| \log\left(\frac{2mn}{\delta_3}\right), \ \forall i \in [m], \forall p, q \in [n].$$

Plug the lower bound on the numerator and upper bound on the denominator back to (15), with probability at least $1 - \delta_2 - \delta_3$ we have

$$\mathbb{P}\left(\sum_{i=1}^m a_i s_i \geq \frac{3R_2}{c_2} \sum_{i=1}^m a_i\right) \leq \exp\left(-\frac{c'' m R_2}{c_2 \log \frac{2mn}{\delta_3}}\right)$$

for some absolute constant $c'' > 0$. The rest of the proof follows the same lines as Appendix E.1. □

# F PROOF FOR SUBSPACE ALIGNMENT

## F.1 PROOF OF LEMMA 4.4

*Proof of Lemma 4.4.* By sketching initialization (8), $\text{col}(\mathbf{V}_0) = \text{col}(\mathbf{Y\Phi}) \subseteq \text{col}(\mathbf{Y})$. Since $\mathbf{K}_0 = \sum_{i=1}^{2m} \mathbf{B}_i^\top \otimes \mathbf{A}_i$ is symmetric, its row space is equivalent to column space. For any

$\mathbf{w} = (\mathbf{w}_1^\top, \ldots, \mathbf{w}_N^\top)^\top \in \mathbb{R}^{Nn}$, $\mathbf{w}_j \in \mathbb{R}^n$, $\forall j \in [N]$, it holds for every $i \in [2m]$ that

$$(\mathbf{B}_i^\top \otimes \mathbf{A}_i)\mathbf{w} = \begin{pmatrix} [\mathbf{B}_i]_{1,1}\mathbf{A}_i & \ldots & [\mathbf{B}_i]_{1,N}\mathbf{A}_i \\ \vdots & & \vdots \\ [\mathbf{B}_i]_{N,1}\mathbf{A}_i & \ldots & [\mathbf{B}_i]_{N,N}\mathbf{A}_i \end{pmatrix} \begin{pmatrix} \mathbf{w}_1 \\ \vdots \\ \mathbf{w}_N \end{pmatrix}$$

$$= \begin{pmatrix} \sum_{j=1}^N [\mathbf{B}_i]_{1,j}\mathbf{A}_i\mathbf{w}_j \\ \vdots \\ \sum_{j=1}^N [\mathbf{B}_i]_{N,j}\mathbf{A}_i\mathbf{w}_j \end{pmatrix} \in \mathbb{R}^N \otimes \mathrm{col}(\mathbf{Y}),$$

where the second line is from that $\mathrm{col}(\mathbf{A}_i) = \mathrm{col}([\mathbf{V}_0]_{:,i}[\mathbf{V}_0]_{:,i}^\top) \subseteq \mathrm{col}(\mathbf{V}_0) \subseteq \mathrm{col}(\mathbf{Y})$. Therefore, we have $\mathrm{col}(\mathbf{K}_0) \subseteq \mathbb{R}^N \otimes \mathrm{col}(\mathbf{Y})$. By assumption, $\mathrm{rank}(\mathbf{Y}) = r$, hence $\dim(\mathbb{R}^N \otimes \mathrm{col}(\mathbf{Y})) = Nr$. By Lemma 4.3, $\lambda_{rN}(\mathbf{K}_0) \geq \frac{c_1^2 d\lambda\lambda_r(\mathbf{Y}\mathbf{Y}^\top)}{2}$, thus we have $Nr \leq \mathrm{rank}(\mathbf{K}_0) \leq \dim(\mathbb{R}^N \otimes \mathrm{col}(\mathbf{Y})) = Nr$, which implies that $\mathrm{col}(\mathbf{K}_0) = \mathbb{R}^N \otimes \mathrm{col}(\mathbf{Y})$. $\qquad\square$

### F.2 PROOF OF LEMMA 4.5

*Proof of Lemma 4.5.* We prove the lemma by induction. By Lemma 4.3, $\mathrm{col}(\mathbf{V}_0) = \mathrm{col}(\mathbf{Y})$. For $t = 0$, $\mathrm{col}(\mathbf{R}_0) = \mathrm{col}(\mathbf{V}_0\sigma(\mathbf{W}_0^\top\mathbf{X}) - \mathbf{Y}) \subseteq \mathrm{col}(\mathbf{Y})$, thus $\mathbf{r}_0 = \mathrm{vec}(\mathbf{R}_0) \in \mathbb{R}^N \otimes \mathrm{col}(\mathbf{Y})$. For $\boldsymbol{\xi}_0$, we examine $\boldsymbol{\zeta}_0$ and $\boldsymbol{\iota}_0$. By definition of $\boldsymbol{\zeta}_t$, recursive definition of $\mathbf{P}_t$, $\mathbf{R}_{-1} = \mathbf{R}_0$ and $\mathbf{P}_{-1} = 0$ in Proposition 3.7, we have $\mathrm{span}(\boldsymbol{\zeta}_0) \subseteq (\mathbb{R}^N \otimes (\mathrm{col}(\mathbf{R}_0) \cup \mathrm{col}(\mathbf{V}_0) \cup \mathrm{col}(\boldsymbol{\varphi}_0))) \subseteq (\mathbb{R}^N \otimes (\mathrm{col}(\mathbf{Y}) \cup \mathrm{col}(\boldsymbol{\varphi}_0)))$. By definition of $\boldsymbol{\varphi}_t$, for any $j \in [N]$,

$$[\boldsymbol{\varphi}_t]_{:,j} = \sum_{i \in S_j^\perp} [\mathbf{A}_{i,t}\mathbf{R}_t\mathbf{B}_{i,t}]_{:,j} \in \mathrm{col}(\mathbf{V}_t),$$

where the inclusion is due to $\mathbf{A}_{i,t} = [\mathbf{V}_t]_{:,i}[\mathbf{V}_t]_{:,i}^\top$. Therefore, $\mathrm{span}(\boldsymbol{\zeta}_0) \subseteq \mathbb{R}^N \otimes \mathrm{col}(\mathbf{Y})$. For $\boldsymbol{\iota}_t$, we have

$$\mathbf{K}_t\mathbf{r}_t = \begin{pmatrix} [\mathbf{B}_{i,t}]_{1,1}\mathbf{A}_{i,t} & \ldots & [\mathbf{B}_{i,t}]_{1,N}\mathbf{A}_{i,t} \\ \vdots & & \vdots \\ [\mathbf{B}_{i,t}]_{N,1}\mathbf{A}_{i,t} & \ldots & [\mathbf{B}_{i,t}]_{N,N}\mathbf{A}_{i,t} \end{pmatrix} \begin{pmatrix} [\mathbf{R}_t]_{:,1} \\ \vdots \\ [\mathbf{R}_t]_{:,N} \end{pmatrix}$$

$$= \begin{pmatrix} \sum_{j=1}^N [\mathbf{B}_{i,t}]_{1,j}[\mathbf{A}_{i,t}\mathbf{R}_t]_{:,j} \\ \vdots \\ \sum_{j=1}^N [\mathbf{B}_{i,t}]_{N,j}[\mathbf{A}_{i,t}\mathbf{R}_t]_{:,j} \end{pmatrix} \in \mathbb{R}^N \otimes \mathrm{col}(\mathbf{V}_t).$$

Similarly, we have

$$\mathbf{G}_t\mathbf{r}_t = \left((\sigma(\mathbf{x}^\top\mathbf{W}_t)\sigma(\mathbf{W}_t^\top\mathbf{x})) \otimes \mathbf{I}_n\right)\mathbf{r}_t$$

$$= \begin{pmatrix} (\sigma(\mathbf{x}_1^\top\mathbf{W}_t)\sigma(\mathbf{W}_t^\top\mathbf{x}_1))\mathbf{I}_n & \ldots & (\sigma(\mathbf{x}_1^\top\mathbf{W}_t)\sigma(\mathbf{W}_t^\top\mathbf{x}_N))\mathbf{I}_n \\ \vdots & & \vdots \\ (\sigma(\mathbf{x}_N^\top\mathbf{W}_t)\sigma(\mathbf{W}_t^\top\mathbf{x}_1))\mathbf{I}_n & \ldots & (\sigma(\mathbf{x}_N^\top\mathbf{W}_t)\sigma(\mathbf{W}_t^\top\mathbf{x}_N))\mathbf{I}_n \end{pmatrix} \begin{pmatrix} [\mathbf{R}_t]_{:,1} \\ \vdots \\ [\mathbf{R}_t]_{:,N} \end{pmatrix}$$

$$= \begin{pmatrix} \sum_{j=1}^N (\sigma(\mathbf{x}_1^\top\mathbf{W}_t)\sigma(\mathbf{W}_t^\top\mathbf{x}_j))[\mathbf{R}_t]_{:,j} \\ \vdots \\ \sum_{j=1}^N (\sigma(\mathbf{x}_N^\top\mathbf{W}_t)\sigma(\mathbf{W}_t^\top\mathbf{x}_j))[\mathbf{R}_t]_{:,j} \end{pmatrix} \in \mathbb{R}^N \otimes \mathrm{col}(\mathbf{R}_t).$$

Therefore, $\boldsymbol{\iota}_0 \in \mathbb{R}^N \otimes (\mathrm{col}(\mathbf{V}_0) \cup \mathrm{col}(\mathbf{R}_0)) \subseteq \mathbb{R}^N \otimes \mathrm{col}(\mathbf{Y})$. Suppose $\{\mathbf{r}_s, \boldsymbol{\xi}_s\}_{s \leq t} \subseteq \mathbb{R}^N \otimes \mathrm{col}(\mathbf{Y})$, then by Proposition 3.7, we have $\mathbf{r}_{t+1} \in \mathrm{span}(\{\mathbf{r}_t, \mathbf{r}_{t-1}, \boldsymbol{\xi}_t\}) \cup \mathrm{col}(\mathbf{K}_0) = \mathbb{R}^N \otimes \mathrm{col}(\mathbf{Y})$, and $\boldsymbol{\xi}_t \in \mathbb{R}^N \otimes$ follows immediately. Thus we complete the proof by induction. $\qquad\square$

# G PROOF FOR CONVERGENCE

## G.1 AUXILIARY LEMMA

**Lemma G.1.** *Suppose $\{a_t\}_{t\geq 0}$ and $\{b_t\}_{t\geq 0}$ are two non-negative sequences satisfying*

$$a_{t+1} \leq \rho \cdot a_t + b_t, \quad b_t \leq \theta^t \cdot c_0,$$

*where $0 \leq \rho < \theta < 1$, $c_0 \geq 0$, then the following holds for all $t \geq 0$:*

$$a_t \leq \theta^t \cdot \left(a_0 + \frac{c_0}{\theta - \rho}\right).$$

*Proof of Lemma G.1.* The inequality holds trivially for $t = 0$. For $t \geq 0$, we have

$$a_{t+1} = \rho^{t+1} \cdot a_0 + \sum_{s=0}^{t} \rho^{t-s}\theta^s \cdot c_0$$

$$= \rho^{t+1} \cdot a_0 + \frac{\theta^{t+1} - \rho^{t+1}}{\theta - \rho} \cdot c_0$$

$$= \theta^{t+1} \cdot \left(a_0 + \frac{1}{\theta - \rho} \cdot c_0\right).$$

$\square$

## G.2 PROOF OF LEMMA 3.10

*Proof of Lemma 3.10.* The proof follows the proof of Claim 3.12 in (Song & Yang, 2019). Recall

$$S_j^\perp = \left\{i \in [2m] \mid \exists t \geq 0, \; \mathbb{1}[\mathbf{W}_t^\top \mathbf{X} \geq 0]_{i,j} \neq \mathbb{1}[\mathbf{W}_0^\top \mathbf{X} \geq 0]_{i,j}\right\}.$$

Given the condition that $\|[\mathbf{W}_t]_{:,i} - [\mathbf{W}_0]_{:,i}\| \leq R_2$ for all $t \geq 0$, we can relax the constraint in $S_j^\perp$ and consider the following random variable $s_j$ for each $j \in [N]$:

$$s_j := \sum_{i=1}^{2m} \mathbb{1}\left[\exists \mathbf{w} \in \mathbb{R}^d, \|\mathbf{w} - [\mathbf{W}_0]_{:,i}\| \leq R_2, (\mathbf{w}^\top \mathbf{x}_j) \cdot ([\mathbf{W}_0]_{:,i}^\top \mathbf{x}_j) < 0\right].$$

By definition, we always have $\left|S_j^\perp\right| \leq s_j$, so we only need to bound $s_j$ with high probability. Denote event $A_{i,j} := \left\{\exists \mathbf{w} \in \mathbb{R}^d, \|\mathbf{w} - [\mathbf{W}_0]_{:,i}\| \leq R_2, (\mathbf{w}^\top \mathbf{x}_j) \cdot ([\mathbf{W}_0]_{:,i}^\top \mathbf{x}_j) < 0\right\}$. By anti-concentration inequality of Gaussian (Lemma A.4 in Song & Yang 2019),

$$\mathbb{P}(A_{i,j}) = \mathbb{P}_{z \sim \mathcal{N}(0,c_2)}(|z| < R_2) \leq \frac{2R_2}{\sqrt{2\pi c_2}} < \frac{R_2}{c_2},$$

hence $\mathbb{E}[\mathbb{1}[A_{i,j}]] \leq \frac{R_2}{c_2}$. We can also bound the absolute value $|\mathbb{1}[A_{i,j}]| \leq 1$ and variance

$$\mathbb{E}[(\mathbb{1}[A_{i,j}] - \mathbb{E}[\mathbb{1}[A_{i,j}]])^2] = \mathbb{E}[\mathbb{1}[A_{i,j}]^2] - (\mathbb{E}[\mathbb{1}[A_{i,j}]])^2 \leq \mathbb{E}[\mathbb{1}[A_{i,j}]] \leq \frac{R_2}{c_2}.$$

Since $\{A_{i,j}\}_{i=1}^m$ are independent, we can apply Bernstein's inequality and get that for all $\Delta > 0$,

$$\mathbb{P}\left(\sum_{i=1}^{m} \mathbb{1}[A_{i,j}] > m \cdot \frac{R_2}{c_2} + \Delta\right) \leq \exp\left(-\frac{\Delta^2/2}{m\frac{R_2}{c_2} + \Delta/3}\right).$$

By setting $\Delta = \frac{3mR_2}{c_2}$ and using the fact that $A_{i,j} = A_{m+i,j}$ for $i \in [m]$ due to the coupled initialization, we have $s_j = 2\sum_{i=1}^m \mathbb{1}[A_{i,j}]$ and thus

$$\mathbb{P}\left(s_j > \frac{8mR_2}{c_2}\right) \leq \exp\left(-\frac{9mR_2}{4c_2}\right) < \exp\left(-mR_2/c_2\right).$$

Lemma 3.10 follows immediately from the union bound on $j \in [N]$. $\square$

### G.3 PROOF OF LEMMA 3.11

*Proof of Lemma 3.11.* By assumptions and the 1-Lipschitzness of $\sigma(\cdot)$, we have

$$\left\|[\sigma(\mathbf{X}^\top \mathbf{W}_s)]_{:,i}\right\| \leq \|\mathbf{X}\|_{\mathrm{F}} \left(\|[\mathbf{W}_0]_{:,i}\| + \|[\mathbf{W}_s - \mathbf{W}_0]_{:,i}\|\right) \leq \|\mathbf{X}\|_{\mathrm{F}} \left(\|[\mathbf{W}_0]_{:,i}\| + R_2\right) \leq C_W.$$

By recursion, we have

$$\mathbf{P}_t = \beta \mathbf{P}_{t-1} - (1+\beta)\eta \mathbf{R}_t \sigma(\mathbf{X}^\top \mathbf{W}_t) + \beta\eta \mathbf{R}_{t-1}\sigma(\mathbf{X}^\top \mathbf{W}_{t-1})$$

$$= -\eta \mathbf{R}_t \sigma(\mathbf{X}^\top \mathbf{W}_t) - \beta\eta \sum_{s=0}^{t} \beta^{t-s}\mathbf{R}_s \sigma(\mathbf{X}^\top \mathbf{W}_s) + \eta\beta^{t+1}\mathbf{R}_0\sigma(\mathbf{X}^\top \mathbf{W}_0),$$

thus

$$\|[\mathbf{P}_t]_{:,i}\| \leq \left(\theta^t + \beta^{t+1} + \beta \sum_{s=0}^{t} \beta^{t-s}\theta^s\right)\eta C_1 \|\mathbf{R}_0\|_{\mathrm{F}} C_W$$

$$\leq \frac{\theta^t}{1-\theta}\eta C_1 \|\mathbf{R}_0\|_{\mathrm{F}} C_W,$$

and

$$\|\mathbf{P}_t\|_{\mathrm{F}} \leq \frac{\theta^t}{1-\theta}\eta C_1 \sqrt{2m} \|\mathbf{R}_0\|_{\mathrm{F}} C_W,$$

where we use $C_1 \geq 1$ and $\beta \leq \theta^2 < \theta < 1$. Similarly, we have

$$\mathbf{Q}_t = -\eta \nabla_W \mathcal{L}(\mathbf{V}_t, \mathbf{W}_t) - \beta\eta \sum_{s=0}^{t} \beta^{t-s}\nabla_W \mathcal{L}(\mathbf{V}_s, \mathbf{W}_s) + \eta\beta^{t+1}\nabla_W \mathcal{L}(\mathbf{V}_0, \mathbf{W}_0)$$

$$= -\eta \mathbf{X}((\mathbf{R}_t^\top \mathbf{V}_t) \odot \mathbb{1}[\mathbf{X}^\top \mathbf{W}_t \geq 0]) + \eta\beta^{t+1}\mathbf{X}((\mathbf{R}_0^\top \mathbf{V}_0) \odot \mathbb{1}[\mathbf{X}^\top \mathbf{W}_0 \geq 0])$$

$$- \beta\eta \sum_{s=0}^{t} \beta^{t-s}\mathbf{X}((\mathbf{R}_s^\top \mathbf{V}_s) \odot \mathbb{1}[\mathbf{X}^\top \mathbf{W}_s \geq 0]).$$

We can bound the norm of each column of $\mathbf{Q}_t$ by

$$\|[\mathbf{Q}_t]_{:,i}\| \leq \eta \|\mathbf{X}\|_{\mathrm{F}} \left(\left\|[\mathbf{R}_t^\top \mathbf{V}_t]_{:,i}\right\| + \beta \sum_{s=0}^{t} \beta^{t-s} \left\|[\mathbf{R}_s^\top \mathbf{V}_s]_{:,i}\right\| + \beta^{t+1} \left\|[\mathbf{R}_0^\top \mathbf{V}_0]_{:,i}\right\|\right)$$

$$\leq \frac{\theta^t}{1-\theta}\eta C_1 \|\mathbf{R}_0\|_{\mathrm{F}} C_V,$$

and bound the overall Frobenius norm by

$$\|\mathbf{Q}_t\|_{\mathrm{F}} \leq \frac{\theta^t}{1-\theta}\eta C_1 \sqrt{2m} \|\mathbf{R}_0\|_{\mathrm{F}} C_V.$$

Next, we bound $\|\boldsymbol{\zeta}_t\|$ and $\|\boldsymbol{\iota}_t\|$. We begin with $\|\boldsymbol{\zeta}_t\|$.

$$\|\boldsymbol{\zeta}_t\| = \underbrace{\left\|\beta\eta \mathbf{R}_{t-1}\sigma(\mathbf{X}^\top \mathbf{W}_{t-1})(\sigma(\mathbf{W}_t^\top \mathbf{X}) - \sigma(\mathbf{W}_{t-1}^\top \mathbf{X}))\right\|_{\mathrm{F}}}_{\ell_1}$$

$$+ \underbrace{\left\|\beta\eta \mathbf{P}_{t-1}\left((((\mathbf{R}_{t-1}^\top \mathbf{V}_{t-1}) \odot \mathbb{1}[\mathbf{X}^\top \mathbf{W}_{t-1} \geq 0])^\top \mathbf{X}^\top \mathbf{X}) \odot \mathbb{1}[\mathbf{W}_{t-1}^\top \mathbf{X} \geq 0] \odot \mathbf{M}\right)\right\|_{\mathrm{F}}}_{\ell_2}$$

$$+ \underbrace{\left\|T_5 - \beta T_6 + (1+\beta)\eta T_9 - \beta\eta T_{10}\right\|_{\mathrm{F}}}_{\ell_3}$$

$$+ \underbrace{\left\|\beta\mathbf{P}_{t-1}\left((\mathbf{Q}_{t-1}^\top \mathbf{X}) \odot \mathbb{1}[\mathbf{W}_0^\top \mathbf{X} \geq 0] \odot \mathbf{M}\right) + T_3\right\|_{\mathrm{F}}}_{\ell_4},$$

where $T_3$, $T_5$, $T_6$, $T_9$ and $T_{10}$ are defined in the proof of Proposition 3.7 in Appendix D.1. We provide upper bounds for $\|T_3\|_{\mathrm{F}}$, $\|T_5\|_{\mathrm{F}}$, and $\|T_9\|_{\mathrm{F}}$, and the bounds for $\|T_6\|_{\mathrm{F}}$ and $\|T_{10}\|_{\mathrm{F}}$ follows immediately by replacing $t$ with $t-1$. By 1-Lipschitzness of ReLU activation $\sigma(\cdot)$,

$$\|T_3\|_{\mathrm{F}} \le \|\mathbf{P}_t\|_{\mathrm{F}} \left\|\sigma(\mathbf{W}_{t+1}^\top \mathbf{X}) - \sigma(\mathbf{W}_t^\top \mathbf{X})\right\|_{\mathrm{F}} \le \|\mathbf{P}_t\|_{\mathrm{F}} \left\|\mathbf{Q}_t^\top \mathbf{X}\right\|_{\mathrm{F}}.$$

For $T_5$, we have $\|T_5\|_{\mathrm{F}} \le \sqrt{\overline{S}} \|\mathbf{V}_t\|_{\mathrm{F}} \|\mathbf{X}\|_{\mathrm{F}} \max_{1 \le i \le 2m} \|[\mathbf{Q}_t]_{:,i}\|$ due to the followings:

$$\|T_5\|_{\mathrm{F}}^2 \le \|\mathbf{V}_t\|^2 \sum_{j=1}^N \sum_{i \in S_j^\perp} \left(\sigma([\mathbf{W}_{t+1}]_{:,i}^\top \mathbf{x}_j) - \sigma([\mathbf{W}_t]_{:,i}^\top \mathbf{x}_j)\right)^2$$

$$\le \|\mathbf{V}_t\|^2 \sum_{j=1}^N \sum_{i \in S_j^\perp} \left([\mathbf{Q}_t]_{:,i}^\top \mathbf{x}_j\right)^2$$

$$\le \|\mathbf{V}_t\|^2 \overline{S} \max_{1 \le i \le 2m} \|[\mathbf{Q}_t]_{:,i}\|^2 \sum_{j=1}^N \|\mathbf{x}_j\|^2$$

$$= \|\mathbf{V}_t\|^2 \overline{S} \max_{1 \le i \le 2m} \|[\mathbf{Q}_t]_{:,i}\|^2 \|\mathbf{X}\|_{\mathrm{F}}^2,$$

where the second line uses 1-Lipschitzness of ReLU, and the third line is by Cauchy–Schwarz inequality. For $T_9$, we have

$$\|T_9\|_{\mathrm{F}}^2 = \sum_{l=1}^n \sum_{j=1}^N \left(\sum_{i \in S_j^\perp} [\mathbf{A}_{i,t} \mathbf{R}_t \mathbf{B}_{i,t}]_{l,j}\right)^2$$

$$\le \sum_{l=1}^n \sum_{j=1}^N |S_j^\perp| \sum_{i \in S_j^\perp} [\mathbf{A}_{i,t} \mathbf{R}_t \mathbf{B}_{i,t}]_{l,j}^2$$

$$= \sum_{j=1}^N |S_j^\perp| \sum_{i \in S_j^\perp} \|[\mathbf{A}_{i,t} \mathbf{R}_t \mathbf{B}_{i,t}]_{:,j}\|^2$$

$$\le \sum_{j=1}^N |S_j^\perp| \sum_{i \in S_j^\perp} \|\mathbf{A}_{i,t}\|^2 \|\mathbf{R}_t\|^2 \|[\mathbf{B}_{i,t}]_{:,j}\|^2$$

$$\le \|\mathbf{R}_t\|^2 \max_{1 \le i \le 2m} \|\mathbf{A}_{i,t}\|^2 \sum_{j=1}^N \sum_{k=1}^N |S_j^\perp| \sum_{i \in S_j^\perp} [\mathbf{B}_{i,t}]_{k,j}^2$$

$$\le \|\mathbf{R}_t\|^2 \max_{1 \le i \le 2m} \|\mathbf{A}_{i,t}\|^2 \overline{S}^2 N^2,$$

where the second line is by convexity of quadratic function, and the fifth line is by $|[\mathbf{B}_{i,t}]_{k,j}| \le 1$.

We now bound terms $\ell_1$, $\ell_2$, $\ell_3$ and $\ell_4$. For $\ell_1$, we have
$$\ell_1 = \beta\eta \left\|\mathbf{R}_{t-1}\sigma(\mathbf{X}^\top \mathbf{W}_{t-1})(\sigma(\mathbf{W}_t^\top \mathbf{X}) - \sigma(\mathbf{W}_{t-1}^\top \mathbf{X}))\right\|_{\mathrm{F}}$$
$$\le \beta\eta \|\mathbf{R}_{t-1}\|_{\mathrm{F}} \left\|\sigma(\mathbf{X}^\top \mathbf{W}_{t-1})\right\|_{\mathrm{F}} \left\|\sigma(\mathbf{W}_t^\top \mathbf{X}) - \sigma(\mathbf{W}_{t-1}^\top \mathbf{X})\right\|_{\mathrm{F}}$$
$$\le \beta\eta \|\mathbf{R}_{t-1}\|_{\mathrm{F}} \left\|\mathbf{X}^\top \mathbf{W}_{t-1}\right\|_{\mathrm{F}} \left\|\mathbf{Q}_{t-1}^\top \mathbf{X}\right\|_{\mathrm{F}}$$
$$\le \frac{2\theta^{2t}}{1-\theta}\eta^2 C_1^2 m C_V C_W \|\mathbf{X}\|_{\mathrm{F}} \|\mathbf{R}_0\|_{\mathrm{F}}^2,$$

where the second inequality uses 1-Lipschitzness of ReLU. For $\ell_2$, we have
$$\ell_2 = \beta\eta \left\|\mathbf{P}_{t-1}\left((((\mathbf{R}_{t-1}^\top \mathbf{V}_{t-1}) \odot \mathbb{1}[\mathbf{X}^\top \mathbf{W}_{t-1} \ge 0])^\top \mathbf{X}^\top \mathbf{X}) \odot \mathbb{1}[\mathbf{W}_{t-1}^\top \mathbf{X} \ge 0] \odot \mathbf{M}\right)\right\|_{\mathrm{F}}$$
$$\le \beta\eta \|\mathbf{P}_{t-1}\|_{\mathrm{F}} \|\mathbf{R}_{t-1}\|_{\mathrm{F}} \|\mathbf{V}_{t-1}\|_{\mathrm{F}} \|\mathbf{X}^\top \mathbf{X}\|$$
$$\le \frac{2\theta^{2t}}{1-\theta}\eta^2 C_1^2 m C_V C_W \|\mathbf{X}\|_{\mathrm{F}} \|\mathbf{R}_0\|_{\mathrm{F}}^2.$$

For $\ell_3$, we have

$$\ell_3 \leq \|T_5\|_{\mathrm{F}} + \beta \|T_6\|_{\mathrm{F}} + (1+\beta)\eta \|T_9\|_{\mathrm{F}} + \beta\eta \|T_{10}\|_{\mathrm{F}}$$

$$\leq \|T_5\|_{\mathrm{F}} + \theta^2 \|T_6\|_{\mathrm{F}} + \eta \bar{S} N (R_V + R_1)^2 C_1 \left(1 + \theta + \theta^2\right) \theta^t \|\mathbf{R}_0\|_{\mathrm{F}}$$

$$\leq \frac{2\theta^t}{1-\theta}\eta C_1 \sqrt{\bar{S}} C_V^2 \|\mathbf{R}_0\|_{\mathrm{F}} + \eta \bar{S} N (R_V + R_1)^2 C_1 \left(1 + \theta + \theta^2\right) \theta^t \|\mathbf{R}_0\|_{\mathrm{F}}$$

where we use the bound for $T_5$, $T_9$ and $T_{10}$ in Proposition 3.7. For $\ell_4$, we have

$$\ell_4 = \left\| \beta \mathbf{P}_{t-1} \left( (\mathbf{Q}_{t-1}^\top \mathbf{X}) \odot \mathbb{1}[\mathbf{W}_0^\top \mathbf{X} \geq 0] \odot \mathbf{M} \right) + \mathbf{P}_t \left( \sigma(\mathbf{W}_{t+1}^\top \mathbf{X}) - \sigma(\mathbf{W}_t^\top \mathbf{X}) \right) \right\|_{\mathrm{F}}$$

$$\leq \beta \|\mathbf{P}_{t-1}\|_{\mathrm{F}} \left\| \mathbf{Q}_{t-1}^\top \mathbf{X} \right\|_{\mathrm{F}} + \|\mathbf{P}_t\|_{\mathrm{F}} \left\| \mathbf{Q}_t^\top \mathbf{X} \right\|_{\mathrm{F}}$$

$$\leq \frac{4\theta^{2t}}{(1-\theta)^2}\eta^2 C_1^2 m C_V C_W \|\mathbf{X}\|_{\mathrm{F}} \|\mathbf{R}_0\|_{\mathrm{F}}^2 \,.$$

Combine the results, we get

$$\|\boldsymbol{\zeta}_t\| \leq \frac{\theta^{2t}(8-4\theta)}{(1-\theta)^2}\eta^2 C_1^2 m C_V C_W \|\mathbf{X}\|_{\mathrm{F}} \|\mathbf{R}_0\|_{\mathrm{F}}^2$$

$$+ \frac{2\theta^t}{1-\theta}\eta C_1 \sqrt{\bar{S}} C_V^2 \|\mathbf{R}_0\|_{\mathrm{F}} + \eta \bar{S} N (R_V + R_1)^2 C_1 \left(1 + \theta + \theta^2\right) \theta^t \|\mathbf{R}_0\|_{\mathrm{F}} \,.$$

Now we turn to $\|\boldsymbol{\iota}_t\|$. By Lemma 3.9, we have

$$\|\mathbf{K}_0 - \mathbf{K}_t\| \leq 10 d N \frac{R_2}{c_2} \|[\mathbf{V}_0]_{:,i}\|^2 \,, \quad \|\mathbf{G}_t\| \leq 2 m C_W^2 \,.$$

Therefore, we get

$$\|\boldsymbol{\iota}_t\| \leq (1+\beta)\eta \|(\mathbf{K}_0 - \mathbf{H}_t)\mathbf{r}_t\| + \beta\eta \|(\mathbf{K}_0 - \mathbf{H}_{t-1})\mathbf{r}_{t-1}\|$$

$$\leq (1+\beta)\eta \left( \|\mathbf{K}_0 - \mathbf{K}_t\| + \|\mathbf{G}_t\| \right) \|\mathbf{R}_t\|_{\mathrm{F}} + \beta\eta \left( \|\mathbf{K}_0 - \mathbf{K}_{t-1}\| + \|\mathbf{G}_{t-1}\| \right) \|\mathbf{R}_{t-1}\|_{\mathrm{F}}$$

$$\leq 6 \left( 5N\sqrt{\frac{R_2}{c_2}} R_V^2 + C_W^2 \right) m\eta C_1 \theta^t \|\mathbf{R}_0\|_{\mathrm{F}} \,.$$

Combining the results for $\|\boldsymbol{\zeta}_t\|$ and $\|\boldsymbol{\iota}_t\|$ with $\theta < 1$ we get the final result. $\qquad\square$

### G.4 PROOF OF LEMMA 3.12

*Proof of Lemma 3.12.* By Lemma 3.11, $\|[\mathbf{P}_s]_{:,i}\| \leq \frac{\theta^s}{1-\theta}\eta C_1 C_W \|\mathbf{Y}\|_{\mathrm{F}}$. When $\eta C_1 C_W \|\mathbf{Y}\|_{\mathrm{F}} \leq (1-\theta)^2 R_1$, for any $i \in [2m]$ we have

$$\|[\mathbf{V}_{t+1}]_{:,i} - [\mathbf{V}_0]_{:,i}\| \leq \sum_{s=0}^{t} \|[\mathbf{P}_s]_{:,i}\| \leq \sum_{s=0}^{t} \frac{\theta^s}{1-\theta}\eta C_1 \|\mathbf{Y}\|_{\mathrm{F}} C_W \leq \frac{1}{(1-\theta)^2}\eta C_1 \|\mathbf{Y}\|_{\mathrm{F}} C_W \leq R_1.$$

Similarly, when $\eta C_1 C_V \|\mathbf{Y}\|_{\mathrm{F}} \leq (1-\theta)^2 R_2$ we have $\|[\mathbf{W}_{t+1}]_{:,i} - [\mathbf{W}_0]_{:,i}\| \leq R_2$, $\forall i \in [2m]$. $\quad\square$

### G.5 PROOF OF THEOREM 3.1

*Proof of Theorem 3.1.* Let $a_t = \|\mathbf{r}_t\|$, $b_t = \|\boldsymbol{\xi}_t\|$, $\rho = 1 - \frac{1}{\kappa(\mathbf{K}_0)}$, $\theta = 1 - \frac{1}{2\kappa(\mathbf{K}_0)}$, and define $c_0$ as

$$c_0 = C_1 \|\mathbf{Y}\|_{\mathrm{F}} \left[ \frac{4}{L}\sqrt{\frac{8mR_2}{c_2}} C_V^2 + \frac{24mR_2}{c_2 L}N(R_V + R_1)^2 \right.$$

$$+ \frac{32m}{L^2}C_1 C_V C_W \|\mathbf{X}\|_{\mathrm{F}} \|\mathbf{Y}\|_{\mathrm{F}}$$

$$\left. + \frac{6m}{L}\left( 5N\frac{R_2}{c_2}R_V^2 + C_W^2 \right) \right]. \tag{16}$$

By Lemma 3.8, $a_{t+1} \leq \rho \cdot a_t + b_t$ for any $t \geq 0$. Take $C_1 = 2$ and $R_1 = \frac{R_2}{3c_2} R_V$, ensuring $R_1 < R_V$ and $R_V^2 \cdot \frac{R_2}{c_2} \geq R_1(2R_V + R_1)$. Let

$$m \geq \max\left\{\widetilde{\Omega}\left(\frac{Nn}{\lambda}\right), \widetilde{\Omega}\left(\frac{c_2}{R_2}\right), \Omega\left(\frac{1}{\delta}\right)\right\}.$$

Then conditions in Lemmas 3.5, 3.6, 3.9 and 3.10 hold at $t = 0$ with probability at least $1 - \delta$, and

$$\tfrac{1}{2}c_1\sqrt{n} \leq R_V \leq c_1(\sqrt{n} + 2\sqrt{\log m}), \tag{17}$$

$$\tfrac{1}{2}c_2\sqrt{d} \leq R_W \leq c_2(\sqrt{d} + 2\sqrt{\log m}), \tag{18}$$

$$\tfrac{1}{2}c_1^2 m\lambda \leq \mu \leq L \leq 9c_1^2 nm\lambda_1(\overline{\mathbf{K}}). \tag{19}$$

Suppose these conditions hold until step $t$. In GD, $0 = \beta < \theta^2 < \theta$, thus by Lemma 3.11, $b_t \leq c_0 \cdot \theta^t$, and

$$\|\mathbf{r}_{t+1}\| \leq a_{t+1} \leq \left(a_0 + \frac{c_0}{\theta - \rho}\right)\theta^{t+1}.$$

For the induction to hold at step $t + 1$, we need:

$$a_0 + \frac{c_0}{\theta - \rho} \leq C_1 \|\mathbf{Y}\|_{\mathrm{F}}, \tag{20}$$

$$4LC_1 C_W \|\mathbf{Y}\|_{\mathrm{F}} \leq \mu^2 R_1, \tag{21}$$

$$4LC_1 C_V \|\mathbf{Y}\|_{\mathrm{F}} \leq \mu^2 R_2. \tag{22}$$

By plugging in the initial conditions (17) to (19), we get the following sufficient conditions for Equations (20) to (22):

$$\sqrt{\frac{R_2}{mc_2}} \frac{N}{\lambda\kappa^2(\mathbf{K}_0)}(n + \log m) \lesssim 1, \tag{23}$$

$$\frac{R_2}{c_2} \frac{N}{\lambda\kappa^2(\mathbf{K}_0)}(n + \log m) \lesssim 1, \tag{24}$$

$$\frac{c_2 N^2}{c_1^3 m\lambda_N^2(\overline{\mathbf{K}})\kappa^3(\mathbf{K}_0)}(\sqrt{nd} + \log m) \lesssim 1, \tag{25}$$

$$\frac{R_2}{c_2} \frac{c_2^2 N}{c_1^2 \lambda\kappa^2(\mathbf{K}_0)}(d + \log m) \lesssim 1, \tag{26}$$

$$c_2(\sqrt{d} + \sqrt{\log m})\sqrt{N}\kappa(\mathbf{K}_0) \lesssim c_1^3 m\lambda \frac{R_2}{c_2}\sqrt{n}, \tag{27}$$

$$(\sqrt{n} + \sqrt{\log m})\sqrt{N}\kappa(\mathbf{K}_0) \lesssim c_1 c_2 m\lambda \frac{R_2}{c_2}, \tag{28}$$

where Equations (23) to (26) are sufficient for (20) by making each monomial in the square bracket in (16) to be less than $\frac{1}{20}$, (27) is sufficient for (21), and (28) is sufficient for (22). These conditions can be satisfied by the following specification of $c_1$, $c_2$ and $R_2$:

$$\frac{c_2}{R_2} \geq \widetilde{\Omega}\left(\frac{Nn}{\lambda}\right), \ \frac{c_1^3}{c_2} \geq \widetilde{\Omega}\left(\frac{\sqrt{Nd}(\sqrt{N} \vee \kappa^4(\mathbf{K}_0))}{\sqrt{n}\lambda\kappa^3(\mathbf{K}_0)}\right),$$

$$\frac{c_1^2}{c_2^2} \geq \widetilde{\Omega}\left(\frac{d}{n\kappa^2(\mathbf{K}_0)}\right), \ c_1 c_2 \geq \widetilde{\Omega}\left(\frac{\sqrt{Nn}\kappa(\mathbf{K}_0)}{\lambda}\right).$$

The convergence result follows by induction, with rate $\theta \leq 1 - \frac{\Theta(1)}{n\kappa}$ by Lemma 3.6. $\square$

G.6 PROOF OF THEOREM 3.2

*Proof of Theorem 3.2.* We prove the theorem by induction Lemma G.1, where we set $a_t = \|(\mathbf{r}_t^\top, \mathbf{r}_{t-1}^\top)^\top\|$, $b_t = \|\boldsymbol{\xi}_t\|$, $\rho = 1 - \frac{1}{\sqrt{\kappa(\mathbf{K}_0)}}$, $\theta = 1 - \frac{1}{2\sqrt{\kappa(\mathbf{K}_0)}}$, and

$$c_0 = C_1 \|\mathbf{Y}\|_{\mathrm{F}} \left[4\sqrt{\frac{8mR_2}{c_2\mu L}}C_V^2 + \frac{24mR_2}{c_2 L}N(R_V + R_1)^2 + \frac{32m}{\mu L}C_1 C_V C_W \|\mathbf{X}\|_{\mathrm{F}}\|\mathbf{Y}\|_{\mathrm{F}} + \frac{6m}{L}\left(5N\frac{R_2}{c_2}R_V^2 + C_W^2\right)\right]. \tag{29}$$

By Lemma 3.8, $a_{t+1} \leq \rho \cdot a_t + b_t$ for any $t \geq 0$. Our choice of $\eta = \frac{1}{L}$ and $\beta = \frac{\sqrt{L} - \sqrt{\mu}}{\sqrt{L} + \sqrt{\mu}}$ satisfies $\beta < \theta^2 < \theta < 1$. Take $C_1 = 2\sqrt{2}$, then at the 0-th step, $\|\mathbf{r}_0\| \leq a_0 = \sqrt{2} \|\mathbf{Y}\|_F \leq C_1 \|\mathbf{Y}\|_F$. Take $R_1 = \frac{R_2}{3c_2} R_V$, then $R_1 < R_V$ and $R_V^2 \cdot \frac{R_2}{c_2} \geq R_1(2R_V + R_1)$. Let

$$
m \geq \max \left\{ \widetilde{\Omega} \left( \frac{Nn}{\lambda} \right), \widetilde{\Omega} \left( \frac{c_2}{R_2} \right), \Omega \left( \frac{1}{\delta} \right) \right\},
$$

then the conditions in Lemmas 3.5, 3.6, 3.9 and 3.10 hold for the 0-th step, and $R_2 > 0$ can be arbitrarily small as the initial weight change is 0. In particular, the initialization yields

$$
\frac{c_1 \sqrt{n}}{2} \leq R_V \leq c_1(\sqrt{n} + 2\sqrt{\log m}),
$$

$$
\frac{c_2 \sqrt{d}}{2} \leq R_W \leq c_2(\sqrt{d} + 2\sqrt{\log m}),
$$

$$
L \leq 9c_1^2 nm\lambda_1(\overline{\mathbf{K}}),
$$

$$
\mu \geq \frac{1}{2}c_1^2 m\lambda.
$$

Now suppose these conditions hold from step 0 to $t$, then with probability at least $1 - \delta$, all results in Lemmas 3.5, 3.6, 3.9 and 3.10 hold until the $t$-th step. By Lemma 3.11, $b_t \leq c_0 \cdot \theta^t$, and thus

$$
\|\mathbf{r}_{t+1}\| \leq a_{t+1} \leq \left( a_0 + \frac{c_0}{\theta - \rho} \right) \theta^{t+1}
$$

by Lemma G.1. To show the induction conditions in Lemmas 3.5, 3.6, 3.9 and 3.10 for step $t + 1$, it remains to show the followings:

$$
a_0 + \frac{c_0}{\theta - \rho} \leq C_1 \|\mathbf{Y}\|_F, \tag{30}
$$

$$
4C_1 C_W \|\mathbf{Y}\|_F \leq \mu R_1, \tag{31}
$$

$$
4C_1 C_V \|\mathbf{Y}\|_F \leq \mu R_2, \tag{32}
$$

where the first condition corresponds to residual convergence, and the last two conditions correspond to bounds on the weight changes. By plugging in the initial conditions, we get the following sufficient conditions for Equations (30) to (32):

$$
\sqrt{\frac{R_2}{mc_2}} \frac{N}{\lambda}(n + \log m) \lesssim 1, \tag{33}
$$

$$
\frac{R_2}{c_2} \frac{N}{\lambda\sqrt{\kappa(\mathbf{K}_0)}}(n + \log m) \lesssim 1, \tag{34}
$$

$$
\frac{c_2 N^2}{c_1^3 m\lambda_N^2(\overline{\mathbf{K}})\sqrt{\kappa(\mathbf{K}_0)}}(\sqrt{nd} + \log m) \lesssim 1, \tag{35}
$$

$$
\frac{R_2}{c_2} \frac{c_2^2 N}{c_1^2 \lambda\sqrt{\kappa(\mathbf{K}_0)}}(d + \log m) \lesssim 1, \tag{36}
$$

$$
c_2(\sqrt{d} + \sqrt{\log m})\sqrt{N} \lesssim c_1^3 m\lambda \frac{R_2}{c_2}\sqrt{n}, \tag{37}
$$

$$
(\sqrt{n} + \sqrt{\log m})\sqrt{N} \lesssim c_1 c_2 m\lambda \frac{R_2}{c_2}, \tag{38}
$$

where Equations (33) to (36) are sufficient for (30) by making each summand in the square bracket in (29) to be less than $\frac{1}{20}$, Equation (37) is sufficient for (31), and Equation (38) is sufficient for (32).

By additionally choosing $c_1$ and $c_2$ so that

$$\frac{c_2}{R_2} \geq \widetilde{\Omega}\left(\frac{Nn}{\lambda}\right) > 1,$$

$$\frac{c_1^3}{c_2} \geq \widetilde{\Omega}\left(\frac{N\sqrt{d}}{\sqrt{n}\lambda}\right),$$

$$\frac{c_1^2}{c_2^2} \geq \widetilde{\Omega}\left(\frac{d}{n\sqrt{\kappa(\mathbf{K}_0)}}\right),$$

$$c_1 c_2 \geq \widetilde{\Omega}\left(\frac{\sqrt{Nn}}{\lambda}\right),$$

Equations (33) to (38) are satisfied and the induction conditions for the $t+1$ step hold. The final convergence result follows immediately from induction, where the convergence speed is $\theta$. By Lemma 3.6, $\theta = 1 - \frac{1}{2\sqrt{\kappa(\mathbf{K}_0)}} \leq 1 - \frac{\Theta(1)}{\sqrt{n\kappa}}$. □

### G.7 PROOF OF THEOREM 4.1

*Proof of Theorem 4.1.* We prove the theorem by induction Lemma G.1, where we set $a_t = \left\|(\mathbf{r}_t^\top, \mathbf{r}_{t-1}^\top)^\top\right\|$, $b_t = \|\boldsymbol{\xi}_t\|$, $\rho = 1 - \frac{1}{\sqrt{\kappa(\mathbf{K}_0)}}$, $\theta = 1 - \frac{1}{2\sqrt{\kappa(\mathbf{K}_0)}}$, and

$$c_0 = C_1 \|\mathbf{Y}\|_{\mathrm{F}} \left[4\sqrt{\frac{8mR_2}{c_2\mu L}}C_V^2 + \frac{24mR_2}{c_2 L}N(R_V + R_1)^2 + \frac{32m}{\mu L}C_1 C_V C_W \|\mathbf{X}\|_{\mathrm{F}} \|\mathbf{Y}\|_{\mathrm{F}} + \frac{6m}{L}\left(5N\frac{R_2}{c_2}R_V^2 + C_W^2\right)\right]. \tag{39}$$

By Lemma 4.6, $a_{t+1} \leq \rho \cdot a_t + b_t$ for any $t \geq 0$. Our choice of $\eta = \frac{1}{L}$ and $\beta = \frac{\sqrt{L}-\sqrt{\mu}}{\sqrt{L}+\sqrt{\mu}}$ satisfies $\beta < \theta^2 < \theta < 1$. Take $C_1 = 2\sqrt{2}$, then at the 0-th step, $\|\mathbf{r}_0\| \leq a_0 = \sqrt{2}\|\mathbf{Y}\|_{\mathrm{F}} \leq C_1 \|\mathbf{Y}\|_{\mathrm{F}}$. Take $R_1 = \frac{R_2}{3c_2}R_V$, then $R_1 < R_V$ and $R_V^2 \cdot \frac{R_2}{c_2} \geq R_1(2R_V + R_1)$. Let

$$m \geq \max\left\{\widetilde{\Omega}\left(\frac{Nr\kappa(\mathbf{Y}\mathbf{Y}^\top)}{\lambda}\right), \widetilde{\Omega}\left(\frac{c_2}{R_2}\right), \Omega\left(\frac{1}{\delta}\right)\right\},$$

then the conditions in Lemmas 3.10, 4.2, 4.3 and E.2 hold for the 0-th step, and $R_2 > 0$ can be arbitrarily small as the initial weight change is 0. In particular, the initialization yields

$$\frac{c_1 \sigma_r(\mathbf{Y})\sqrt{r}}{2} \leq R_V \leq c_1 \|\mathbf{Y}\|(\sqrt{r} + 2\sqrt{\log m}),$$

$$\frac{c_2\sqrt{d}}{2} \leq R_W \leq c_2(\sqrt{d} + 2\sqrt{\log m}),$$

$$\frac{1}{2}c_1^2 m\lambda\sigma_r^2(\mathbf{Y}) \leq \mu \leq L \leq 9c_1^2 mr\lambda_1(\overline{\mathbf{K}})\|\mathbf{Y}\|^2.$$

Now suppose these conditions hold from step 0 to $t$, then with probability at least $1 - \delta$, all results in Lemmas 3.10, 4.2, 4.3 and E.2 hold until the $t$-th step. By Lemma 3.11, $b_t \leq c_0 \cdot \theta^t$, and thus

$$\|\mathbf{r}_{t+1}\| \leq a_{t+1} \leq \left(a_0 + \frac{c_0}{\theta - \rho}\right)\theta^{t+1}$$

by Lemma G.1. To show the induction conditions in Lemmas 3.10, 4.2, 4.3 and E.2 for step $t+1$, it remains to show the followings:

$$a_0 + \frac{c_0}{\theta - \rho} \leq C_1 \|\mathbf{Y}\|_{\mathrm{F}}, \tag{40}$$

$$4C_1 C_W \|\mathbf{Y}\|_{\mathrm{F}} \leq \mu R_1, \tag{41}$$

$$4C_1 C_V \|\mathbf{Y}\|_{\mathrm{F}} \leq \mu R_2, \tag{42}$$

where the first condition corresponds to residual convergence, and the last two conditions correspond to bounds on the weight changes. By plugging in the initial conditions, we get the following sufficient conditions for Equations (40) to (42):

$$\sqrt{\frac{R_2}{mc_2}}\frac{N\kappa(\mathbf{Y}\mathbf{Y}^\top)}{\lambda}(r+\log m) \lesssim 1, \tag{43}$$

$$\frac{R_2}{c_2}\frac{N\kappa(\mathbf{Y}\mathbf{Y}^\top)}{\lambda\sqrt{\kappa(\mathbf{K}_0)}}(r+\log m) \lesssim 1, \tag{44}$$

$$\frac{c_2 N^2}{c_1^3 m\lambda_N^2(\overline{\mathbf{K}})\lambda_r^2(\mathbf{Y}\mathbf{Y}^\top)\sqrt{\kappa(\mathbf{K}_0)}}(\sqrt{dr}+\log m)\|\mathbf{Y}\| \lesssim 1, \tag{45}$$

$$\frac{R_2}{c_2}\frac{c_2^2 N}{c_1^2\lambda\lambda_r(\mathbf{Y}\mathbf{Y}^\top)\sqrt{\kappa(\mathbf{K}_0)}}(d+\log m) \lesssim 1, \tag{46}$$

$$c_2(\sqrt{d}+\sqrt{\log m})\sqrt{N} \lesssim c_1^3 m\lambda\lambda_r(\mathbf{Y}\mathbf{Y}^\top)\frac{R_2}{c_2}\sqrt{r}\sigma_r(\mathbf{Y}), \tag{47}$$

$$(\sqrt{r}+\sqrt{\log m})\sqrt{N}\|\mathbf{Y}\| \lesssim c_1 c_2 m\lambda\lambda_r(\mathbf{Y}\mathbf{Y}^\top)\frac{R_2}{c_2}, \tag{48}$$

where Equations (43) to (46) are sufficient for (40) by making each summand in the square bracket in (39) to be less than $\frac{1}{20}$, Equation (47) is sufficient for (41), and Equation (48) is sufficient for (42).

By additionally choosing $c_1$ and $c_2$ so that

$$\frac{c_2}{R_2} \geq \widetilde{\Omega}\left(\frac{Nr\kappa(\mathbf{Y}\mathbf{Y}^\top)}{\lambda}\right) > 1,$$

$$\frac{c_1^3}{c_2} \geq \widetilde{\Omega}\left(\frac{\sqrt{Nd}(\kappa(\mathbf{Y})\vee\sqrt{N})}{\sqrt{r}\lambda\lambda_r(\mathbf{Y}\mathbf{Y}^\top)\|\mathbf{Y}\|}\right),$$

$$\frac{c_1^2}{c_2^2} \geq \widetilde{\Omega}\left(\frac{d}{r\|\mathbf{Y}\|^2}\right),$$

$$c_1 c_2 \geq \widetilde{\Omega}\left(\frac{\sqrt{Nr}\kappa(\mathbf{Y})}{\lambda\lambda_r(\mathbf{Y})}\right),$$

Equations (43) to (48) are satisfied and the induction conditions for the $t+1$ step hold. The final convergence result follows immediately from induction, where the convergence speed is $\theta$. By Lemma 4.3, $\theta = 1 - \frac{1}{2\sqrt{\kappa(\mathbf{K}_0)}} \leq 1 - \frac{\Theta(1)}{\sqrt{r\kappa(\mathbf{Y})}}$. $\qquad\square$

# H LLM USAGE

In preparing this paper, large language models (LLMs) such as ChatGPT were used only for light editing purposes, including minor grammar checking and sentence polishing. No part of the research ideation, methodology design, experimental execution, or analysis was conducted with the assistance of LLMs.

