# OpenReview forum: "Convergence of Near-Linear Width ReLU Networks with Unbalanced Initialization"
_ICLR.cc/2026/Conference — Submitted to ICLR 2026_

### Official Review · Reviewer_CSXy · 2025-10-31

**Soundness:** 2
**Presentation:** 3
**Contribution:** 4
**Rating:** 4
**Confidence:** 4

**Summary:**

This paper proves a conjecture that has been existed for a long time, which states that probably there need only $\Omega(N)$ neurons exist to prove that neural networks are trainable, where $N$ is the number of data. More specifically, they discuss the vector-output setting, and show that $\Omega(Nn/\lambda)$ neurons are enough to guarantee that gradient descent converges to a global optimum, where $N$ is the number of data, $n$ is the dimension of the output, and $\lambda$ is the minimal eigenvalue of the NTK. The proof is quite standard, where they bound the eigenvalue of the NTK and show that along training, the NTK does not change that much. The main innovation comes from not relying on the Frobenius norm bound like the other papers did, but using matrix concentration inequalities directly. Another innovation is assymetric initialization. The authors extend their result to accelerated gradients and obtain an acceleration result, and gave a better lower bound when the $Y$ is low rank by sketching.

**Strengths:**

One apparent strength is, if the proof is correct, it solves a conjecture that was yet remained to be solved (of which I have doubts - see weaknesses).

Lemma 3.6 seems correct and identifies an important inefficiency in proofs of other papers. Other papers have had similar ideas, e.g. see Theorem B.4. in [1].

At last, the paper is nice to read because it solves some additional natural questions that one might have in these line of results - e.g. can the proof be extended to accelerate gradients? Can it be extended to low-rank structure (because it is intuitive that for low rank responses we won't need that much neurons). The sketching idea is very interesting too.

[1] Kim, Sungyoon, and Mert Pilanci. "Convex relaxations of relu neural networks approximate global optima in polynomial time." arXiv preprint arXiv:2402.03625 (2024).

**Weaknesses:**

The biggest suspicion that I have is about Lemma 3.10. I have read [2] and [3] and I believe both papers use Lemma 3.10, provided that **they are only training the first-layer weight**. In this paper the authors discuss the problem where they are training both first and second later weights. So I don't understand how Lemma 3.10. can be directly applied. Maybe it could be applied, or maybe it holds - at least some clarification is needed. Also, the form is not identical to the statement in [2]; there is no statement about weight bounds and there is only one parameter $R_1$ in [2]. So I think it would be good if the authors derive the Lemma again.

Some minor comments:

- In Table 1, certain results use random initialization (e.g. [4]), while others (especially this paper) uses a different initialization. I think it would be good if clarified.
- I think it would be good to add results that discuss loss landscapes in similar regime, e.g. [5]. [5] shows that if $m > N$ for regularized neural networks, all local optima are global. Does this imply that gradient descent will find a global optimum always? Why or why not?
- [6] also has coupled initialization: how is this different from the initialization in this paper? I think discussion is needed, because they seem very similar to me.

[2] Jun-Kun Wang, Chi-Heng Lin, and Jacob D Abernethy. A modular analysis of provable acceleration via polyak’s momentum: Training a wide relu network and a deep linear network. In International Conference on Machine Learning, pp. 10816–10827. PMLR, 2021.

[3] Song, Z. and Yang, X. Quadratic suffices for over-parametrization via matrix chernoff bound

[4] Du, Simon S., et al. "Gradient descent provably optimizes over-parameterized neural networks." arXiv preprint arXiv:1810.02054 (2018).

[5] Haeffele, Benjamin D., and René Vidal. "Global optimality in tensor factorization, deep learning, and beyond." arXiv preprint arXiv:1506.07540 (2015).

[6] Munteanu, Alexander, et al. "Bounding the width of neural networks via coupled initialization a worst case analysis." International Conference on Machine Learning. PMLR, 2022.

**Questions:**

How feasible would it be to extend the result to random initialization?

---

> ### Author Response · Authors · 2025-11-21
>
> We thank the reviewer for acknowledging our theoretical contributions and providing valuable comments and references like [1]. We will add [1] in related works. We address the main concern about Lemma 3.10 and other comments below.
>
> > **W1: Deriving Lemma 3.10.**
>
> Thank you for the opportunity to clarify. Lemma 3.10 is a **geometric property** of ReLU features under perturbation, derived solely from the randomness of the initialization. It is independent of the specific optimization trajectory or whether one or both layers are trained.
>
> The lemma states: if a weight matrix $W$ stays within a radius $R_2$ of a Gaussian initialization $W_0$, the number of neurons that flip activation signs is bounded. Since our induction arguments (Lemmas 3.11 and 3.12) guarantee that our trained weights $W_t$ stay within this radius throughout training, the lemma applies there directly.
>
> We will provide a complete proof in the uploaded revision to resolve the ambiguity. Here we provide a sketch.
>
> >> **Proof Sketch:**
> >> Let $S_j^\perp$ be the set of neurons that flip activation on input $x_j$.
> >> 1. **Geometric Condition:** A neuron $i$ flips its sign only if the pre-activation crosses zero. Given the condition that $\Vert [W_t]\_{:,i} - [W_0]\_{:,i}\Vert \leq R_2$, a necessary condition for a flip is that the initial pre-activation is small:
> >> $$|[W_0]\_{:,i}^\top x_j| \le \max\_{w\in\mathbb{R}^d,\Vert w- [W_0]\_{:,i}\Vert \leq R_2}|(w- [W_0]\_{:,i})^\top x_j| \le R_2 \Vert  x_j\Vert = R_2.$$
> >> 2. **Probability:** At initialization, the weights are Gaussian $[W_0]_{:,i} \sim \mathcal{N}(0, c_2^2 I)$. Thus, the pre-activation $z \sim \mathcal{N}(0, c_2^2)$. The probability of the event $|z| \le R_2$ is bounded by the Gaussian anti-concentration inequality:
> >> $$\mathbb{P}(|z| \le R_2) \le \frac{2R_2}{\sqrt{2\pi}c_2} < \frac{R_2}{c_2}.$$
> >> 3. **Concentration:** Let $s_j$ be the count of neurons satisfying this geometric condition. Since the first $m$ neurons are initialized independently and the last $m$ neurons share the same magnitudes as the first half, $s_j$ is a sum of $m$ independent Bernoulli random variables, multiplied by a factor of $2$. Applying Bernstein's inequality, we bound the sum with high probability:
> >> $$\mathbb{P}\left(s_j > \frac{8mR_2}{c_2}\right) \le \exp\left(-\frac{mR_2}{c_2}\right).$$
> >> Since $|S_j^\perp| \le s_j$ (as we use necessary condition), the lemma follows from union bound.
>
> > **W2: Clarify initialization in Table 1.**
>
> Thanks for the suggestion. We will update Table 1 to explicitly state the initialization schemes (Random vs. Coupled/Unbalanced) for clearer comparison.
>
> > **W3: Loss landscapes.**
>
> Thanks for bringing up the reference [5], and we will add it regarding loss landscapes.
>
> > **W4: Initialization difference with [6].**
>
> Our initialization consists of coupling *and unbalanced scaling* (and an optional sketching step). The coupling part is the same as [6]. Our improved rate comes from the additional unbalanced scaling and finer analysis.
>
> > **Q1: Random initialization.**
>
> As mentioned to Reviewer ZAZi, extending this tight linear-width result to pure random initialization is extremely difficult because the initial residual scales with width $m$ and factors $c_1$ and $c_2$, breaking the feasibility of induction conditions (e.g., Eq. (23-28)) with near-linear width $m$. Coupled initialization cancels this out. To provide convergence guarantee for purely random initialization, one typically needs substantially larger width (often $O(N^4)$) to effectively "wash out" the initial noise relative to the signal.

---

> > ### Comment · Reviewer_CSXy · 2025-11-26
> >
> > Dear Authors,
> >
> > Thank you for your clear explanation on Lemma 3.10. I now understand that the lemma is not about training trajectory but is about how sign patterns change with small perturbations of the weights, and the proof strategy is valid as in the following induction phase you indeed prove that the weights do not move that much. I think this is an important contribution and the proof is correct, hence I raise my score.
> >
> > One final remark is that I couldn't give a higher score because it does improve over existing results which is nice, but the gain that I previously thought ($N^4 \rightarrow N$) is maybe not essentially true, because the results have different initialization schemes.

---

> > > ### Author Response · Authors · 2025-11-30
> > >
> > > Thank you very much for your positive reassessment and for raising the score to 6. We sincerely appreciate it and have updated the paper to reflect all the points discussed. The revised parts are marked in red.
> > >
> > > In particular, we have added an "Initialization" column in Table 1 for clarity. We have added a discussion about [1] and related works after Lemma 3.6. We have provided a complete proof of Theorem 3.10 in Appendix G.2.
> > >
> > > Regarding [5], we note that the solution of the regularized problem is generally different from the unregularized one, so their conclusion does not directly apply here. Regarding the differences in initialization schemes, we hope the new Table 1 provides a clearer picture of how initialization improves the width requirement, from pure random ($N^4$) to coupled ($N^2$) to our unbalanced initialization ($N$).
> > >
> > > Thank you again for the constructive feedback.

---

### Official Review · Reviewer_qQdC · 2025-10-31

**Soundness:** 3
**Presentation:** 3
**Contribution:** 2
**Rating:** 4
**Confidence:** 4

**Summary:**

This paper significantly narrows the theoretical gap for training two-layer ReLU networks by proving that a near-linear network width suffices for global convergence, moving closer to the long-standing conjecture that linear width is sufficient. The authors achieve this through a novel unbalanced Gaussian initialization scheme that biases training towards the Neural Tangent Kernel (NTK) regime, enabling tight control over kernel dynamics despite the non-smooth ReLU activation.

**Strengths:**

1.  The paper makes a contribution in closing the gap between theory and practice for ReLU networks. By proving convergence with a near-linear width of $\tilde \Omega (Nn/\lambda)$, it improves upon the best-known prior quadratic and polynomial bounds, addressing a long-standing conjecture in the field.

2. The introduction of the unbalanced initialization scheme biases the training dynamics into a regime where the kernel shift can be tightly controlled. This core idea enables the entire analysis and is a creative solution to the fundamental hurdle of non-smoothness in ReLU networks.

**Weaknesses:**

1. The near-linear width bound scales with $1/\lambda$, where $\lambda$ is the smallest eigenvalue of the limiting NTK.  In practice, $\lambda$ can be extremely small for real-world datasets, potentially making the required width large again and limiting the practical relevance of the theoretical guarantee.

2. While the analysis is a major step for two-layer ReLU networks, the modern deep learning landscape is dominated by deeper architectures and losses like cross-entropy. The paper does not provide a clear pathway for extending its novel techniques (unbalanced initialization, subspace analysis) to these more complex and practical settings, which limits the immediate significance of the contributions for broader applications.

**Questions:**

1. The width requirement's dependence on $1/\lambda$ is a significant bottleneck, as $\lambda$ can be exceedingly small. Do you have any empirical evidence or theoretical intuition suggesting that this dependence is unavoidable for your method, or is it an artifact of the analysis?

2. The paper's analysis is a landmark for two-layer networks. What do you perceive as the primary obstacles in extending your core techniques—specifically the unbalanced initialization and the tight control of kernel shift—to deep ReLU networks?

---

> ### Author Response · Authors · 2025-11-21
>
> We thank the reviewer for acknowledging our theoretical contributions and providing valuable comments. We address the concerns and questions regarding the dependence on $1/\lambda$ and extensions to more realistic settings below.
>
> > **W1 & Q1: Dependence on $1/\lambda$.**
>
> We clarify that the dependence on $\lambda$ (the smallest eigenvalue of the limiting NTK) is inherent to the NTK regime analysis. However, our work significantly **improves** this dependency compared to the state-of-the-art. Prior works require widths scaling with [1] $\Omega(N^6\lambda^{-4})$, [2] $\tilde{\Omega}(N^4\lambda^{-4})$, and [3] $\tilde{\Omega}(N^2\lambda^{-2})$. Our result achieves $\tilde{\Omega}(N\lambda^{-1})$. Therefore, we have reduced the dependence on $1/\lambda$ from quartic or quadratic to **linear**, in addition to reducing the dependence on sample size $N$ to linear.
>
> Furthermore, as noted in Remark 3.4, prior work [4] has shown that for standard data distributions (e.g., uniform on the sphere), $\lambda$ is well-behaved ($\tilde{\Omega}(1)$) in moderately high dimensions $d\gtrsim \log N$ and thus does not act as a bottleneck.
>
> > **W2 & Q2: Extension to deep networks and other losses.**
>
> We acknowledge that our current theoretical guarantees are limited to two-layer networks with squared loss. As noted in Section 5, extending these results to deeper architectures and losses like cross-entropy is a next step to align with modern practice.
>
> However, we emphasize that establishing linear-width convergence for **non-smooth ReLU networks** has been a major theoretical bottleneck even for the two-layer case. Prior to this work, the gap between the conjectured linear width and the best-known polynomial bounds was significant. By resolving this open problem, we establish the necessary analytical foundation for deeper architectures.
>
> Specifically, our work introduces the **unbalanced scaling** technique to tightly control kernel shifts. The primary theoretical obstacle for deep networks is the recursive accumulation of these shifts across layers. We believe our technique of applying unbalanced scaling layer-wise to dampen "feature movement" is a potential approach to extend these results, though the increased algebraic complexity may require significantly more effort in future work.
>
> [1] Du, Simon S., et al. "Gradient descent provably optimizes over-parameterized neural networks." ICLR 2018.
>
> [2] Song, Z. and Yang, X. "Quadratic suffices for over-parametrization via matrix chernoff bound" arXiv 2019
>
> [3] Munteanu, Alexander, et al. "Bounding the width of neural networks via coupled initialization a worst case analysis." ICML 2022.
>
> [4] Karhadkar, Kedar et al. “Bounds for the smallest eigenvalue of the NTK for arbitrary spherical data of arbitrary dimension.” NeurIPS 2024.

---

### Official Review · Reviewer_ZAZi · 2025-11-09

**Soundness:** 3
**Presentation:** 3
**Contribution:** 2
**Rating:** 4
**Confidence:** 2

**Summary:**

This work studies the convergence rate and required network width of ReLU networks in the NTK regime. Under the assumption of unbalanced initialization, they proved that the required network width scales linearly with the number of data samples, up to a logarithmic factor.
Furthermore, they provided an accelerated convergence result under the Nesterov’s Accelerated Gradient (NAG) setting.

**Strengths:**

To the best of my understanding, the result for non-differentiable activation functions is novel. Although the reviewer has not fully verified the details of the proof, no obvious errors were found. The convergence result of the NTK with ReLU activation clearly contributes to our understanding of neural networks. In addition, the extension to the multidimensional case is a valuable and meaningful result.

**Weaknesses:**

I believe that the justification for the unbalanced initialization is rather insufficient.
There are two major conditions that must be satisfied under this assumption.
First, the matrices $W$ and $V$ are required to satisfy specific *symmetry* and *anti-symmetry* conditions.
Second, the coefficients $c_1$ and $c_2$ must satisfy three inequalities given in Equation (5).
In my opinion, both assumptions are somewhat artificial and overly restrictive in order to make the theoretical result hold.

Regarding the inequality conditions, the requirement $c_1 c_2 = \Omega(\sqrt{N})$ significantly deviates from the initialization schemes commonly used in practice (e.g., He initialization).
Moreover, the paper provides insufficient discussion on the plausibility of the symmetry and anti-symmetry constraints imposed on $W$ and $V$.
Are these assumptions realistic? Do real-world neural network initializations actually satisfy them? (I would argue that they clearly do not.)
If they do not, how sensitive are the main results to deviations from these assumptions?

The paper does not appear to provide a satisfactory answer to these important questions.

When the data $Y$ exhibits a low-rank structure, assuming that the initialization already aligns with $Y$ is an overly strong assumption.
The authors should discuss how the training dynamics behave---specifically, whether and how the network converges to the desired subspace---when the initialization is not already within that subspace.

**Questions:**

1. Can the unbalanced initialization be theoretically or empirically justified?

2. If not, how could the result be improved by relaxing or weakening this assumption?

3. Could the authors provide a more appropriate explanation for the initialization associated with $Y$?

---

> ### Author Response · Authors · 2025-11-21
>
> We thank the reviewer for their valuable comments. We summarize the concerns and questions and respond to them below.
>
> > **W1 & Q1/Q2: Justification of unbalanced/coupled/large initialization.**
>
> We understand the concerns regarding the initialization assumptions. We address the three components (coupled, unbalanced, and large scaling) separately:
>
> **1. Coupled Initialization (Symmetry/Anti-Symmetry):**
> This is a standard theoretical device adopted in prior works (e.g., [1]) to reduce the width requirement. Its primary benefit is that it makes the initial residual $R_0 \equiv Y$ independent of the network width $m$ and the scaling factors $c_1, c_2$. This allows us to choose scaling parameters freely without generating an explosive residual.
> Technically, without coupling, the initial residual scales with $c_1 c_2 \sqrt{m}$. This term would enter the sufficient conditions (Eqs. 23-28) and eventually break the feasibility of the near-linear width requirement (i.e., no feasible solution would exist for $c_1, c_2$ that satisfies both the kernel regime and the near linear width constraint).
> While convergence can happen in practice without coupling and large scaling (see point 3), applying coupled initialization provides the theoretical guarantee of a configuration-agnostic initial error, essential for the proof.
>
> **2. Unbalancedness ($c_1 > c_2$):**
> This ensures the network trains in the NTK regime rather than the frozen Random Feature regime (Remark 3.3). Our experiments in the paper (Figures 3 and 4) empirically justify this choice.
> **Figure 3** shows that our scaling ($c_1 > c_2$) allows the hidden layer to move more (more feature change), and **Figure 4** demonstrates that this regime yields significantly faster convergence and a better-conditioned kernel compared to the opposite ($c_1 \le c_2$). Thus, the unbalancedness provides a concrete optimization benefit.
>
> **3. Large Scaling ($c_1 c_2$):**
> We acknowledge that the theoretical condition $c_1 c_2 = \Omega(\sqrt{N})$ is a sufficient condition derived to ensure high-probability convergence in the worst case. In practice, moderate scaling often suffices. Nevertheless, the benefit of unbalancedness *is robust* to this scale.
> For instance, **Figure 3 (Left)** in our paper sweeps $c_1 c_2$ from $0.1$ to $100$. Even at $c_1 c_2 = 0.1$ (which violates the theoretical lower bound of $\approx 10$), our unbalanced scaling ($c_1 \gg c_2$) consistently leads to more hidden layer weight change (feature movement). Similarly, **Figure 4** fixes $c_1 c_2 = 1$ (small scaling) and shows successful convergence. Therefore, our main results are not limited to the large scaling condition; this particular condition, unlike the unbalanced one, is an artifact of the worst-case analysis which could potentially be relaxed with more sophisticated techniques.
>
> To further justify unbalanced initialization in the small scaling setting beyond our theory, we conduct experiments similar to the setting in Figure 3 and 4, and summarize the loss and weight change after NAG 200 steps in the following tables.
>
> **Table 1: Training Loss**
> | Scaling ($c_1 c_2$) | $c_1/c_2=0.01$ | $c_1/c_2=1$ | $c_1/c_2=100$ (Ours) |
> | :--- | :--- | :--- | :--- |
> | **0.001** | 1.21e-04 | nan | **8.80e-10** |
> | **0.01** | 9.88e-05 | 1.03e-11 | **6.74e-13** |
> | **0.1** | 1.25e-04 | 5.35e-10 | **3.64e-11** |
> | **1.0** | 7.97e-05 | 4.21e-08 | **3.91e-09** |
> | **10.0** | 9.80e-05 | 2.77e-06 | **3.50e-07** |
>
> **Table 2: Relative Weight Change ($\|W_T - W_0\|_F / \|W_0\|_F$)**
>
> | Scaling ($c_1 c_2$) | $c_1/c_2=0.01$ | $c_1/c_2=1$ | $c_1/c_2=100$ (Ours) |
> | :--- | :--- | :--- | :--- |
> | **0.001** | 2.52e-03 | nan | **3.20e+00** |
> | **0.01** | 9.11e-05 | 1.75e-01 | **2.55e-01** |
> | **0.1** | 8.54e-06 | 1.71e-02 | **2.30e-02** |
> | **1.0** | 6.36e-07 | 1.69e-03 | **2.35e-03** |
> | **10.0** | 2.16e-08 | 1.70e-04 | **2.29e-04** |
>
> As shown in the tables, our unbalanced scaling exhibits faster convergence and allows more feature movement across a wide range of overall scaling $c_1c_2$.
>
> > **W2 & Q3: Low-rank initialization assumption.**
>
> We clarify that the initialization's alignment with $Y$ is not an assumption, but a *consequence* of the sketching initialization in Eq. (8), supported by Lemma 4.2. This sketching step is an algorithmic enhancement explicitly designed to exploit low-rank structure. It ensures that the optimization starts *and stays* (via Lemma 4.5) in the relevant subspace.
>
> If we were to initialize as Eq. (4) without the sketching step, the dynamics would eventually align with the top singular vectors of $Y$, but the "effective" dimension during the early phase would be the ambient dimension $n$. This would necessitate the larger width derived in Theorem 3.2 to avoid kernel degradation before alignment occurs.
>
>
> [1] Munteanu, Alexander, et al. "Bounding the width of neural networks via coupled initialization a worst case analysis." ICML 2022.

---

### Official Review · Reviewer_f6gq · 2025-11-10

**Soundness:** 3
**Presentation:** 2
**Contribution:** 2
**Rating:** 6
**Confidence:** 3

**Summary:**

This work analyzes convergence for near-linear width two-layer ReLU networks using unbalanced initialization. With the same initialization scheme, the paper proves Nesterov accelerated gradient enjoys the usual quadratic speedup while still only needing near-linear width unlike prior work which needed much larger width. In addition, the paper extends the convergence analysis to low-rank kernel regimes via a subspace analysis.

**Strengths:**

The unbalanced initialization scheme is interesting as in practice (or at least in pytorch default) the initialization scheme is also unbalanced (namely for a two layer ReLU network with width m, the outer layer is scaled by $\frac{1}{\sqrt{m}$ while the inner layer is scaled by $\frac{1}{\sqrt{d}}$).

The extension to the low rank setting as well as the techniques used to deal with this setting are interesting.

**Weaknesses:**

The weaknesses are readability of the results and some correctness concerns regarding the proof
(see questions).

**Questions:**

1. What are the constants $C_1$ and $C_2$ (as used in lemma 3.12)?

2. In Theorem 3.1 and 3.2, can you clarify the dependence of $m$ on $c_1$ and $c_2$?

3. Why is there no upper bound constraint on $c_1$ in Theorem 3.1 and 3.2? To be more concrete,
the gradients of the risk with respect to the inner layer $W$ scales with $c_1$. Hence, the amount that the inner layer moves (namely $R_2$) should scale with $c_1$. If $c_1$ is allowed to be arbitrarily large, every activation can change and hence the kernel shift can be arbitrarily bad. Am I missing something here?

---

> ### Author Response · Authors · 2025-11-21
>
> We thank the reviewer for their positive comments and valuable feedback. We address the concerns and questions regarding the readability and correctness below.
>
> > **Q1: What are the constants $C_1$ and $C_2$ (as used in lemma 3.12)?**
>
> $C_1$ is a constant used to bound the residual norm in the induction hypothesis, specifically $\|r_t\| \le C_1 \theta^t \|Y\|_F$ in Lemma 3.11. In our proofs (Appendices G.4 and G.5), setting $C_1 = 2$ for GD and $C_1 = 2\sqrt{2}$ for NAG suffices.
> $C_V$ and $C_W$ (likely referred to as $C_2$ in the question) are upper bounds on the norms of the weight matrices during training. Specifically: $C_V := \|X\|_F (R_V + R_1)$ and $C_W := \|X\|_F (R_W + R_2)$, where $R_1$ and $R_2$ are the norms of weight changes, and $R_V$ and $R_W$ are the norms of weight matrices at initialization.
>
>
> > **Q2: In Theorem 3.1 and 3.2, can you clarify the dependence of $m$ on $c_1$ and $c_2$?**
>
> The lower bound on width $m$ depends on the data properties ($N, n, \lambda$) but is **independent** of the specific values of $c_1$ and $c_2$, provided they satisfy the threshold conditions in Eq. (5) and (6).
> Specifically, the width requirement $m = \tilde{\Omega}(Nn/\lambda)$ mainly comes from the concentration of the empirical kernel $K_0$ (Lemma 3.6), which is scale-invariant (the factors $c_1^2$ cancel out in the relative eigenvalue bounds). The scaling factors $c_1, c_2$ appear in the analysis to ensure the weight updates are small enough to maintain this kernel stability. As long as $c_1, c_2$ are sufficiently unbalanced (satisfying the lower bounds in Eq. 6), the "weight movement" constraints on $m$ are looser than the concentration constraint (e.g., Eq. (35)). Thus, increasing the unbalancing ratio further does not increase the required width.
>
> > **Q3: Why is there no upper bound constraint on $c_1$ ... If $c_1$ is allowed to be arbitrarily large, every activation can change ... Am I missing something here?**
>
> This is an excellent question. You are correct that the gradient $\nabla_W \mathcal{L}$ scales with $c_1$, and that activation changes depend on the **relative** weight movement (Lemma 3.10). However, the **step size** $\eta$ also scales with $c_1$ in our setting, which acts as a counter-balance that actually suppresses these relative changes as $c_1$ increases.
>
> Specifically, the smoothness constant scales as $L = \lambda_1(K_0) \propto c_1^2$. Thus, our step size $\eta = 1/L$ scales with $1/c_1^2$. The **absolute** movement of the inner layer $W$ (bounded by $R_2$) scales as:
> $$R_2 \approx \sum \eta \|\nabla_W \mathcal{L}\| \propto \frac{1}{c_1^2} \cdot c_1 = \frac{1}{c_1}$$
> Crucially, Lemma 3.10 states that the number of activation flips depends on the **relative** change $R_2/c_2$. Since $R_2 \propto 1/c_1$, this relative change scales as:
> $$\frac{R_2}{c_2} \propto \frac{1}{c_1 c_2}$$
> Therefore, increasing $c_1$ (unbalancing the network) actually **decreases** the relative weight movement, thereby strictly restricting the change in activation patterns. Similarly, while the absolute kernel shift $\|K_0 - K_t\|$ may grow, the relevant quantity for convergence is the perturbation to the dynamics, $\eta \|K_0 - K_t\|$, which scales as $\frac{1}{c_1^2} \cdot c_1^2 \frac{R_2}{c_2} \propto \frac{1}{c_1 c_2}$, which also vanishes. This mechanism ensures the kernel regime is maintained.

---

### Meta-Review · Area_Chair_YxJN · 2026-01-03

**Summary:**

The reviewers had the following concerns:
1. Limited significance and practical relevance. Since the paper introduced a specific symmetric/antisymmetric and unbalanced initialization for the weights, the reviewers are concerned whether random initialization would satisfy those conditions, and whether the theoretical insights would apply to networks with standard random initialization. Moreover, one reviewer is unclear whether the result can be extended to deep networks.
2. Potential issues with the theoretical results. Reviewers asked clarification questions on the results for the case of low-rank Y. Reviewers had technical questions about how the initialization scale depends on the width. Reviewers raised concerns about the usage of certain lemmas from prior works.

**Reviewer Concerns:**

Since the authors have answered all the clarification questions about the theoretical results satisfactorily in their rebuttal, I consider the second concern addressed.

However, I believe the first concern is still outstanding. During the rebuttal, the authors have explained the technical necessity of the proposed initialization scheme (to control the output, to control the movement of NTK), with some additional experiments (under the same setup as the theory). Yet the reviewers' concerns were primarily about practical relevance (networks under standard initialization, and those with larger depth), which are not well addressed by the rebuttal.

As such, although the technical contributions in this paper are solid, their significance and practical relevance remain unclear.

**Reviewer Scores:**

I think the reviewer CSXy would have raised their score (4 to 6), since their concern about the technical lemmas is addressed. I think the other reviewers f6gq (6), ZAZi (4), and qQdC (4) would keep their scores.

---

### Decision · Program_Chairs · 2026-01-26

Reject